# Bridging Fairness and Explainability: Can Input-Based Explanations Promote Fairness in Hate Speech Detection?

**Yifan Wang[1], Mayank Jobanputra[1], Ji-Ung Lee[1], Soyoung Oh[1],**
**Isabel Valera[1,3], Vera Demberg[1,2]**
[1] Saarland University, Germany
[2] Max Planck Institute for Informatics, Saarland Informatics Campus, Germany
[3] Max Planck Institute for Software Systems, Saarland Informatics Campus, Germany
{yifwang,mayank,soyoung,vera}@lst.uni-saarland.de,
ji-ung.lee@uni-saarland.de, ivalera@cs.uni-saarland.de

## Abstract

Natural language processing (NLP) models often replicate or amplify social bias from training data, raising concerns about fairness. At the same time, their black-box nature makes it difficult for users to recognize biased predictions and for developers to effectively mitigate them. While some studies suggest that input-based explanations can help detect and mitigate bias, others question their reliability in ensuring fairness. Existing research on explainability in fair NLP has been predominantly qualitative, with limited large-scale quantitative analysis. In this work, we conduct the first systematic study of the relationship between explainability and fairness in hate speech detection, focusing on both encoder- and decoder-only models. We examine three key dimensions: (1) identifying biased predictions, (2) selecting fair models, and (3) mitigating bias during model training. Our findings show that input-based explanations can effectively detect biased predictions and serve as useful supervision for reducing bias during training, but they are unreliable for selecting fair models among candidates. Our code is available at https://github.com/Ewanwong/fairness_x_explainability.

## 1 Introduction

Language models (LMs) pre-trained on large-scale natural language datasets have shown great capacities in various NLP tasks (Wang et al., 2018; Gao et al., 2023). However, previous studies have shown that they can replicate and amplify stereotypes and social bias present in their training data and demonstrate biased behaviors (Sheng et al., 2021; Gupta et al., 2024; Gallegos et al., 2024). Such behaviors risk the underrepresentation of marginalized groups and the unfair allocation of resources, raising serious concerns in critical applications (Blodgett et al., 2020).

Meanwhile, current NLP models are mostly based on black-box neural networks. Despite their strong capacities, the complex architecture and large number of parameters of these models make it hard for humans to understand their behaviors (Bommasani et al., 2021). To understand neural NLP models, different types of explanations have been devised, such as input-based explanations (Yin & Neubig, 2022; Deiseroth et al., 2023; Madsen et al., 2024; Wang et al., 2025b), natural language explanations (Ramnath et al., 2024; Wang et al., 2025a), and concept-based explanations (Yu et al., 2024; Raman et al., 2024). Among these, input-based explanations, often referred to as rationales, indicate the contribution of each token to models' predictions, and thus provide the most direct insights into models' behaviors (Arras et al., 2019; Atanasova et al., 2022; Lyu et al., 2024).

Explainability has long been deemed critical to improving fairness. Researchers believe that if the use of sensitive features is evidenced by model explanations, then they can easily detect biased predictions and impose fairness constraints by guiding models to avoid such faulty reasoning (Meng et al., 2022; Sogancioglu et al., 2023). However, recent studies have challenged this assumption, suggesting that the relationship between explainability and fairness is complex and that explanations

may not always reliably detect or mitigate bias (Dimanov et al., 2020; Slack et al., 2020; Pruthi et al., 2020). Unfortunately, to the best of our knowledge, current studies are mostly limited to qualitative analysis on a small set of explanation methods (Balkir et al., 2022; Deck et al., 2024). Our work takes a step toward bridging explainability and fairness by providing the first comprehensive quantitative analysis in the context of hate speech detection, a task where both fairness and explainability are particularly critical. Specifically, we address the following three research questions to investigate the role of explainability in promoting fairness within the task of hate speech detection:

- **RQ1: Can input-based explanations be used to identify biased predictions?**
- **RQ2: Can input-based explanations be used to automatically select fair models?**
- **RQ3: Can input-based explanations be used to mitigate bias during model training?**

Our experiments demonstrate that input-based explanations can effectively detect biased predictions (RQ1), are less reliable for automatic fair model selection (RQ2), and can help reduce bias during model training (RQ3). Furthermore, our analyses indicate that explanation-based bias detection remains robust even when models are trained to reduce reliance on sensitive features, and that these explanations outperform LLM judgments in identifying bias.

## 2 RELATED WORK

**Bias in NLP**  The presence of social bias and stereotypes has significantly shaped human language and LMs trained on it (Blodgett et al., 2020; Sheng et al., 2021). As a result, these models often exhibit biased behaviors (Gallegos et al., 2024), such as stereotypical geographical relations in the embedding space (Bolukbasi et al., 2016; May et al., 2019) and stereotypical associations between social groups and certain concepts in the model outputs (Fang et al., 2024; Wan & Chang, 2025). More critically, disparities in model predictions and performance across social groups (Zhao et al., 2018; Sheng et al., 2019) can significantly compromise user experiences of marginalized groups and risk amplifying bias against them, therefore drawing great concerns in critical use cases.

**Input-based Model Explanations**  Input-based explanations in NLP models aim to attribute model predictions to each input token (Lyu et al., 2024). They can be broadly categorized based on how they generate explanations: gradient-based (Simonyan et al., 2014; Kindermans et al., 2016; Sundararajan et al., 2017; Enguehard, 2023), propagation-based (Bach et al., 2015; Shrikumar et al., 2017; Ferrando et al., 2022; Modarressi et al., 2022; 2023), perturbation-based (Li et al., 2016; Ribeiro et al., 2016; Lundberg & Lee, 2017; Deiseroth et al., 2023), and attention-based methods (Bahdanau et al., 2015; Abnar & Zuidema, 2020). While most prior work has focused on encoder-only models, recent studies have also explored explaining the behaviors of generative models (Yin & Neubig, 2022; Ferrando et al., 2022; Enouen et al., 2024; Cohen-Wang et al., 2024).

**Bridging Explainability and Fairness**  Explainability is often considered essential for achieving fairness in machine learning systems (Balkir et al., 2022; Deck et al., 2024). One line of research investigates model bias by analyzing explanations (Prabhakaran et al., 2019; Jeyaraj & Delany, 2024; Sogancioglu et al., 2023). For instance, Muntasir & Noor (2025) shows that a biased model relied on gendered words as key features in its predictions, as revealed by LIME explanations. Similarly, Stevens et al. (2020) demonstrates that biased models often place high importance on gender and race features when examined with SHAP explanations. Extending this line of evidence, Meng et al. (2022) finds that features with higher importance scores are associated with larger disparities in model performance on a synthetic medical dataset using deep learning models.

Another line of research focuses on mitigating bias with explanations (Dimanov et al., 2020; Kennedy et al., 2020; Rao et al., 2023; Liu et al., 2024). For example, Hickey et al. (2020) improves fairness by reducing reliance on sensitive features during training with SHAP explanations. Bhargava et al. (2020) and González-Silot et al. (2025) first identify predictive sensitive features using LIME and SHAP, respectively, and then remove them prior to model training. Relatedly, Grabowicz et al. (2022) traces unfairness metrics to input features and adjusts them to mitigate bias.

However, recent research has challenged the assumption that input-based explanations can be reliably used to detect and mitigate bias. First, current explanation methods may be unfaithful, meaning

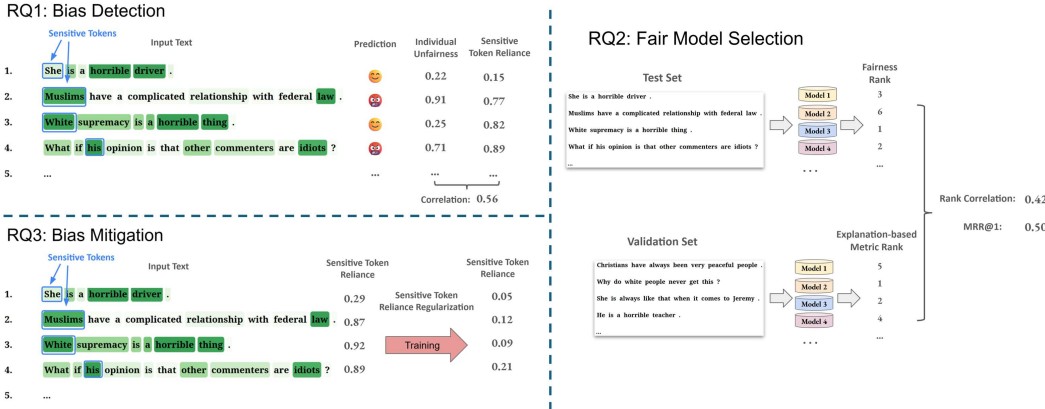

Figure 1: Workflow diagram illustrating the processes used to address each research question. Sensitive tokens are shown in blue boxes, and the intensity of the green shading reflects each word's contribution to the model's prediction.

that they may not always reflect the true decision-making process of models (Jain & Wallace, 2019; Ye et al., 2025). This makes it difficult to reliably detect the use of sensitive features in predictions. Second, efforts to reduce the influence of sensitive features can lead to unintended consequences, sometimes degrading both task performance and fairness of models (Dimanov et al., 2020). Finally, models can be deliberately trained to assign lower importance to sensitive features, thereby masking biased predictions when explanations are inspected (Slack et al., 2020; Pruthi et al., 2020).

Despite growing interest in this topic, most existing work remains qualitative or restricted to limited setups (Balkir et al., 2022; Deck et al., 2024). To the best of our knowledge, this is the first study to quantitatively and comprehensively examine the relationship between explainability and fairness in NLP models. We focus on hate speech detection as a particularly critical application. Prior research has shown that biased NLP models often rely on demographic information such as race and gender, leading to inferior performance on marginalized groups in this task (Sap et al., 2019; Mathew et al., 2021). Detecting and mitigating such biased behaviors are therefore essential for equitable social media participation across all groups. We define hate speech and social bias and review related fairness and explainability work in hate speech detection in Appendix A.

## 3 EXPERIMENTAL SETUP

**Notations** Let an input text $\mathbf{x}$ consist of tokens $t_1, t_2, \ldots, t_n$. The task of hate speech detection is to predict a binary label $\hat{y} \in \{\text{toxic}, \text{non-toxic}\}$. A classifier outputs the probability of class $c$ as $f_c(x)$, where $f$ is implemented by a neural model.

In the context of social bias, we assume that a bias type (e.g., race) involves a set of social groups $G$ (e.g., black, white, ...). A subset of tokens $t_{g_1}, t_{g_2}, \ldots, t_{g_m}$ in $\mathbf{x}$ denotes the sensitive feature $g \in G$ of the speaker or target. We refer to these tokens as *sensitive tokens*. By replacing the sensitive tokens of group $g$ with those of another group $g'$, we obtain a counterfactual version of $\mathbf{x}$ that refers to $g'$, denoted as $\mathbf{x}^{(g')}$.

An input-based explanation assigns an attribution score to each token in $\mathbf{x}$ for class $c$: $a_1^c, a_2^c, \ldots, a_n^c$, indicating their contribution to the prediction of class $c$. Following Dimanov et al. (2020), we compute attribution scores on the sensitive tokens, $a_{g_1}^c, a_{g_2}^c, \ldots, a_{g_m}^c$, which we refer to as the *sensitive token reliance* scores. To handle cases where multiple sensitive tokens appear in the same sentence, we take the maximum absolute attribution value as the reliance score for that example[1]:

$$\text{sensitive token reliance}(\mathbf{x}, c) = a_{j^*}^c, \text{ where } j^* = \underset{j \in \{g_1, \ldots, g_m\}}{\arg\max} \left| a_j^c \right|$$

---

[1]We have experimented with normalizing feature importance scores but found that using raw scores yielded the best results. We also evaluated sum and average aggregation methods beyond taking the max absolute value and observed similar outcomes.

**Datasets and Vocabulary**   We use two hate speech detection datasets: Civil Comments (Borkan et al., 2019) and Jigsaw (cjadams et al., 2019). To ensure coverage, we focus on three bias types and their associated groups: race (black/white), gender (female/male), and religion (Christian/Muslim/Jewish). We include examples containing identity-marking terms but exclude those with derogatory or slur-based references, as the latter can reasonably serve as direct evidence for toxic predictions. The sensitive token vocabulary is derived from Caliskan et al. (2017) and Wang & Demberg (2024a). Further details on dataset pre-processing are provided in Appendix B.

**Models**   We evaluate two major classes of NLP models: encoder-only models (BERT (Devlin et al., 2019) and RoBERTa (Liu et al., 2019)) and decoder-only large language models (Llama3.2-3B-Instruct (Dubey et al., 2024), Qwen3-4B, and Qwen3-8B (Yang et al., 2025a), all of which are aligned to human values). We fine-tune encoder-only models on data subsets that either target a single bias type or combine all bias types. For decoder-only models, we use an instruction-based setup where the model is prompted to decide whether a test example contains hate speech. The prompt includes the definition of hate speech, the test example, and a corresponding question. As a baseline, we adopt the zero-shot setting as the default configuration.

Beyond conventional fine-tuning and prompting, we also investigate the interaction between explainability and fairness in debiased models. For encoder-only models, we apply pre-processing techniques such as group balance (Kamiran & Calders, 2012), group-class balance (Dixon et al., 2018), and counterfactual data augmentation (CDA, Zmigrod et al., 2019), as well as in-processing techniques including dropout (Webster et al., 2020), attention entropy (Attanasio et al., 2022), and causal debias (Zhou et al., 2023). For decoder-only models, we incorporate bias reduction through prompt design, including few-shot, fairness imagination (Chen et al., 2025), and fairness instruction prompting (Chen et al., 2025). We do not include reasoning models and chain-of-thought prompting, as we find that their predictions are primarily attributed to intermediate reasoning steps rather than the input text, which complicates analysis and falls beyond the scope of this work. Further details are provided in Appendix B.

**Fairness Metrics**   We evaluate fairness in model predictions using two categories of metrics: **group fairness** and **individual fairness**. Group fairness metrics capture disparities in performance across demographic groups:

$$\text{Disp}_{\text{metric}} = \sum_{g \in G} |\text{metric}_g - \overline{\text{metric}_G}|,$$

where $\overline{\text{metric}_G}$ is the average metric value across all groups $G$ in a bias type. We specifically measure disparities in accuracy (ACC), false positive rate (FPR), and false negative rate (FNR).

Individual fairness measures the extent to which a model's prediction for a given example changes when the associated social group is altered. To maintain consistency with the direction of group fairness metrics, we compute the individual unfairness (IU) score of $\mathbf{x}_i$ and the predicted class $\hat{y}_i$:

$$\text{IU}(\mathbf{x}_i) = |f_{\hat{y}_i}(\mathbf{x}_i) - \frac{1}{|G \backslash \{g_i\}|} \sum_{g' \in G \backslash \{g_i\}} f_{\hat{y}_i}(\mathbf{x}_i^{(g')})|$$

The Average IU score ($\text{Avg}_{\text{iu}}$) is then computed over a dataset to reflect the overall level of individual unfairness in a model.

For both types of metrics, higher scores indicate more bias in model predictions. It is worth noting that individual unfairness can be evaluated at the level of each example, whereas group fairness metrics are defined over sets of validation or test examples. To compute the fairness metrics, we randomly sample a subset of examples for each bias type such that each social group contributes an equal number of examples. Further details on test set sampling are provided in Appendix B.

**Explanation Methods**   We employ 16 variants of commonly used input-based post-hoc explanation methods, selected to represent a diverse range of methodological categories: Attention (Bahdanau et al., 2015), Attention rollout (Attn rollout, Abnar & Zuidema, 2020), Attention flow (Attn flow, Abnar & Zuidema, 2020), Gradient (Grad, Simonyan et al., 2014), Input x Gradient (IxG, Kindermans et al., 2016), Integrated Gradients (IntGrad, Sundararajan et al., 2017), Occlusion (Li et al.,

2016), DeepLift (Shrikumar et al., 2017), KernelSHAP (Lundberg & Lee, 2017), DecompX (Modarressi et al., 2023), and Progressive Inference (ProgInfer, Kariyappa et al., 2024)[2]. For methods that attribute predictions to embeddings, we aggregate attribution scores into a single feature importance value using either the mean or the L2 norm. For Occlusion, we additionally report results obtained by taking the absolute value of each attribution score prior to computing sensitive token reliance scores (denoted as Occlusion abs). The time and GPU memory costs for each method are shown in Appendix C. We also study rationales generated by LLMs and find that they are not as reliable as input-based explanations in detecting bias (Section 6).

Table 1: Task performance and fairness of default and debiased models on Civil Comments. Results are provided for race/gender/religion biases. Green (red) indicates the results are better (worse) than the default/zero-shot models. No debiasing method consistently reduces bias across all metrics and bias types.

| Model | Method | Accuracy ($\uparrow$) | $\text{Disp}_{acc}(\downarrow)$ | $\text{Disp}_{fpr}(\downarrow)$ | $\text{Disp}_{fnr}(\downarrow)$ | $\text{Avg}_{iu}(\downarrow)$ |
|---|---|---|---|---|---|---|
| BERT | Default | 78.38/88.05/85.93 | 2.05/3.30/18.07 | 0.50/0.03/5.77 | 10.04/11.98/30.9 | 3.17/0.66/1.27 |
| | Group balance | 79.25/87.25/86.83 | 3.10/2.80/13.53 | 0.25/1.73/11.53 | 10.46/5.38/30.31 | 3.79/0.42/2.01 |
| | Group-class balancing | 78.00/87.02/85.77 | 1.80/2.75/14.73 | 2.42/0.99/3.09 | 10.63/7.26/33.14 | 4.43/0.98/0.71 |
| | CDA | 76.83/86.70/84.83 | 2.35/3.60/14.13 | 5.88/2.00/5.67 | 18.45/7.57/24.12 | 0.50/0.50/0.90 |
| | Dropout | 78.53/88.20/85.03 | 2.25/2.10/15.67 | 0.78/1.46/5.93 | 10.82/3.50/27.16 | 3.43/0.52/1.51 |
| | Attention entropy | 79.15/87.67/84.93 | 2.60/2.05/15.07 | 0.99/0.10/4.99 | 11.71/7.11/26.52 | 2.95/0.67/1.58 |
| | Causal debias | 78.80/86.17/86.40 | 0.00/2.65/16.40 | 3.90/0.46/8.82 | 7.98/10.67/30.46 | 3.83/0.48/2.10 |
| Qwen3-4B | Zero-shot | 69.55/79.75/77.50 | 0.60/0.00/17.40 | 7.13/1.40/21.07 | 13.25/3.71/5.17 | 2.55/2.41/3.32 |
| | Few-shot | 70.15/80.73/79.53 | 1.80/0.65/18.93 | 10.02/2.50/19.31 | 11.89/9.15/5.57 | 3.18/3.34/3.76 |
| | Fairness imagination | 71.23/80.40/80.83 | 0.85/1.00/18.27 | 4.03/2.11/10.51 | 11.62/9.21/4.28 | 2.98/3.16/2.20 |
| | Fairness instruction | 70.40/79.77/80.47 | 0.60/1.35/19.33 | 4.30/0.39/4.67 | 11.11/5.24/5.08 | 2.02/1.83/1.71 |

## 4 QUANTITATIVE ANALYSES OF FAIRNESS AND EXPLAINABILITY

To comprehensively understand the relationship between explainability and fairness in NLP models, we examine three ways in which model explanations can be applied to promote fairness. The subsequent sections detail the experimental setups for each application and report the corresponding results. The workflow for our research questions is shown in Figure 1. For brevity, we report results on Civil Comments using BERT trained on single bias types and Qwen3-4B. We further demonstrate that our findings extend to other models and the Jigsaw dataset (Appendix D–G), and that they generalize across tasks, model alignment types, and sensitive token sets beyond those considered in the main experiments (Appendix H).

### 4.1 MODEL PERFORMANCE AND FAIRNESS

As a prerequisite, we first summarize the performance and fairness of the evaluated models. The results in Table 1 show that no single debiasing method consistently improves all fairness metrics. For BERT and Qwen3-4B, CDA and fairness instruction achieve the largest reductions in individual unfairness, yet they may simultaneously amplify biases on other metrics. Other debiasing methods show a similar pattern: they reduce bias for a specific metric or bias type, but the improvement does not generalize across different setups. These limitations underscore the importance of leveraging explanations for bias detection and mitigation. We find similar results for other models and for Jigsaw, which we provide in Appendix D along with a discussion on model performance and fairness.

### 4.2 RQ1: EXPLANATIONS FOR BIAS DETECTION

Our first research question asks whether explanations can be used to detect biased predictions. We address the question through three steps: (1) obtain model predictions and compute individual unfairness scores; (2) generate input-based explanations for the predictions; and (3) compute sensitive token reliance scores and evaluate their Pearson correlation with individual unfairness, which we refer to as *fairness correlation*. A higher fairness correlation indicates that the explanation method

---

[2] We apply DecompX only to encoder-only models and Progressive Inference only to decoder-only models, following the setups of the original papers.

is more effective in identifying predictions with high individual unfairness. To ensure robustness, we compute the fairness correlation separately for each prediction class–group pair and report the average absolute score as the final result for each explanation method.

We present results for default and debiased models where individual unfairness remains high after debiasing, as bias detection is particularly critical in these cases. Specifically, we report results for models with the highest average $\text{Avg}_{iu}$ scores across bias types, namely default, group balance, and causal debias for BERT, and zero-shot, few-shot, and fairness imagination prompting for Qwen3-4B. Results for religion as well as other models and the Jigsaw dataset are provided in Appendix E.

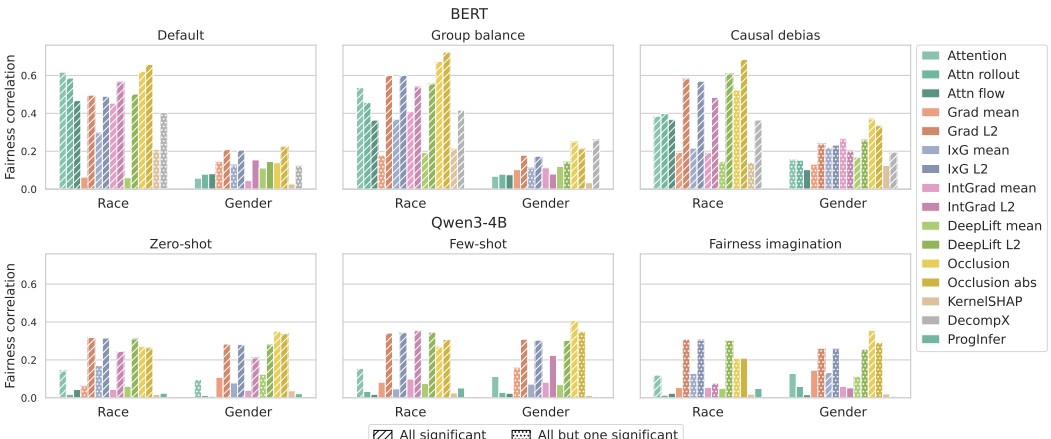

Figure 2: Fairness correlation results for each explanation method. Occlusion- and L2-based explanations are effective for bias detection across different bias types and models.

**Results**   Figure 2 shows that the best-performing explanation methods, such as Grad L2, IxG L2, DeepLift L2, Occlusion, and Occlusion abs, generally achieve high fairness correlations across different models and bias types, indicating a strong ability to detect biased predictions. Besides, their fairness correlations are mostly statistically significant ($p < \alpha = 0.05$) in all, or in all but one, class-group categories, which confirms their reliability. Among these methods, Occlusion and Occlusion abs perform best with BERT models, whereas the L2-based methods Grad L2, IxG L2, and DeepLift L2 are most effective with Qwen3-4B.

When comparing different variants of the same explanation family, mean-based approaches perform considerably worse than their L2-based counterparts, and also underperform compared to undirected attention-based methods. We attribute this limitation to their dependence on accurately determining the direction of each token's contribution, a challenge that attention- and L2-based explanations do not face. Our analysis in Appendix I further shows that the effectiveness of explanation-based bias detection is not determined by explanation faithfulness, underscoring the need for careful evaluation when selecting methods for bias identification. In addition, our human study in Appendix J shows that explanation-based signals can assist human fairness auditing.

**Takeaway**: Input-based explanation methods, particularly Occlusion- and L2-based ones, are effective for identifying biased predictions at inference time.

## 4.3   RQ2: EXPLANATIONS FOR MODEL SELECTION

Given that explanations can detect biased predictions (RQ1), we next investigate whether they can also be used to select fair models among candidates. Prior work has demonstrated that input-based explanations on validation examples can help humans identify spurious correlations in models (Lertvittayakumjorn & Toni, 2021; Pezeshkpour et al., 2022). Extending this idea, we examine whether explanations can be leveraged for automatic fair model selection, thereby removing the need for human intervention.

Our experiments consist of three steps: (1) for all default and debiased models (seven encoder-only and four decoder-only), we generate predictions on a validation set and compute explanation-based metrics; (2) we compute fairness metrics on the test set for each model; and (3) we evaluate model selection ability using two measures: Spearman's rank correlation ($\rho$) between validation set explanation-based metrics and test set fairness metrics, which reflects the ability to rank models, and mean reciprocal rank of the fairest model (MRR@1), which reflects the ability to select the fairest model. Higher rank correlations and MRR@1 indicate that an explanation method is useful for ranking models and selecting the fairest one. Specifically, we use the average absolute sensitive token reliance on the validation set as the explanation-based metric to rank and select models based on average individual unfairness on the test set.[3]

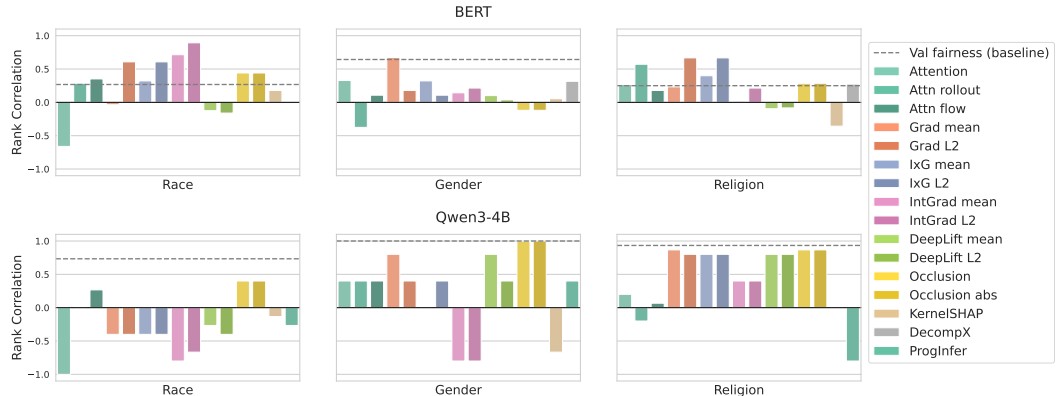

Figure 3: Rank correlations between validation set average absolute sensitive token reliance and test set individual unfairness. The validation set sizes are 500 for race and gender, and 200 for religion. None of the explanation methods consistently achieve performance on par with the baseline.

**Results** As a baseline, we report results of using the validation set average individual unfairness as the predictor of test set fairness performance. The results are averaged over six and three random validation set selections for encoder- and decoder-only models, respectively. Results for more models and the Jigsaw dataset are presented in Appendix F.

The results in Figures 3 and 4 highlight the limitations of using explanations for model selection. Although some methods occasionally show high rank correlations (e.g., Grad L2 for race and religion biases in BERT and Occlusion-based methods for gender and religion biases in Qwen3-4B),

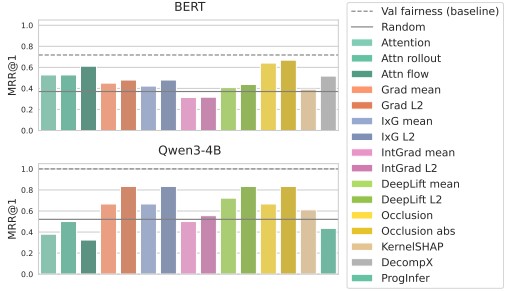

Figure 4: Average MRR@1 across bias types. Explanation methods perform worse than the baseline in identifying the fairest models.

none of them consistently reach the baseline of using the individual unfairness on the validation set. This limitation is particularly evident in decoder-only models, where the baseline achieves a perfect rank correlation of 1. Similarly, the baseline consistently achieves the highest MRR@1 scores, further showing the limited effectiveness of explanation-based methods in selecting the fairest models. Considering that these explanations are often more computationally expensive to generate than evaluating validation set fairness, they are not practically useful as a model fairness indicator. Therefore, we do not recommend explanation-based model selection, especially in decoder-only models. The difference in findings between RQ1 and RQ2 may stem from the fact that debiasing methods can alter model behaviors and thereby affect explanation attributions. As a result, comparing explana-

---

[3]We have evaluated other metrics to predict group fairness outcomes. However, neither explanation-based metrics nor validation set fairness achieved rank correlations beyond random chance with the test set results. The full set of evaluated metrics is provided in Appendix F.

tions across default and debiased models is less reliable, whereas comparing explanations within the same model remains effective for detecting biased predictions.

**Takeaway**: Input-based explanation methods are not reliable tools for selecting fair models.

## 4.4 RQ3: EXPLANATIONS FOR BIAS MITIGATION

Having shown that explanations can reliably reveal biased predictions (RQ1), we now investigate whether they can also be leveraged to mitigate model bias. Building on prior work demonstrating that explanation regularization can reduce spurious correlations while also improving performance and generalization (Kennedy et al., 2020; Rao et al., 2023), we investigate bias mitigation by minimizing sensitive token reliance during training. Following Dimanov et al. (2020), we define a debiasing regularization term, $L_{debias}$, which penalizes the average sensitive token reliance of all such tokens in an input, in addition to the task loss:

$$L = L_{task} + \alpha L_{debias}$$

Here, $\alpha$ is a hyperparameter that controls the strength of sensitive token reliance reduction. For embedding-level attributions, we apply either an L1 or L2 norm penalty, corresponding to minimizing mean- or L2-based reliance scores, respectively.

While Dimanov et al. (2020) tune hyperparameters based on task accuracy, we search $\alpha \in \{0.01, 0.1, 1, 10, 100\}$ using a fairness-balanced metric (the harmonic mean of accuracy and 100–unfairness) on the validation set[4]. Models are selected separately for each fairness criterion and results are averaged over three runs. Due to computational cost, we restrict training to single bias types. We exclude DeepLift, DecompX, and KernelSHAP, as they are not easily differentiable and thus cannot be incorporated into model training. Integrated Gradients is substantially more expensive in time and memory for generating explanations and tracking gradients, sp we apply them only to race bias mitigation in BERT and report the results in Table 7 in Appendix G. More implementation details are provided in Appendix B.

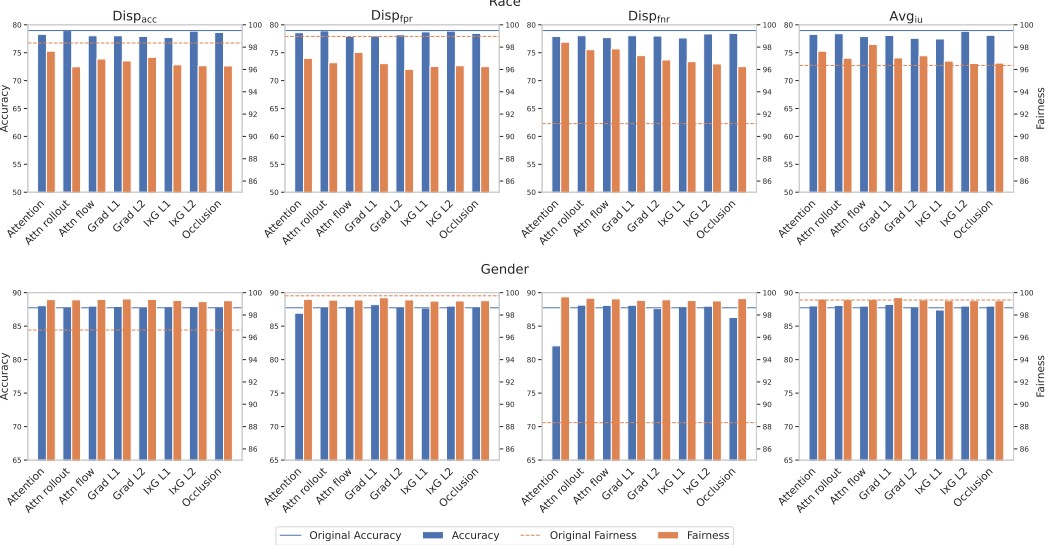

Figure 5: Accuracy and fairness results for bias mitigation using different explanation methods. Each column corresponds to models selected by maximizing the fairness-balanced metric with respect to the indicated bias metric. We find that explanation methods can improve fairness across many metrics while maintaining reasonable task accuracy.

---

[4]As Occlusion is sensitive to the debiasing strength, we use $\alpha \in \{0.002, 0.004, 0.006, 0.008, 0.01\}$.

**Results** In Figure 5, we present race and gender bias mitigation results. For consistency with accuracy, fairness results are reported as $100 - \{\text{Disp}_{\text{acc}}, \text{Disp}_{\text{fpr}}, \text{Disp}_{\text{fnr}}, \text{Avg}_{\text{iu}}\}$, so that higher values indicate lower bias. We find that explanation-based bias mitigation effectively improves fairness across multiple metrics. Most notably, it consistently and substantially reduces $\text{Disp}_{\text{fnr}}$ for all bias types. For gender bias, it also yields considerable reductions in $\text{Disp}_{\text{acc}}$, and $\text{Avg}_{\text{iu}}$ is mitigated for race bias. Moreover, as shown in Figure 19, all group fairness disparity metrics decrease for religion bias. The bias mitigation effects are consistent across all models and are also observed on the Jigsaw dataset (see Figures 19, 20, 21, 22 in Appendix G).

At the same time, explanation-based debiasing maintains a good balance between fairness and accuracy. For example, Grad L1 both increases accuracy and reduces $\text{Disp}_{\text{acc}}$, $\text{Disp}_{\text{fnr}}$, and $\text{Avg}_{\text{iu}}$ for gender bias, while most other explanation methods also achieve better $\text{Disp}_{\text{acc}}$ and $\text{Disp}_{\text{fnr}}$ with marginal or no accuracy loss. Our harmonic fairness–accuracy mean results (Figures 23, 24, 25, 26) further confirm this by showing that explanation-based debiasing almost always achieves comparable or higher harmonic means than both default models and traditional debiasing methods.

Among individual explanation methods, attention and attn flow achieve strong debiasing performance on BERT, while IxG L1 and L2 consistently yield a good balance between accuracy and fairness across models. Overall, IxG L2 and attention-based methods provide robust debiasing while maintaining a favorable fairness–accuracy trade-off across bias types, models, and datasets, as reflected in the harmonic mean results. Our findings differ from those of Dimanov et al. (2020), which we attribute to our fairness-based hyperparameter tuning strategy. We further show in Appendix K that combining traditional and explanation-based debiasing methods outperforms either approach in isolation.

> **Takeaway**: Input-based explanations can provide effective supervision for mitigating model bias during training while maintaining a good fairness–performance trade-off. In particular, IxG L2 and attention-based methods achieve robust debiasing with strong overall balance.

## 5 BIAS DETECTION IN EXPLANATION-DEBIASED MODELS

While explanation-based methods are effective in reducing bias (RQ3), their suppression of attributions on sensitive tokens could potentially mislead users into believing that model predictions are unbiased (Dimanov et al., 2020; Slack et al., 2020; Pruthi et al., 2020). To investigate this concern, we reapply the bias detection procedure from RQ1 to explanation-debiased models and compare their fairness correlations with those from the corresponding default models. For this analysis, we use the models debiased for race bias with respect to individual unfairness, as described in RQ3.

The fairness correlation differences from default models are shown in Figure 6. We observe that the impact

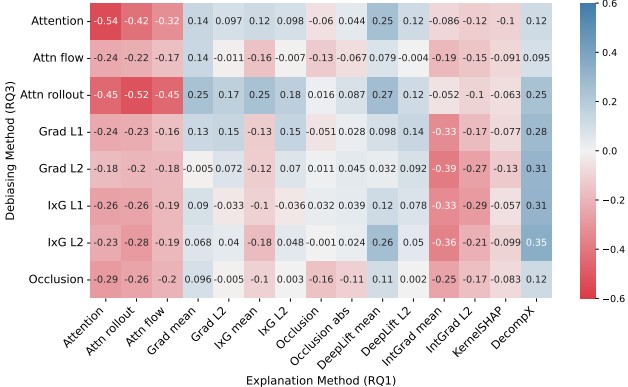

Figure 6: Fairness correlation differences between default and explanation-debiased BERT. Occlusion- and L2-based explanations (except IntGrad L2) are less affected by explanation-based debiasing and remain effective for bias detection.

of explanation-based debiasing on fairness correlations depends on both the explanations used for debiasing and those used for bias detection. Some approaches, such as Grad mean/L2, IxG L2, DeepLift mean/L2, Occlusion, and Occlusion abs, are only marginally, or even positively, affected by debiasing. Their fairness correlation scores (see Figure 32 in Appendix L) further indicate that Occlusion- and L2-based methods remain reliable for revealing bias in explanation-debiased models. In contrast, attention-based explanations experience substantial drops, particularly when the models themselves are debiased using attention-based methods. Similarly, IntGrad-based explanations show a reduced bias detection ability when the debiasing procedure is

also gradient-based. Overall, these findings demonstrate that certain input-based explanations remain effective for detecting biased predictions even in explanation-debiased models. Our results are different from those of Dimanov et al. (2020), likely because their analysis focused solely on attribution magnitudes without considering their relationship to fairness metrics.

## 6 EXPLANATION-BASED BIAS DETECTION VS. LLM-AS-A-JUDGE

Existing research suggests that LLMs could identify and correct biased model outputs (Bai et al., 2022; Furniturewala et al., 2024). In this section, we compare the bias detection ability of input-based explanations against LLMs' judgments under two paradigms: (1) LLM decision, where LLMs are asked to indicate whether a model's prediction rely on bias or stereotypes, and (2) LLM attribution, where LLMs choose a K-word rationale from the input, which we then examine for the presence of sensitive tokens. We conduct this analysis using two LLMs, Qwen3-4B and GPT-OSS-120B, on predictions made by Qwen3-4B on the race subset of Civil Comments (see Appendix B for the prompts used).

Table 2 shows the results of LLM-as-a-judge for bias detection. Under the LLM decision setup, Qwen3-4B is extremely conservative: it flags only 86 out of 4000 predictions as biased, and all of them correspond to toxic predictions. Moreover, the predictions labeled as biased by the model exhibit lower average individual unfairness than those labeled as non-biased, indicating poor precision as well. Under LLM attribution, Qwen3-4B performs slightly better: predictions whose rationales contain sensitive tokens show higher average individual unfairness than those without. However, this still falls short of a simple input-based explanation baseline that flags the top 50% of predictions ranked by absolute Grad L2 reliance scores (Grad L2 Binary). The larger GPT-OSS-120B exhibits improved bias detection ability in the LLM decision setting, but its performance under LLM attribution remains comparable to Qwen3-4B and still substantially worse than input-based explanations. Overall, we conclude that input-based explanations are more reliable than LLM-as-a-judge for bias detection. This finding aligns with the observations of Yang et al. (2025b), who also report that LLM-as-a-judge is unreliable for bias detection.

Table 2: Results of LLM-as-a-judge for bias detection using Qwen3-4B and GPT-OSS-120B. Predictions come from Qwen3-4B on race-related Civil Comments examples. "Biased/Unbiased" denotes whether an example is judged as biased or unbiased by the LLM through LLM decision or LLM attribution. If the judgments are reliable, $Avg_{iu}$ should be higher for biased examples than unbiased ones. For LLM decision with Qwen3-4B, fairness correlation cannot be computed because the model labels no non-toxic predictions as biased. Input-based explanations reveal bias more reliably than LLM-as-a-judge.

| LLM | Method | # Biased/Unbiased | $Avg_{iu}$ (Biased/Unbiased) | Fairness Correlation |
|---|---|---|---|---|
| Qwen3-4B | LLM decision | 86/3914 | 0.065/2.59 | - |
| | LLM attribution (K=5) | 2063/1904 | 3.55/1.49 | 0.104 |
| | LLM attribution (K=10) | 2176/1474 | 2.93/1.56 | 0.070 |
| GPT-OSS-120B | LLM decision | 399/3601 | 4.42/2.35 | 0.051 |
| | LLM attribution (K=5) | 2153/1843 | 3.33/1.65 | 0.092 |
| | LLM attribution (K=10) | 2729/1238 | 2.88/1.74 | 0.063 |
| — | Grad L2 Binary | 2000/2000 | 5.02/0.09 | 0.194 |

## 7 CONCLUSION

In this work, we present the first comprehensive study linking input-based explanations and fairness in hate speech detection. Our experiments show that (1) input-based explanations can effectively identify biased predictions, (2) they are not reliable for selecting fair models, and (3) they can serve as effective supervision signals during training, mitigating bias while preserving a strong balance between fairness and task performance. We further provide practical recommendations on which explanation methods are best suited for bias detection and bias mitigation. Finally, our analyses demonstrate that explanation-based bias detection remains effective in explanation-debiased models, and they outperforms LLM-as-a-judge in identifying biased predictions[5].

---

[5]Limitations and future directions are discussed in Appendix M.

ETHICS STATEMENT

This work investigates explainability and fairness in hate speech detection. Despite the diverse experimental setups explored and the additional generalization tests in Appendix H, the findings are still constrained by the specific configurations considered here. As such, the results may not fully generalize across demographic groups, domains, or tasks, and they may remain vulnerable to adversarial manipulation. We further caution that explanation methods and debiasing techniques cannot fully eliminate residual harms, and that LLM-generated bias judgments are unreliable for bias detection. We hope that our study will contribute to the development of NLP systems that are more transparent, reliable, and fair.

REPRODUCIBILITY STATEMENT

We include full implementation details in the main text and appendix, covering data pre-processing details, model architectures, training procedures, and hyperparameters. Our code and scripts for data pre-processing and experiments are available at https://github.com/Ewanwong/fairness_x_explainability.

ACKNOWLEDGEMENTS

This work was funded by the DFG project GRK 2853 "Neuroexplicit Models of Language, Vision, and Action" (project number 471607914).

We gratefully acknowledge the computing time made available to them on the high-performance computer at the NHR Center of TU Dresden. This center is jointly supported by the Federal Ministry of Research, Technology and Space of Germany and the state governments participating in the NHR (www.nhr-verein.de/unsere-partner).

Mayank Jobanputra is supported by Deutsche Forschungsgemeinschaft (DFG, German Research Foundation) Project ID 389792660-TRR 248 (Foundations of Perspicuous Software Systems).

We also sincerely thank the reviewers and area chairs for their exceptionally detailed and constructive feedback.

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

## A  FAIRNESS AND EXPLAINABILITY IN HATE SPEECH DETECTION

To better motivate our focus on fairness and explainability in the hate speech detection task, we provide additional background in this section. We begin by clarifying our definitions of hate speech and social bias, then review relevant work on fairness and explainability in hate speech detection. Finally, we explain why our study specifically focuses on input-based explanations.

**Hate Speech**  We follow (Fortuna & Nunes, 2018) in defining hate speech as a specific form of abusive or toxic language that targets and attacks protected or identifiable social groups. Under this view, hate speech is a subset of abusive language. This definition is consistent with widely adopted formulations in prior work on hate speech detection (e.g., Nobata et al., 2016; Davidson et al., 2017). Because our study focuses on fairness and, in particular, analyzes model behavior on examples involving specific social groups (race, gender, religion), this standard definition of hate speech aligns well with the scope and goals of our work. We still use the toxic vs. non-toxic labels following the terminology used in the Civil Comments (Borkan et al., 2019) and Jigsaw (cjadams et al., 2019) datasets. Although these datasets include multiple subtypes of abusive content, they group them under the broader notion of toxicity.

**Social Bias**   Following the conceptualization of (Blodgett et al., 2020), we define social bias as the presence of stereotypical associations between social groups and certain attributes, as well as disparities in how these groups are treated as a result. Such biases can lead to both representational harms (e.g., demeaning or misrepresenting targeted groups) and allocational harms (e.g., unfair distribution of opportunities or resources). Given the potential for NLP systems to reproduce or amplify these harms, and their growing influence in everyday life, it is essential to detect and mitigate social bias in these models.

**Social Bias in Hate Speech Detection**   Social bias has been widely documented in hate speech detection systems. Dixon et al. (2018) showed that training data often contain uneven distributions of identity terms and stereotypical associations, which in turn propagates bias into downstream models. Subsequent studies revealed multiple dimensions of such disparities: Sap et al. (2019) demonstrated systematic dialectal prejudice against African-American English (AAE), while Park et al. (2018) reported significant performance gaps across gendered identities. Garg et al. (2019) further found that models frequently assign different toxicity labels to otherwise identical content when only the referenced social group is varied. Sahoo et al. (2022) expanded the scope of this line of study by curating the ToxicBias dataset and examining bias across a broader set of social categories. More recently, Roy et al. (2023) found that LLMs exhibit similar bias in hate speech detection. Together, these studies underscore the persistence and multifaceted nature of social bias in hate speech detection.

To address such biases, a rich line of work has proposed mitigation techniques at different stages of the modeling pipeline. Pre-processing methods include debiasing word embeddings to reduce spurious associations between identity terms and toxicity (Park et al., 2018), re-sampling or re-weighting examples to obtain more balanced label distributions across identity groups (Dixon et al., 2018), and counterfactual data augmentation (Park et al., 2018; Garg et al., 2019). In-processing approaches mostly modify the training objective, for instance by adding fairness-aware regularizers that penalize correlations between identity terms and toxic predictions (Garg et al., 2019; Davani et al., 2020; Attanasio et al., 2022; Schäfer et al., 2024). Post-processing methods adjust model outputs without retraining (Wang & Demberg, 2024b): threshold adjustment per group has been used to trade off subgroup false positive and false negative rates and reduce disparities (Dixon et al., 2018), while Mamta et al. (2024) identify neurons associated with biased behavior and prune or edit them to improve fairness.

Despite substantial progress on identifying and mitigating social bias in hate speech detection, relatively little work has systematically explored whether model explanations can be leveraged to detect or reduce such biases.

**Explainability in Hate Speech Detection**   In parallel, there is a growing line of work on input-based explanations for hate speech detection. HateXplain (Mathew et al., 2021) introduces a benchmark with human-annotated rationales and shows that models trained with rationale supervision improve both interpretability and reduce unintended bias towards target communities. Building on this, Kim et al. (2022) and Saha et al. (2023) train models to jointly predict human rationales and toxicity labels, leading to more robust and explainable hate speech detection systems. More recent work further leverages LLM-generated rationales to supervise hate speech classifiers, achieving improved performance and interpretability (Nirmal et al., 2024).

However, while existing efforts focus primarily on improving hate speech detection performance, relatively little work examines whether and how input-based explanations can be systematically leveraged to improve fairness in hate speech detection models. Since fairness and explainability have both been extensively studied in this task, hate speech detection serves as an ideal setting for a thorough empirical examination of how these two dimensions interact in NLP models.

**Input-Based Explanations**   We focus on input-based explanations because they offer the most direct view into which parts of the input influence a model's prediction (Wang & Yin, 2021), and they have long been regarded as central tools for fairness auditing in ML (Balkir et al., 2022; Deck et al., 2024). Their methodological diversity also makes them an ideal testbed for our study, enabling a comprehensive examination of whether and how explanations can improve fairness (Lyu et al., 2024). In addition, both automated and human-centric metrics for evaluating explanation properties (e.g., faithfulness, interpretability) are well established (DeYoung et al., 2020; Jacovi &

Goldberg, 2020; Lage et al., 2019). This allows us to analyze how these properties relate to an explanation method's (in)effectiveness in fairness-related tasks. Finally, input-based explanations are often mandated by laws, such as the EU Artificial Intelligence Act, making it practically important to understand how their use interacts with fairness considerations.

## B DETAILS ON EXPERIMENTAL SETUPS

Table 3: Splits for the Civil Comments and Jigsaw datasets used in this work. The instances are sampled randomly from the original dataset.

| Split | Civil Comments | | | Jigsaw | | |
|---|---|---|---|---|---|---|
| | race | gender | religion | race | gender | religion |
| Test | 2,000 | 2,000 | 1,000 | 400 | 800 | 200 |
| Training | 8,000 | 8,000 | 6,300 | 8,000 | 8,000 | 6,300 |

**Datasets and Models**  Table 3 shows the dataset statistics for the Civil Comments[6] and Jigsaw[7] datasets, respectively. When constructing subsets for each social group, we include only examples that contain words from the corresponding group's vocabulary and exclude those containing words from other groups. Additionally, we remove examples that include insulting references to social groups.

For encoder-only models, we either train separate models for each bias type or a single model on the combined set of all bias types. The latter are marked as (all) in the results. Models are trained for 5 epochs with a batch size of 8, a learning rate of $2 \times 10^{-5}$, and 10% warm-up steps. We implement debiasing methods of encoder-only models ourselves. Debiased models are trained on the same number of examples for comparability, therefore we do not sample more examples from external datasets for causal debias. All experiments are carried out using a maximum of 4x NVIDIA H100 GPUs.

We use the base version of BERT and RoBERTa models for our encoder-only experiments, and for decoder-only experiments, we adopt the instruct versions of the models from HuggingFace.

**Prompts**  We employ the following prompt for zero-shot and few-shot prompting with decoder-only models.

```
Task: Determine whether the following sentence contains any toxic
content. Toxicity includes insults, threats, hate speech, or any
language that is likely to offend or harm others.

([EXAMPLES])
(Now, please analyze the following sentence and provide your
answer. Answer only Yes or No.)

Sentence: [TEST EXAMPLE]

Question: Does this sentence contain toxicity?

Answer (Yes or No):
```

We use one toxic and one non-toxic example for each social group associated with a bias type to construct the corresponding few-shot prompt. For fairness imagination, we prepend the task instruction with:

```
Imagine a world with no bias regarding [BIAS TYPE].
```

---

[6]https://huggingface.co/datasets/lighteval/civil_comments_helm
[7]https://huggingface.co/datasets/google/jigsaw_unintended_bias

For fairness instruction, we add:

```
In the following, do not rely on stereotypes or biases regarding
[BIAS TYPE] to make predictions.
```

**Input-Based Explanation Methods**   We implement Attention, Attention Rollout, Attention Flow, and Progressive Inference ourselves, and use the official codebase from the DecompX paper for its implementation. For other explanation methods, we generate them using the Captum API[8] Single-pass Progressive Inference is used for simplicity.

**Explanation-Based Bias Mitigation**   We reimplement all explanation methods used in RQ3 to enable gradient tracking during training and ensure consistency with the explanations provided by the Captum API. For Occlusion, we simplify computation by penalizing the probability difference between the original input and the input with all sensitive tokens masked.

**LLM decision and LLM attribution**   To elicit bias judgments from LLMs, we append the following instructions to their predictions.

**LLM decision:**

```
Do not re-answer the task. Instead, reflect only on your previous
answer.

Question: Was your previous answer influenced by stereotypes or
biases related to [BIAS TYPE]?

Answer (Yes or No):
```

**LLM attribution:**

```
Now select exactly [NUM_TOKENS] words from the input sentence that
most influenced your decision. List the words separated by commas.
Please begin directly with the selected word.

Selected words:
```

## C   EXPLANATION EFFICIENCY

Table 4 reports the time and GPU memory costs for each explanation method. Most post-hoc explanation methods are lightweight when applied to BERT, whereas IntGrad, Occlusion, and KernelSHAP require substantially more time and computational resources when generating explanations for LLMs.

## D   MODEL PERFORMANCE AND FAIRNESS RESULTS

Tables 5 and 6 show the task performance and fairness scores for the default/zero-shot and debiased models on the Civil Comments and Jigsaw datasets respectively. To better identify the differences between different debiasing methods, we conduct an analysis based on how often a debiasing method successfully reduces the average individual unfairness ($Avg_{iu}$) and maintains the task performance (Accuracy) of the default/zero-shot model.

**Encoder-only models**   Analyzing the results with respect to the dataset, we find that the models are able to better preserve their original accuracy on the Civil Comments dataset (48.61% of the cases) compared to the Jigsaw dataset (40.28% of the cases). In contrast, mitigating bias seems substantially easier on the Jigsaw dataset (in 63.88% of the cases) than on the Civil Comments (only 50% of the cases). On closer inspection, we find that this skew comes from religion bias

---

[8]https://captum.ai/api/

Table 4: Computational costs per example for generating explanations across 200 instances on BERT and Qwen3-4B. Results are computed on the race subset of the Civil Comments dataset using a batch size of 1 and are averaged over three runs. All methods are run on a single 80-GB H100 GPU, except Integrated Gradients, which uses two H100 GPUs with gradient checkpointing to reduce memory usage. Because explanation methods within the same family incur similar computational costs, we report each family only once.

| | **BERT** | | | **Qwen3-4B** | |
| Method | Time (s/example) | Memory (GB) | Method | Time (s/example) | Memory (GB) |
|---|---|---|---|---|---|
| Attention | 0.027 | 0.529 | Attention | 0.112 | 16.598 |
| Grad | 0.026 | 0.603 | Grad | 0.237 | 19.631 |
| IxG | 0.025 | 0.603 | IxG | 0.236 | 19.631 |
| IntGrad | 0.064 | 7.010 | IntGrad | 1.784 | 101.694 |
| DeepLift | 0.027 | 0.748 | DeepLift | 0.323 | 23.530 |
| Occlusion | 0.330 | 0.508 | Occlusion | 4.204 | 15.639 |
| KernelSHAP | 0.138 | 0.508 | KernelSHAP | 1.374 | 20.013 |
| DecompX | 0.074 | 7.222 | ProgInfer | 0.068 | 15.810 |

Table 5: Task performance and fairness of default and debiased models on the Civil Comments dataset. Results are provided for race/gender/religion biases. Green (red) indicates the results are better (worse) than the default/zero-shot models. *All* indicates the model is trained on data containing all bias types.

| Model | Method | Accuracy (↑) | $\text{Disp}_{acc}$(↓) | $\text{Disp}_{fpr}$(↓) | $\text{Disp}_{fnr}$(↓) | $\text{Avg}_{iu}$(↓) |
|---|---|---|---|---|---|---|
| BERT | Default | 78.38/88.05/85.93 | 2.05/3.30/18.07 | 0.50/0.03/5.77 | 10.04/11.98/30.90 | 3.17/0.66/1.27 |
| | Group balance | 79.25/87.25/86.83 | 3.10/2.80/13.53 | 0.25/1.73/11.53 | 10.46/5.38/30.31 | 3.79/0.42/2.01 |
| | Group-class balance | 78.00/87.02/85.77 | 1.80/2.75/14.73 | 2.42/0.99/3.09 | 10.63/7.26/33.14 | 4.43/0.98/0.71 |
| | CDA | 76.83/86.70/84.83 | 2.35/3.60/14.13 | 5.88/2.00/5.67 | 18.45/7.57/24.12 | 0.50/0.50/0.90 |
| | Dropout | 78.53/88.20/85.03 | 2.25/2.10/15.67 | 0.78/1.46/5.93 | 10.82/3.50/27.16 | 3.43/0.52/1.51 |
| | Attention entropy | 79.15/87.67/84.93 | 2.60/2.05/15.07 | 0.99/0.10/4.99 | 11.71/7.11/26.52 | 2.95/0.67/1.58 |
| | Causal debias | 78.80/86.17/86.40 | 0.00/2.65/16.40 | 3.90/0.46/8.82 | 7.98/10.67/30.46 | 3.83/0.48/2.10 |
| BERT (all) | Default | 78.30/88.20/87.43 | 2.00/3.20/13.47 | 0.02/1.11/6.24 | 8.44/8.58/23.53 | 3.99/0.96/1.76 |
| | Group balance | 79.05/88.85/87.47 | 3.50/2.80/13.67 | 1.72/0.31/6.92 | 8.83/11.08/23.91 | 4.13/1.17/2.15 |
| | Group-class balance | 78.17/88.25/86.90 | 1.95/1.70/14.60 | 1.35/0.51/8.52 | 9.33/4.66/33.13 | 4.83/0.93/1.37 |
| | CDA | 78.08/87.70/86.83 | 2.65/2.70/14.33 | 6.38/1.05/4.70 | 20.35/6.92/30.23 | 0.60/0.46/0.71 |
| | Dropout | 78.08/87.60/87.67 | 2.45/3.10/13.47 | 0.30/1.05/5.53 | 9.99/8.39/33.12 | 3.60/0.89/1.59 |
| | Attention entropy | 78.35/87.90/87.77 | 2.10/2.30/11.67 | 1.28/0.10/6.55 | 5.92/8.01/36.15 | 4.98/0.96/2.10 |
| | Causal debias | 79.40/88.75/87.70 | 2.20/2.60/12.60 | 2.51/0.70/6.70 | 13.13/7.44/31.28 | 3.54/0.80/2.12 |
| RoBERTa | Default | 78.50/88.33/85.23 | 2.80/2.78/17.07 | 2.84/1.66/6.59 | 15.46/2.78/31.64 | 2.56/0.60/1.55 |
| | Group balance | 78.25/88.50/87.03 | 2.00/2.20/16.93 | 2.10/1.27/11.36 | 9.85/4.57/29.48 | 3.95/0.68/1.19 |
| | Group-class balance | 78.57/84.50/83.60 | 1.65/2.30/18.80 | 3.31/0.76/3.89 | 12.91/5.82/38.88 | 3.28/0.42/0.87 |
| | CDA | 76.75/87.58/85.20 | 1.60/1.75/14.20 | 6.37/0.31/4.10 | 15.91/5.41/35.70 | 0.82/0.42/1.19 |
| | Dropout | 78.33/88.92/86.73 | 2.15/1.55/14.53 | 2.42/0.58/8.86 | 11.11/3.96/27.05 | 4.08/0.56/2.10 |
| | Attention entropy | 78.33/88.42/86.67 | 1.75/1.75/15.73 | 2.89/0.23/9.23 | 10.91/5.60/24.68 | 3.82/0.69/1.75 |
| | Causal debias | 78.83/87.52/86.00 | 2.65/2.45/15.60 | 1.48/0.85/10.56 | 11.34/6.51/30.14 | 4.06/0.56/1.34 |
| RoBERTa (all) | Default | 78.88/88.70/87.90 | 2.95/2.40/13.80 | 2.24/0.58/9.50 | 13.55/7.19/33.47 | 4.14/0.95/2.35 |
| | Group balance | 79.30/88.65/87.93 | 2.90/2.00/14.73 | 1.27/0.17/12.30 | 11.03/7.74/31.69 | 5.02/1.06/2.80 |
| | Group-class balance | 79.40/89.15/87.93 | 1.70/1.10/12.73 | 4.43/0.24/5.08 | 13.65/3.24/25.90 | 4.17/0.75/1.58 |
| | CDA | 77.75/88.25/86.90 | 2.50/2.00/13.80 | 5.93/1.25/6.33 | 18.80/3.71/22.62 | 1.13/0.55/1.18 |
| | Dropout | 78.88/88.40/87.70 | 2.75/3.00/14.80 | 1.80/1.33/6.66 | 12.46/7.34/33.39 | 4.26/0.99/2.13 |
| | Attention entropy | 78.80/88.72/87.83 | 2.10/2.15/13.53 | 2.64/1.33/7.55 | 11.31/4.18/28.68 | 4.46/1.09/2.57 |
| | Causal debias | 79.27/89.78/87.80 | 3.35/1.25/15.00 | 3.24/0.51/11.86 | 16.00/3.05/37.57 | 3.56/0.74/2.70 |
| Llama3.2-3B-Instruct | Zero-shot | 63.78/74.62/71.27 | 1.45/2.35/24.67 | 11.03/3.52/36.81 | 10.54/1.03/2.95 | 2.13/2.94/3.83 |
| | Few-shot | 67.80/79.80/80.10 | 1.60/1.70/18.20 | 2.49/0.08/6.73 | 6.73/6.05/10.77 | 1.39/2.05/1.90 |
| | Fairness imagination | 64.95/75.92/73.37 | 0.80/0.85/21.87 | 8.70/3.61/32.54 | 9.44/6.79/5.98 | 2.65/3.58/3.50 |
| | Fairness instruction | 65.90/76.95/78.07 | 2.60/1.70/21.53 | 1.89/0.39/7.00 | 3.79/6.35/4.24 | 1.35/1.13/1.71 |
| Qwen3-4B | Zero-shot | 69.55/79.75/77.50 | 0.60/0.00/17.40 | 7.13/1.40/21.07 | 13.25/3.71/5.17 | 2.55/2.41/3.32 |
| | Few-shot | 70.15/80.73/79.53 | 1.80/0.65/18.93 | 10.02/2.50/19.31 | 11.89/9.15/5.57 | 3.18/3.34/3.76 |
| | Fairness imagination | 71.23/80.40/80.83 | 0.85/1.00/18.27 | 4.03/2.11/10.51 | 11.62/9.21/4.28 | 2.98/3.16/2.20 |
| | Fairness instruction | 70.40/79.77/80.47 | 0.60/1.35/19.33 | 4.30/0.39/4.67 | 11.11/5.24/5.08 | 2.02/1.83/1.71 |
| Qwen3-8B | Zero-shot | 59.27/69.23/66.30 | 1.25/0.15/26.80 | 8.18/0.07/42.05 | 4.65/0.80/3.02 | 3.27/3.40/4.74 |
| | Few-shot | 66.97/77.30/77.47 | 0.05/0.00/23.27 | 6.14/2.73/29.51 | 7.95/7.64/2.34 | 4.23/4.58/5.96 |
| | Fairness imagination | 62.10/72.92/69.97 | 1.60/0.55/21.87 | 7.80/2.62/32.27 | 4.28/5.42/9.43 | 2.54/2.08/2.58 |
| | Fairness instruction | 66.50/75.15/73.90 | 0.90/0.10/21.20 | 8.03/1.76/28.60 | 8.79/4.59/7.45 | 2.45/2.95/3.15 |

in the Jigsaw dataset which is improved in 95.83% of the cases after debiasing, followed by race bias (50%) and gender bias (45.83%). In the Civil Comments dataset, we find that gender bias is mitigated best (improvement in 62.5% of the cases), followed by religion bias (54.17%) and race bias (33.33%).

Table 6: Task performance and fairness results of default and debiased models on the Jigsaw dataset. Results are provided for race/gender/religion biases. Green (red) indicates the results are better (worse) than the default/zero-shot models. *All* indicates the model is trained on data containing all bias types.

| Model | Method | Accuracy (↑) | $\mathrm{Disp_{acc}}(\downarrow)$ | $\mathrm{Disp_{fpr}}(\downarrow)$ | $\mathrm{Disp_{fnr}}(\downarrow)$ | $\mathrm{Avg_{iu}}(\downarrow)$ |
|---|---|---|---|---|---|---|
| BERT | Default | 85.50/93.00/90.50 | 0.50/2.25/6.00 | 0.64/2.34/5.22 | 0.70/3.28/21.54 | 2.02/0.36/1.33 |
| | Group balance | 84.88/92.75/89.67 | 2.75/1.00/10.67 | 1.28/0.82/3.90 | 7.77/4.56/38.29 | 1.90/0.36/0.67 |
| | Group-class balance | 84.38/92.81/90.83 | 0.25/0.62/6.33 | 1.58/0.15/1.98 | 8.03/9.64/43.57 | 0.97/0.65/0.34 |
| | CDA | 85.25/91.81/90.50 | 4.00/3.63/10.00 | 4.12/3.44/5.10 | 2.97/7.38/37.39 | 0.39/0.28/0.45 |
| | Dropout | 85.62/92.69/89.83 | 1.25/3.37/9.67 | 0.31/3.03/5.46 | 6.51/8.41/27.37 | 2.75/0.36/1.00 |
| | Attention entropy | 85.00/92.06/89.83 | 0.00/3.12/9.33 | 0.62/3.03/4.29 | 1.72/6.00/28.06 | 2.93/0.50/0.98 |
| | Causal debias | 85.50/93.38/89.83 | 4.00/0.75/7.33 | 1.28/0.28/3.55 | 13.73/12.77/17.12 | 3.16/0.43/1.10 |
| BERT (all) | Default | 85.62/93.19/90.33 | 1.25/1.12/9.33 | 1.59/1.51/4.65 | 12.69/0.36/21.76 | 1.30/0.33/1.18 |
| | Group balance | 83.38/93.19/90.17 | 1.75/1.12/9.67 | 1.56/1.10/4.66 | 3.10/3.23/26.79 | 2.81/0.40/0.76 |
| | Group-class balance | 84.88/92.94/90.00 | 1.25/0.87/10.00 | 1.27/0.41/2.09 | 0.37/7.49/58.07 | 1.29/0.28/0.47 |
| | CDA | 85.62/92.19/90.00 | 3.25/1.88/7.00 | 2.86/1.78/4.02 | 4.17/4.41/38.24 | 0.69/0.29/0.46 |
| | Dropout | 86.50/93.44/91.00 | 3.00/1.38/7.00 | 1.26/1.10/5.60 | 10.24/6.10/13.16 | 1.91/0.33/1.27 |
| | Attention entropy | 85.25/93.75/91.50 | 0.50/2.75/8.00 | 0.65/2.62/5.19 | 0.57/5.54/34.85 | 2.57/0.41/1.07 |
| | Causal debias | 84.50/93.44/90.50 | 1.00/1.38/9.00 | 2.22/1.38/4.27 | 4.35/3.38/24.10 | 1.40/0.40/1.00 |
| RoBERTa | Default | 84.50/93.00/90.33 | 1.00/3.75/10.33 | 2.87/3.44/1.82 | 6.54/8.31/47.47 | 2.55/0.30/0.89 |
| | Group balance | 85.50/92.31/89.83 | 2.50/0.62/11.33 | 0.94/0.27/1.55 | 9.11/6.41/38.00 | 2.44/0.26/0.46 |
| | Group-class balance | 85.00/92.50/90.67 | 1.00/1.50/5.33 | 1.59/0.26/2.01 | 11.53/14.87/24.59 | 1.55/0.53/0.62 |
| | CDA | 85.12/93.19/89.33 | 0.75/1.88/8.67 | 4.12/1.10/3.90 | 12.64/11.13/25.89 | 0.36/0.23/0.40 |
| | Dropout | 83.88/93.69/90.17 | 1.75/0.88/7.67 | 1.29/0.82/2.97 | 3.10/3.28/26.86 | 2.71/0.23/0.87 |
| | Attention entropy | 85.00/93.50/90.33 | 0.50/1.75/6.67 | 2.23/2.06/1.01 | 6.55/0.62/22.78 | 2.39/0.24/0.81 |
| | Causal debias | 86.25/92.19/89.50 | 2.00/3.37/10.00 | 2.23/2.33/1.84 | 0.60/14.77/43.47 | 2.09/0.39/0.66 |
| RoBERTa (all) | Default | 85.50/93.75/91.50 | 0.50/1.75/7.00 | 0.01/1.51/5.56 | 3.06/5.74/31.14 | 2.52/0.35/1.55 |
| | Group balance | 85.38/93.62/91.67 | 1.75/3.25/9.33 | 0.01/2.47/4.12 | 9.01/11.90/40.29 | 2.76/0.30/0.96 |
| | Group-class balance | 86.38/92.56/90.17 | 2.25/1.88/10.67 | 0.62/1.37/2.58 | 9.05/8.62/64.35 | 4.75/0.23/0.34 |
| | CDA | 85.25/92.56/90.67 | 1.00/0.62/7.67 | 1.59/0.13/1.80 | 11.53/7.49/31.28 | 0.52/0.23/0.74 |
| | Dropout | 86.00/93.00/90.17 | 2.50/1.75/4.67 | 1.27/1.51/4.19 | 17.51/6.21/28.72 | 1.02/0.33/0.79 |
| | Attention entropy | 86.75/93.50/91.50 | 0.50/2.50/7.00 | 0.96/2.06/3.16 | 6.54/8.05/24.59 | 3.40/0.38/1.19 |
| | Causal debias | 85.38/93.25/91.00 | 0.25/3.50/10.00 | 0.01/2.62/5.41 | 1.88/13.69/34.14 | 2.55/0.40/0.80 |
| Llama3.2-3B-Instruct | Zero-shot | 54.00/70.50/65.17 | 8.50/1.00/25.67 | 10.91/1.53/31.87 | 0.20/4.56/8.33 | 2.39/3.00/4.28 |
| | Few-shot | 73.12/88.62/86.83 | 7.25/0.50/7.67 | 13.01/0.16/4.83 | 15.05/9.08/23.17 | 1.63/1.68/2.00 |
| | Fairness imagination | 57.75/73.56/66.83 | 5.00/1.62/26.33 | 6.47/1.63/30.03 | 0.26/1.74/17.48 | 2.86/3.73/3.92 |
| | Fairness imagination | 57.75/73.56/66.83 | 5.00/1.62/26.33 | 6.47/1.63/30.03 | 0.26/1.74/17.48 | 2.86/3.73/3.92 |
| | Fairness instruction | 77.00/89.00/87.17 | 2.00/0.75/10.67 | 2.87/0.84/3.97 | 2.19/3.33/36.36 | 1.39/0.97/1.87 |
| Qwen3-4B | Zero-shot | 66.75/77.25/77.33 | 3.50/3.75/16.33 | 4.21/3.78/17.40 | 0.80/4.05/5.89 | 3.05/2.31/3.67 |
| | Few-shot | 57.88/68.06/77.83 | 8.75/2.12/9.33 | 11.52/1.80/10.86 | 1.49/5.79/9.45 | 3.60/4.31/4.18 |
| | Fairness imagination | 73.75/82.88/86.33 | 3.00/1.00/10.33 | 5.12/0.89/5.79 | 5.39/0.82/24.13 | 3.14/2.97/2.51 |
| | Fairness instruction | 78.00/89.50/89.33 | 3.00/0.50/9.33 | 4.14/0.26/3.13 | 2.04/5.23/26.74 | 1.95/1.43/1.61 |
| Qwen3-8B | Zero-shot | 48.12/59.50/56.50 | 7.25/0.00/13.00 | 9.68/0.37/18.59 | 1.31/4.92/6.30 | 3.31/3.47/5.52 |
| | Few-shot | 53.75/67.19/77.17 | 5.50/1.12/8.67 | 6.18/1.53/10.51 | 3.41/1.95/4.88 | 4.50/5.04/5.99 |
| | Fairness imagination | 51.62/67.50/61.83 | 4.25/0.75/9.67 | 5.24/0.56/11.23 | 1.05/3.49/6.45 | 2.51/2.02/2.55 |
| | Fairness instruction | 60.25/71.19/67.50 | 8.50/1.87/12.00 | 10.57/2.23/14.22 | 0.93/1.90/2.10 | 2.46/3.13/3.60 |

With respect to the debiasing method, we find that CDA performs best in terms of debiasing, as it reduces $\mathrm{Avg_{iu}}$ across all bias types, datasets, and models. The second best performing method is group-class balance which manages to reduce $\mathrm{Avg_{iu}}$ in 58.33% of the cases on the Civil Comments dataset and in 75% cases on the Jigsaw dataset. For the other methods, the results are mixed as we again observe dataset-specific differences. For example, we find that Attention entropy performs well on the Jigsaw dataset (50%) but performs worst on the Civil Comments dataset (16.67%). These differences become even more pronounced when looking at different bias types. For instance, causal debiasing improves $\mathrm{Avg_{iu}}$ for religion bias across all models on the Jigsaw dataset but at the same time, does not improve a single model in terms of $\mathrm{Avg_{iu}}$ for gender bias in the same dataset. Interestingly, we find an inverse trend on the Civil Comments dataset; i.e., causal debiasing succeeds on all models for gender bias, but only for one model for religion bias. These findings highlight the importance of considering a diverse set of datasets for evaluating debiasing methods, as results on a single dataset can be misleading.

**Decoder-only models** We find that the debiasing methods (fairness imagination and fairness instruction) for the decoder-only models consistently improve the task performance across all bias types and datasets. Contrary to this, we see increases in average individual unfairness of the fairness imagination approach for race and gender bias on Llama3.2-3B-Instruct and Qwen3-4B across both datasets. Only for religion, fairness imagination leads to a consistent decrease of the individual unfairness across models. For fairness instruction, we observe a consistent improvement across all three bias types and both datasets, showing the clear superiority of the approach. The consistency of

the results is especially surprising when considering that both decoder-only models are instruction-tuned and aligned with human values, and that Chen et al. (2025) identify a bias amplification effect from instruction tuning. We conclude that fairness instruction is a good baseline to evaluate other debiasing methods for decoder-only models.

## E  BIAS DETECTION RESULTS

**Fairness correlation**   We present the full fairness correlation results of encoder- and decoder-only models with different debiasing methods on Civil Comments and Jigsaw in Figures 7, 8, 9, 10. Consistent with findings presented in the main text, Occlusion- and L2-based explanation methods achieve strong fairness correlations across different setups.

Comparing different debiasing methods, we find that low correlation scores primarily occur when individual unfairness is less pronounced, such as in CDA models. In these cases, the models themselves produce fewer biased predictions, making the detection of bias through explanations less critical. The lower correlations therefore do not substantially undermine the role of explanations in bias identification.

## F  MODEL SELECTION RESULTS

**Explanation-Based Metrics and Fair Model Selection Results**   We evaluate several explanation-based metrics for selecting fair models with respect to different fairness criteria:

- **Average absolute sensitive token reliance:** used to predict average individual unfairness, under the assumption that higher reliance on sensitive tokens implies greater sensitivity to group substitutions.

- **Group differences in average absolute sensitive token reliance:** used to predict disparities in accuracy, assuming that stronger reliance on sensitive features increases the risk of incorrect predictions.

- **Group differences in average absolute sensitive token reliance for positive/negative predictions:** used to predict disparities in false positive and false negative rates, respectively.

Among these, only average absolute sensitive token reliance exhibits rank correlations above random chance with its target fairness metric (individual unfairness). The correlations for other metrics remain at chance level. Figures 11, 12, 13, 14 demonstrate that no explanation methods can consistently match baseline rank correlation results.

Figures 15, 16, 17, 18 further reveal that explanation methods are not able to robustly select the fairest models. These findings underline the unreliability of explanation-based model selection.

## G  BIAS MITIGATION RESULTS

The complete bias mitigation results are presented in Figures 19, 20, 21, 22. The findings are in line with conclusions from the main paper, that explanation-based debiasing can effectively reduce model biases across different fairness metrics, bias types, models, and datasets. In addition, the accuracy-fairness harmonic mean results shown in Figures 23, 24, 25, 26 demonstrate that explanation-based debiasing achieves comparable or superior balance between fairness and task performance than default models and traditional debiasing approaches.

We additionally report the results of Integrated Gradients for bias mitigation in Table 7. Similar to other explanation methods, IntGrad-based debiasing achieves substantial bias reduction and maintains a good balance between fairness and task performance in $\text{Disp}_{\text{fnr}}$ and $\text{Avg}_{\text{iu}}$.

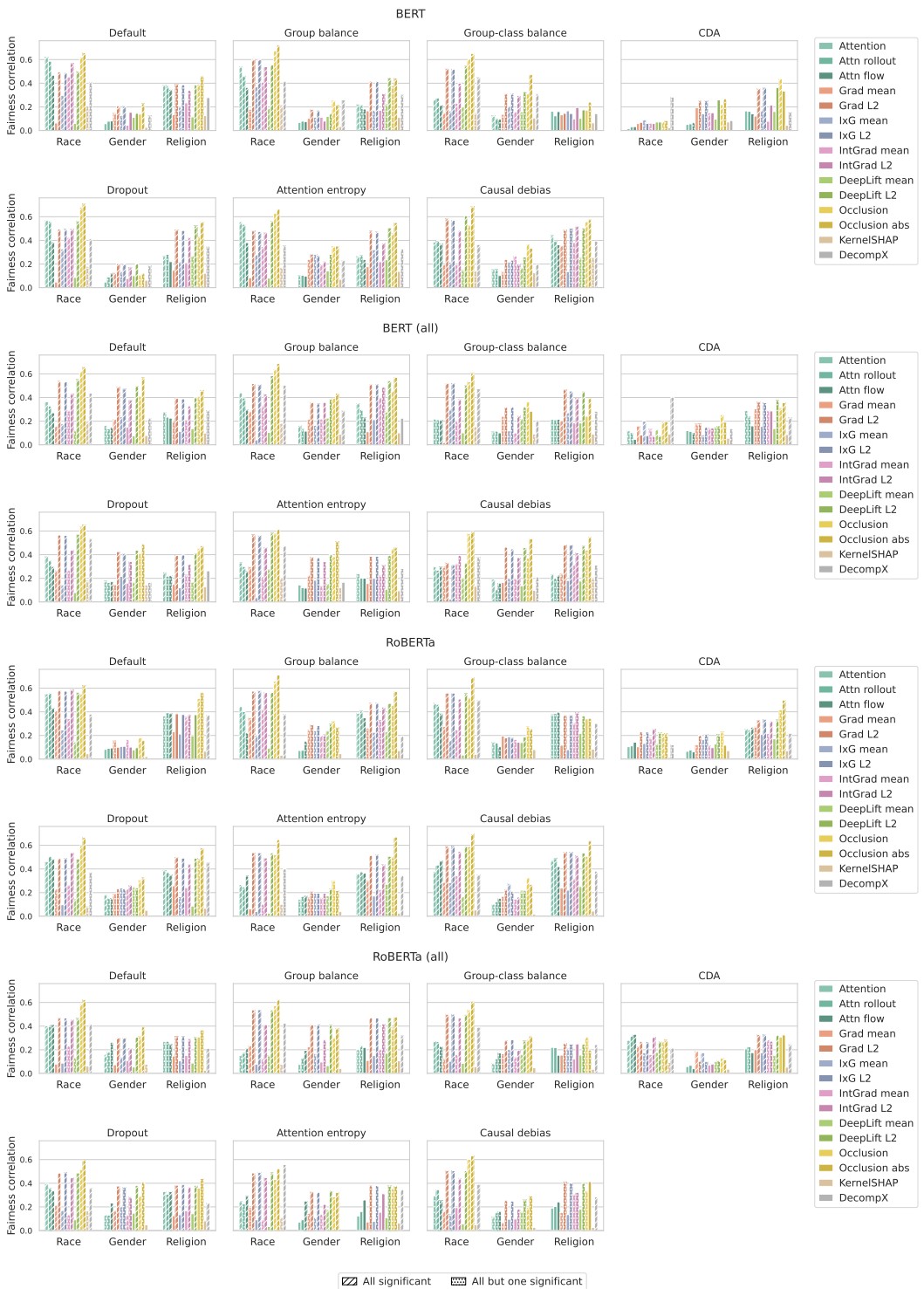

Figure 7: Fairness correlation results on Civil Comments for each explanation method across encoder-only models and bias types. Higher values indicate that the method is more effective and reliable in detecting biased predictions at inference time. *All* indicates the model is trained on data containing all bias types.

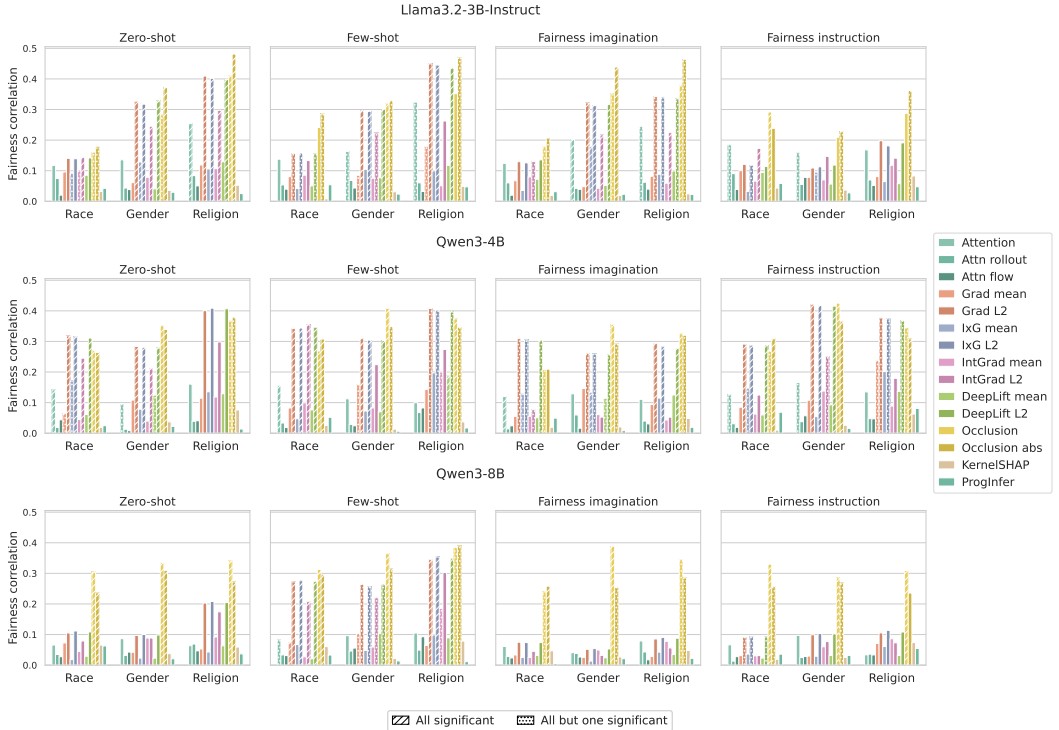

Figure 8: Fairness correlation results on Civil Comments for each explanation method across decoder-only models and bias types. Higher values indicate that the method is more effective and reliable in detecting biased predictions at inference time.

Table 7: Results of mitigating race bias in BERT models using Intgrad explanations on Civil Comments. For consistency with accuracy, fairness results are reported as $100 - \{\text{Disp}_{\text{acc}}, \text{Disp}_{\text{fpr}}, \text{Disp}_{\text{fnr}}, \text{Avg}_{\text{iu}}\}$, so that higher values indicate better debiasing performance. Each column corresponds to models selected by maximizing the fairness-balanced metric with respect to the indicated bias metric. H-Mean indicates harmonic mean between fairness and accuracy. Green (red) indicates the results are better (worse) than the default models.

| Method | Disp$_{\text{acc}}$ | | | Disp$_{\text{fpr}}$ | | | Disp$_{\text{fnr}}$ | | | Avg$_{\text{iu}}$ | | |
|---|---|---|---|---|---|---|---|---|---|---|---|---|
| | Acc | Fairness | H-Mean | Acc | Fairness | H-Mean | Acc | Fairness | H-Mean | Acc | Fairness | H-Mean |
| IntGrad L1 | 78.55 | 96.66 | 86.67 | 78.55 | 96.66 | 86.67 | 77.98 | 97.86 | 86.8 | 78.37 | 97.02 | 86.7 |
| IntGrad L2 | 77.85 | 96.58 | 86.21 | 77.71 | 96.33 | 86.02 | 77.85 | 96.58 | 86.21 | 78.09 | 97.1 | 86.56 |
| Default | 78.97 | 98.37 | 87.61 | 78.97 | 98.96 | 87.84 | 78.97 | 91.15 | 84.62 | 78.97 | 96.36 | 86.8 |

## H  GENERALIZATION OF FINDINGS

To demonstrate the generalizability of our findings, we present results under additional setups that vary in task, model alignment type, and sensitive token vocabulary. We observe similar results across these setups, suggesting that our findings generalize well beyond the main study conditions.

**Task**  We evaluated explanation-based bias detection (RQ1) on an additional task, namely sentiment analysis, using the Twitter Sentiment dataset[9]. Specifically, we selected 1000 gender-related examples (500 referencing males and 500 referencing females) and ran explanation-based bias detection on them. In Figure 27 we report the results on Llama3.2-3B-Instruct and Qwen3-4B models.

In Figure 27, we observe patterns in the sentiment analysis task that are similar to those in our main study: certain explanation methods (e.g., occlusion-based and L2-based approaches) can still

---

[9]https://huggingface.co/datasets/shukdevdatta123/twitter_sentiment_preprocessed

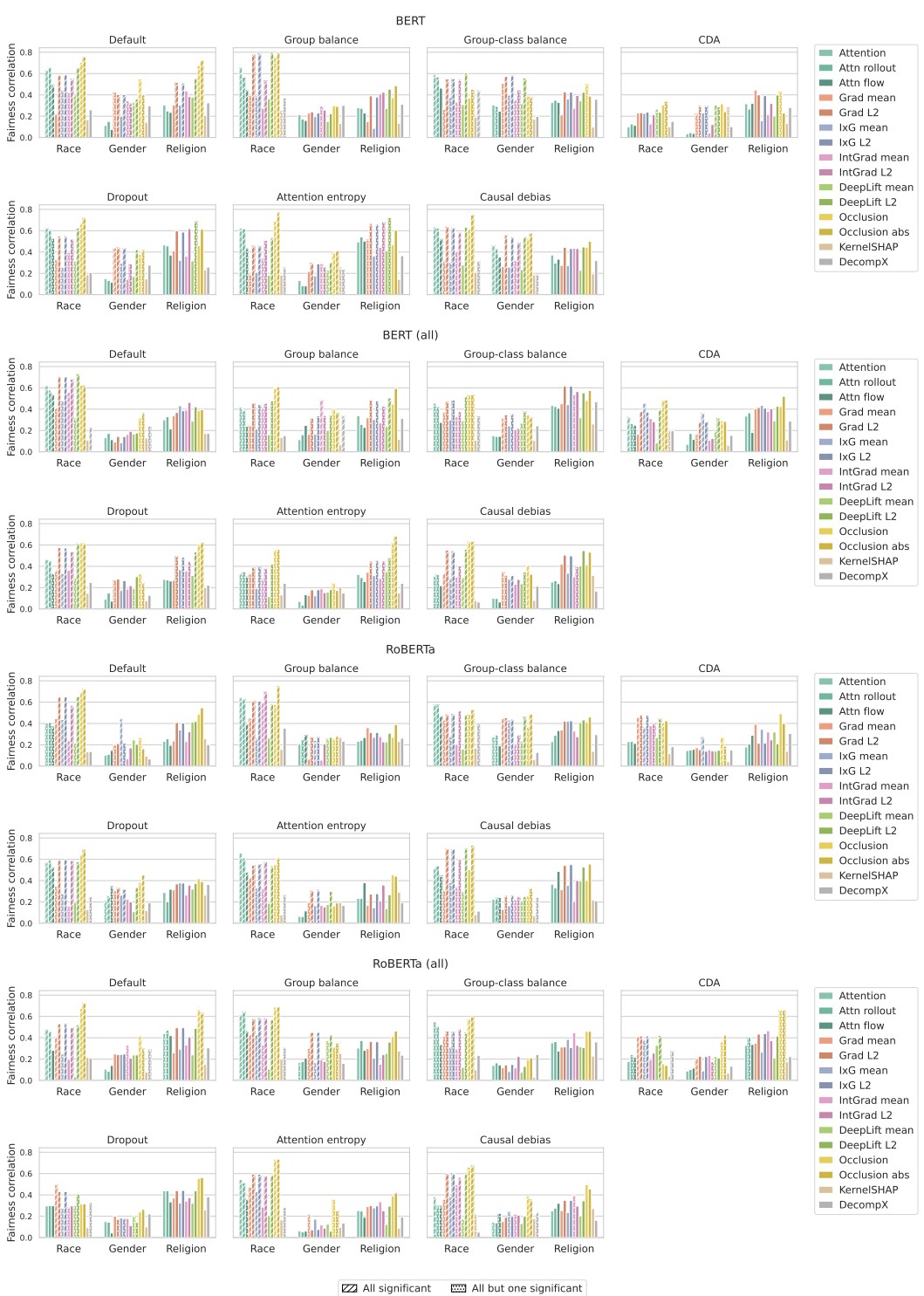

Figure 9: Fairness correlation results on Jigsaw for each explanation method across encoder-only models and bias types. Higher values indicate that the method is more effective and reliable in detecting biased predictions at inference time. *All* indicates the model is trained on data containing all bias types.

achieve high fairness correlation scores. This suggests that our findings could extend beyond the hate speech detection task.

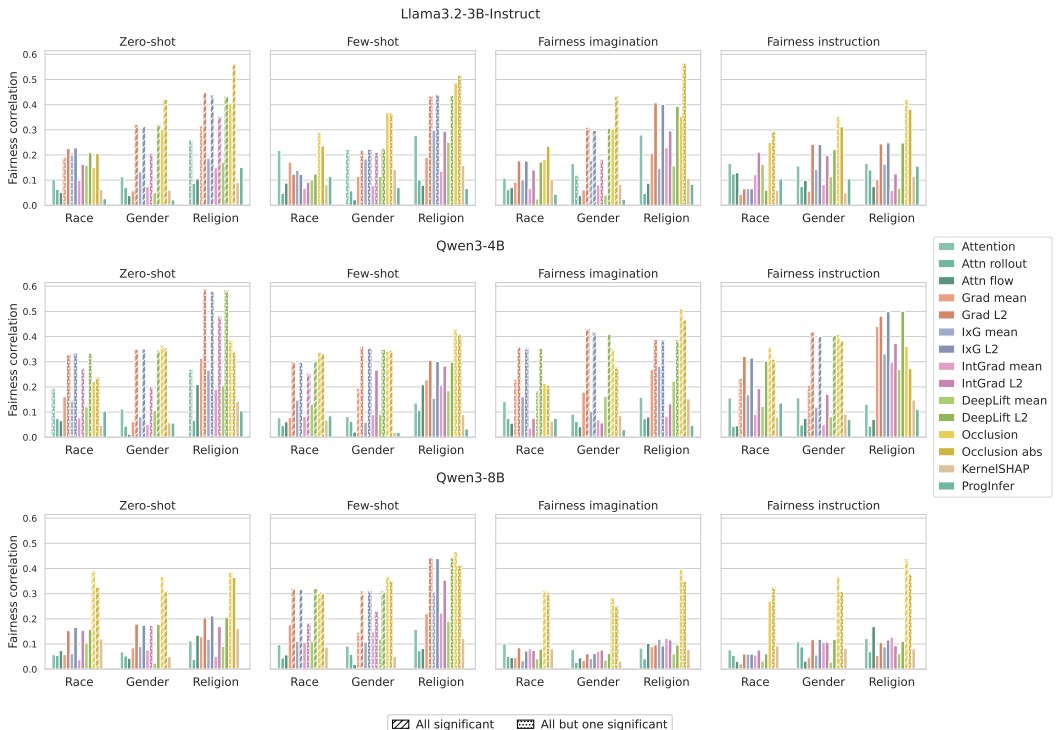

Figure 10: Fairness correlation results on Jigsaw for each explanation method across decoder-only models and bias types. Higher values indicate that the method is more effective and reliable in detecting biased predictions at inference time.

**Model Alignment Type**   We extended our experiments to additional LLMs with different alignment methods. Specifically, we evaluated explanation-based bias detection (RQ1) on two differently aligned LLMs: Llama3.2-3B (pre-trained only, non-instruct, used with few-shot prompting) and Qwen2.5-3B-Instruct (instruction-tuned only). Neither model is aligned to human values, which differs from the models used in our main study. The results are computed for race bias on Civil Comments and shown in Figure 28.

We observe that certain explanation methods, such as Occlusion, consistently achieve high fairness correlations. This suggests that our findings generalize across LLMs with different alignment settings.

**Sensitive Token Vocabulary**   In practice, it is often unrealistic to exhaustively enumerate all vocabulary items that may encode sensitive attributes. To assess the applicability of our findings under such conditions, we analyze how varying the coverage of sensitive tokens affects bias detection and mitigation. Specifically, we focus on gender bias and use a small subset of gendered pronouns ("he/his/him" and "she/her") as sensitive tokens, while computing fairness metrics with the full gender-related vocabulary (222 words per gender). This setup simulates real-world scenarios where the sensitive vocabulary cannot be fully enumerated.

As shown in Figures 29 and 30, reduced vocabulary coverage has minimal impact on explanation-based bias detection and mitigation performance. This result is reassuring, suggesting that explanation methods remain effective in more complex, realistic settings where exhaustive sensitive token coverage is infeasible.

## I   FAITHFULNESS AS AN INDICATOR OF BIAS DETECTION ABILITY

What factors influence the reliability of explanations in detecting bias? In this section, we examine the relationship between explanation faithfulness and their ability to identify bias, reflected by

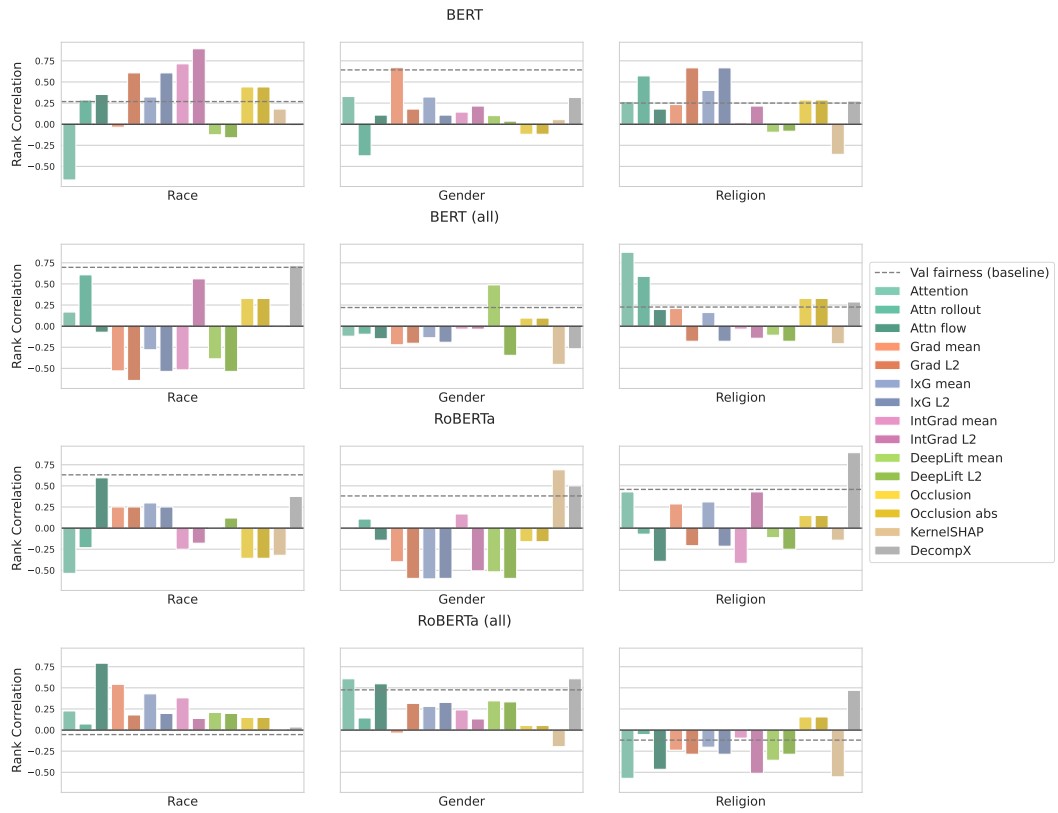

Figure 11: Rank correlations between validation set average absolute sensitive token reliance and individual unfairness on the test set for encoder-only models on Civil Comments. The validation set sizes are 500 for race, 500 for gender, and 200 for religion. Higher correlation values indicate greater effectiveness in ranking models. *All* indicates the model is trained on all bias types.

fairness correlation scores in RQ1. We assess the faithfulness of explanation methods using two perturbation-based metrics: comprehensiveness and sufficiency AOPC (Area Over the Perturbation Curve; DeYoung et al., 2020), computed by masking 5%, 10%, 20%, and 50% of the input tokens. For substitution, we use the [MASK] token in BERT and the [PAD] token in Qwen3-4B. Higher comprehensiveness and lower sufficiency scores indicate more faithful explanations.

Our results on race bias in Civil Comments (Figure 31 and Table 8) reveal no clear link between faithfulness and fairness correlation of explanations. In particular, mean-based explanations may achieve better faithfulness scores than their L2-based counterparts, yet they consistently perform significantly worse in identifying bias. We attribute this discrepancy to two key differences between the faithfulness metrics and our fairness correlation measure. First, faithfulness evaluates attribution scores across all input tokens, whereas our fairness correlation measure only considers sensitive token reliance. Second, perturbation-based faithfulness assesses the impact of masking tokens on model predictions, while our individual unfairness metric compares predictions when one social group is substituted for another. Taken together, these findings suggest that explanation faithfulness is not a reliable indicator of bias detection ability. We therefore do not recommend selecting explanation methods for fairness on the basis of faithfulness results alone.

## J  EXPLANATIONS FOR HUMAN FAIRNESS AUDITING

In addition to evaluating input-based explanations as automatic bias detectors, we also examine their ability to support human auditing of biased model predictions. To this end, we conducted a small-scale human study following the experimental protocol described below.

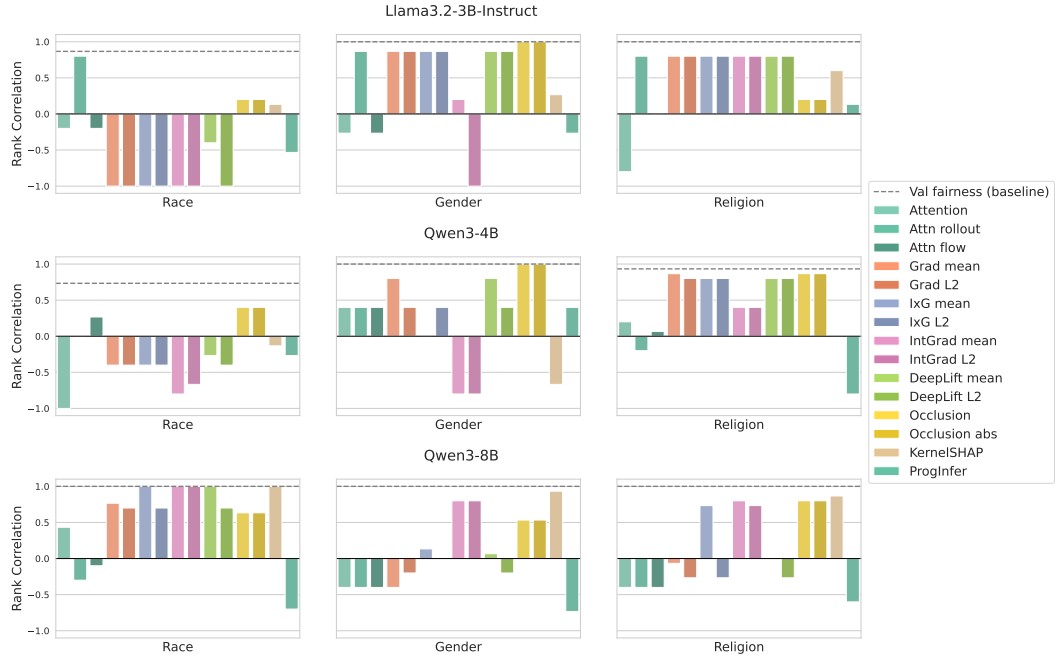

Figure 12: Rank correlations between validation set average absolute sensitive token reliance and individual unfairness on the test set for decoder-only models on Civil Comments. The validation set sizes are 500 for race, 500 for gender, and 200 for religion. Higher correlation values indicate greater effectiveness in ranking models.

Table 8: Faithfulness results of different explanation methods on BERT and Qwen3-4B models.

| Explanation | Comp. ($\uparrow$) | Suff. ($\downarrow$) | Fairness Correlation ($\uparrow$) | Comp. ($\uparrow$) | Suff. ($\downarrow$) | Fairness Correlation ($\uparrow$) |
|---|---|---|---|---|---|---|
| | | BERT | | | Qwen3-4B | |
| Attention | 4.50 | 3.20 | 0.62 | 10.34 | 17.20 | 0.14 |
| Attn rollout | 4.37 | 3.11 | 0.59 | 9.04 | 15.70 | 0.02 |
| Attn flow | 4.01 | 3.46 | 0.47 | 10.57 | 16.82 | 0.04 |
| Grad L2 | 4.82 | 2.99 | 0.50 | 12.30 | 16.09 | 0.32 |
| Grad mean | 0.77 | 6.16 | 0.06 | 11.41 | 17.50 | 0.06 |
| DeepLift L2 | 4.72 | 3.09 | 0.50 | 12.44 | 16.17 | 0.31 |
| DeepLift mean | 1.68 | 5.75 | 0.06 | 10.78 | 18.69 | 0.06 |
| IxG L2 | 4.89 | 2.95 | 0.49 | 12.35 | 16.27 | 0.32 |
| IxG mean | 7.44 | 1.70 | 0.30 | 9.99 | 18.82 | 0.17 |
| IntGrad L2 | 4.81 | 3.02 | 0.57 | 12.33 | 16.86 | 0.25 |
| IntGrad mean | 10.68 | -0.36 | 0.45 | 14.21 | 16.12 | 0.04 |
| Occlusion | 13.16 | -0.90 | 0.62 | 20.05 | 13.73 | 0.27 |
| Occlusion abs | 0.79 | 0.56 | 0.66 | 22.48 | 20.36 | 0.26 |
| KernelSHAP | 5.99 | 2.30 | 0.21 | 11.49 | 17.86 | 0.02 |
| DecompX | 16.08 | -2.77 | 0.40 | - | - | - |
| ProgInfer | - | - | - | 10.32 | 17.96 | 0.025 |

We randomly sample 48 correctly predicted examples related to race bias from Civil Comments (4*6=24 from BERT and 24 from Qwen3-4B, balanced across all group–class categories). We evaluate six explanation methods: three directed methods (Occlusion, IxG mean, KernelSHAP) and three undirected methods (Occlusion abs, IxG L2, Attention), chosen to cover diverse explanation families and performance characteristics observed in RQ1.

For each example, annotators first read the input text and provide their own toxicity prediction. They are then shown either the three directed explanations or the three undirected explanations for that example. For each explanation, annotators give two ratings on a 1–5 scale: one assessing its

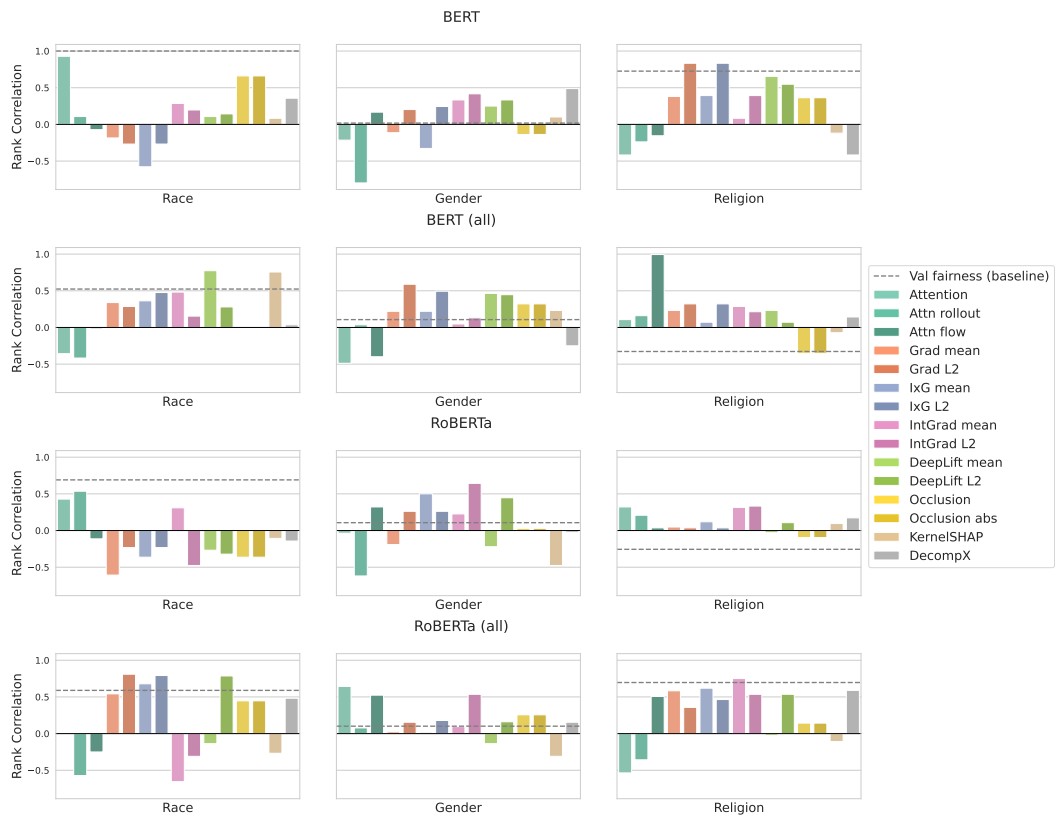

Figure 13: Rank correlations between validation set average absolute sensitive token reliance and individual unfairness on the test set for encoder-only models on Jigsaw. The validation set size is 200. Higher correlation values indicate greater effectiveness in ranking models. *All* indicates the model is trained on all bias types.

interpretability, and another evaluating how much the model's prediction appears to rely on race-related bias or stereotypes, based on the information conveyed by the explanation.

After annotation, we collect annotators' perceived bias ratings and measure their correlation with the ground-truth individual unfairness scores. Higher fairness correlation indicates greater effectiveness of an explanation method for human fairness auditing.

Table 9: Fairness correlation of explanation methods for human fairness auditing and their interpretability. Best scores in each explanation type are marked in **bold**. Higher fairness correlation scores indicate that explanations can better assist humans to detect bias.

|  | | Undirected | | | Directed | |
|  | Attention | IxG L2 | Occlusion abs | KernelSHAP | IxG mean | Occlusion |
|---|---|---|---|---|---|---|
| Fairness correlation | 0.402 | 0.123 | **0.433** | 0.254 | -0.078 | **0.342** |
| Interpretability | 2.256 | **3.179** | 2.920 | 2.518 | 2.439 | **2.780** |

Table 9 shows that certain explanation methods, such as Attention, Occlusion, and Occlusion abs, achieve high fairness correlations, suggesting that they can effectively assist humans in detecting bias. Across explanation types, undirected explanations appear more helpful. For example, Occlusion and Occlusion abs produce the same attribution patterns that differ only in directional encoding, yet participants were better able to identify bias using the undirected variant (Occlusion abs). Furthermore, while some methods support both human and automatic bias detection consistently (e.g., Attention and Occlusion abs), others, such as IxG L2, show substantial gaps in performance. This

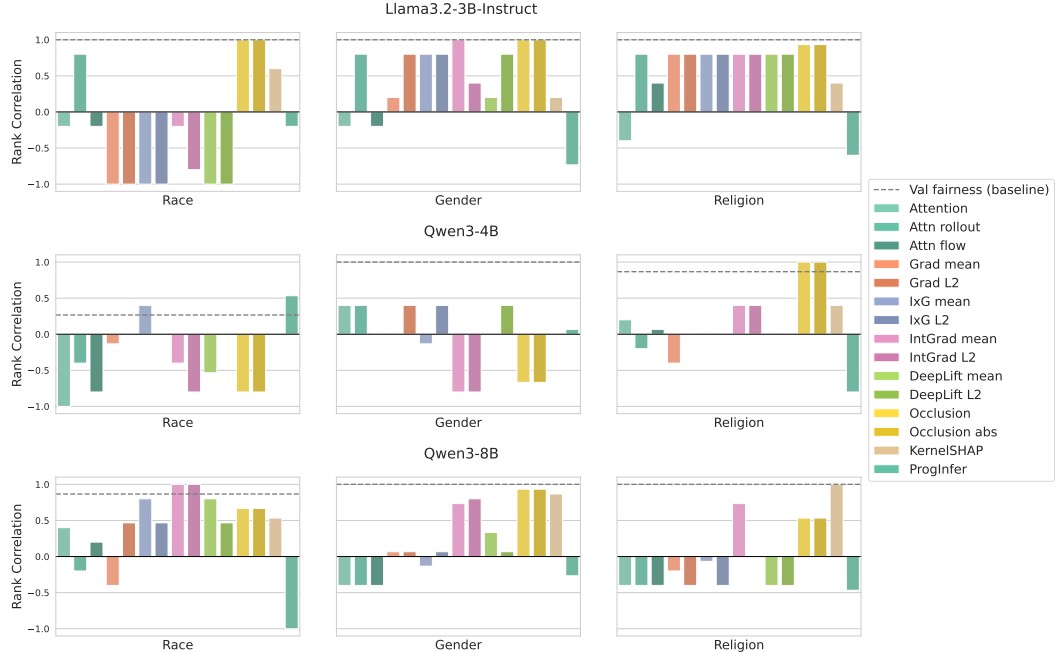

Figure 14: Rank correlations between validation set average absolute sensitive token reliance and individual unfairness on the test set for decoder-only models on Jigsaw. The validation set size is 200. Higher correlation values indicate greater effectiveness in ranking models.

Table 10: Fairness correlation of explanation methods for human fairness auditing under different conditions. Higher fairness correlation scores indicate that explanations can better assist humans to detect bias. Green (red) indicates the results are better (worse) than the baseline (all examples).

|  | Undirected | | | | Directed | |
|---|---|---|---|---|---|---|
|  | Attention | IxG L2 | Occlusion abs | KernelSHAP | IxG mean | Occlusion |
| All examples | 0.402 | 0.123 | 0.433 | 0.254 | -0.078 | 0.342 |
| Correct predictions Only | 0.572 | 0.202 | 0.602 | 0.217 | 0.029 | 0.451 |
| Toxic examples only | 0.637 | 0.118 | 0.374 | 0.231 | 0.134 | 0.315 |
| High interpretability (score $\geq$ 3) | 0.138 | 0.227 | 0.288 | 0.154 | -0.043 | 0.404 |

highlights a potential discrepancy between how humans interpret explanations and how our automatic pipeline evaluates them.

We also observe that high interpretability alone does not guarantee better support for human fairness auditing: methods with strong interpretability scores (e.g., IxG L2) still fail to effectively help humans detect bias. Finally, 4 out of 6 annotators reported that undirected explanations helped them detect bias more effectively, noting that they introduce less noise and make annotation easier.

Table 10 further analyzes explanation-assisted human fairness auditing under different conditions. For correctly predicted examples, explanations generally provide stronger support for bias detection. However, the effects of label type and explanation interpretability appear more nuanced and vary across methods. Overall, these results suggest that explanation-assisted human fairness auditing is a promising and interesting direction for future work and warrants further investigation.

## K  HYBRID BIAS MITIGATION TECHNIQUES

we conducted preliminary experiments that combine several pre-processing techniques (group balance, group–class balance, and CDA) with an effective explanation-based debiasing method (IxG

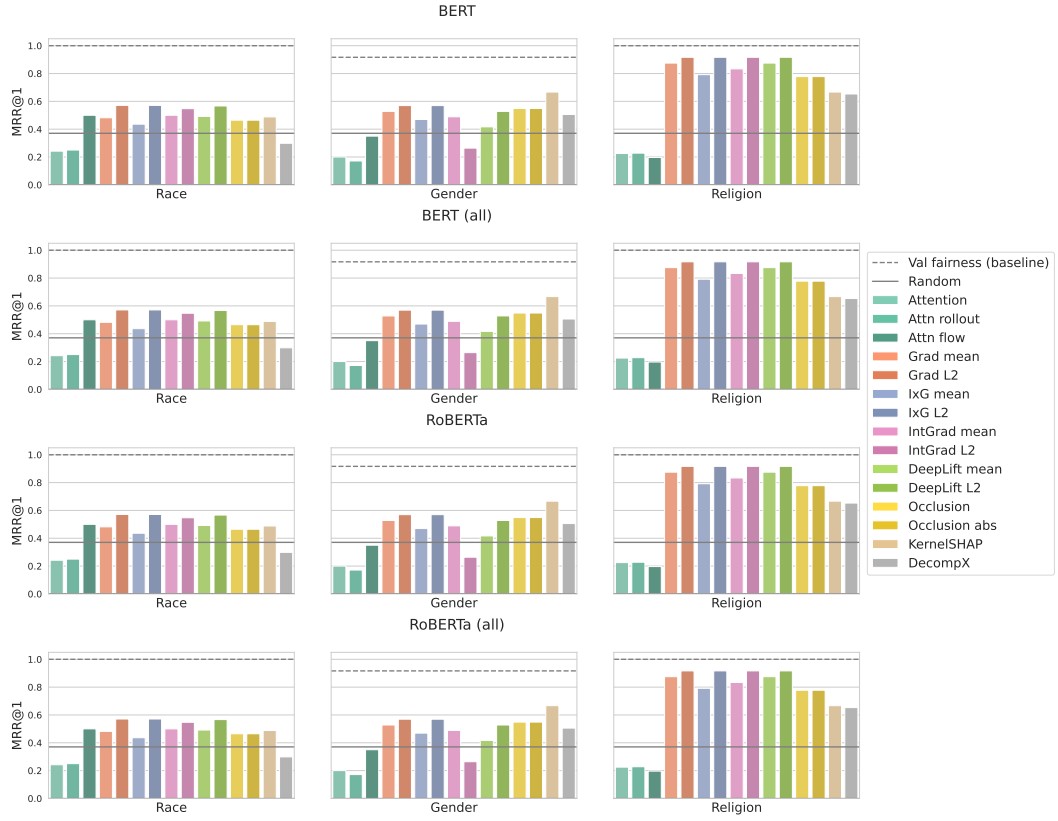

Figure 15: MRR@1 results for encoder-only models on Civil Comments. The validation set sizes are 500 for race, 500 for gender, and 200 for religion. Higher MRR@1 scores indicate explanations are more effective in selecting the fairest models. *All* indicates the model is trained on all bias types.

L1/L2). The resulting individual fairness outcomes, along with comparisons to traditional and explanation-only methods, are presented in Table 11.

Table 11: Each cell shows the $\text{Avg}_{\text{iu}}$ score after applying a combination of pre-processing and explanation-based debiasing methods. Lower values indicate reduced bias. Values in parentheses denote the change relative to using only the corresponding traditional/explanation-based method, where negative values indicate improved debiasing. We observe that hybrid approaches consistently achieve stronger bias mitigation than either method used in isolation.

| | Race | | |
|---|---|---|---|
| | Group balance | Group-class balance | CDA |
| IxG L1 | 0.012 (-4.492/-1.461) | 0.000 (-3.048/-1.473) | 0.001 (-0.547/-1.473) |
| IxG L2 | 2.162 (-2.342/-0.598) | 2.110 (-0.938/-0.650) | 0.210 (-0.338/-2.550) |
| | Gender | | |
| | Group balance | Group-class balance | CDA |
| IxG L1 | 0.005 (-0.594/-0.548) | 0.001 (-0.836/-0.552) | 0.001 (-0.488/-0.551) |
| IxG L2 | 0.308 (-0.291/-0.331) | 0.546 (-0.292/-0.093) | 0.368 (-0.122/-0.271) |

We observe that the hybrid method achieves better bias mitigation effects than using each debiasing method alone, with consistent improvements for both race and gender bias. Based on this, we believe exploring hybrid methods for more effective bias mitigation could be a promising future direction.

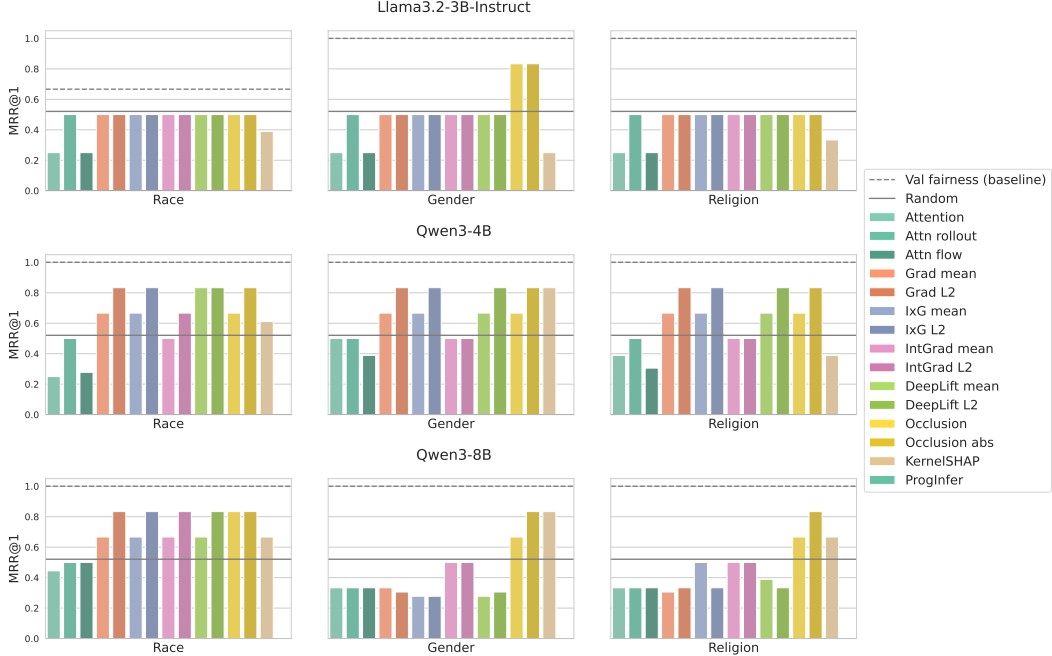

Figure 16: MRR@1 results for decoder-only models on Civil Comments. The validation set sizes are 500 for race, 500 for gender, and 200 for religion. Higher MRR@1 scores indicate explanations are more effective in selecting the fairest models.

## L    FAIRNESS CORRELATIONS IN EXPLANATION-DEBIASED MODELS

Figure 32 presents the fairness correlation scores computed on explanation-debiased models. We find that Grad L2, IxG L2, DeepLift L2, and Occlusion-based explanations still show strong bias mitigation ability in the debiased models.

## M    LIMITATIONS AND FUTURE WORK

Our study has several limitations that we acknowledge and aim to address in future work.

First, as the first quantitative investigation of this topic, our study focuses solely on hate speech detection and uses a limited set of experimental setups. Although the results are consistent across these setups and preliminary experiments (Appendix H) suggest good generalization across tasks, models and sensitive token vocabulary, broader validation is still needed. Future work could extend this evaluation to additional domains and high-stakes applications.

Second, several findings are derived under the specific experimental setups used in this work. For instance, in RQ2, we conclude that the proposed attribution-based metrics are not reliable fairness indicators. However, it remains possible that other metrics could be effective. Similarly, our fairness-balanced metric in RQ3 may not be the optimal validation strategy in all settings. As it is infeasible to exhaustively enumerate and evaluate all potential configurations, we believe our conclusions nonetheless offer valuable guidance and highlight important methodological considerations for the community.

Third, our work focuses on evaluating standalone explanation-based strategies for improving fairness. Ensembles of multiple explanation methods, or hybrid approaches that combine explanation techniques with established debiasing methods, may yield better outcomes. Additionally, incorporating human oversight may further enhance the effectiveness and robustness of explanation-based fairness auditing. Our preliminary experiments show promising results in using hybrid debiasing techniques K, and demonstrates the possibility for human fairness auditing based on explanations J.

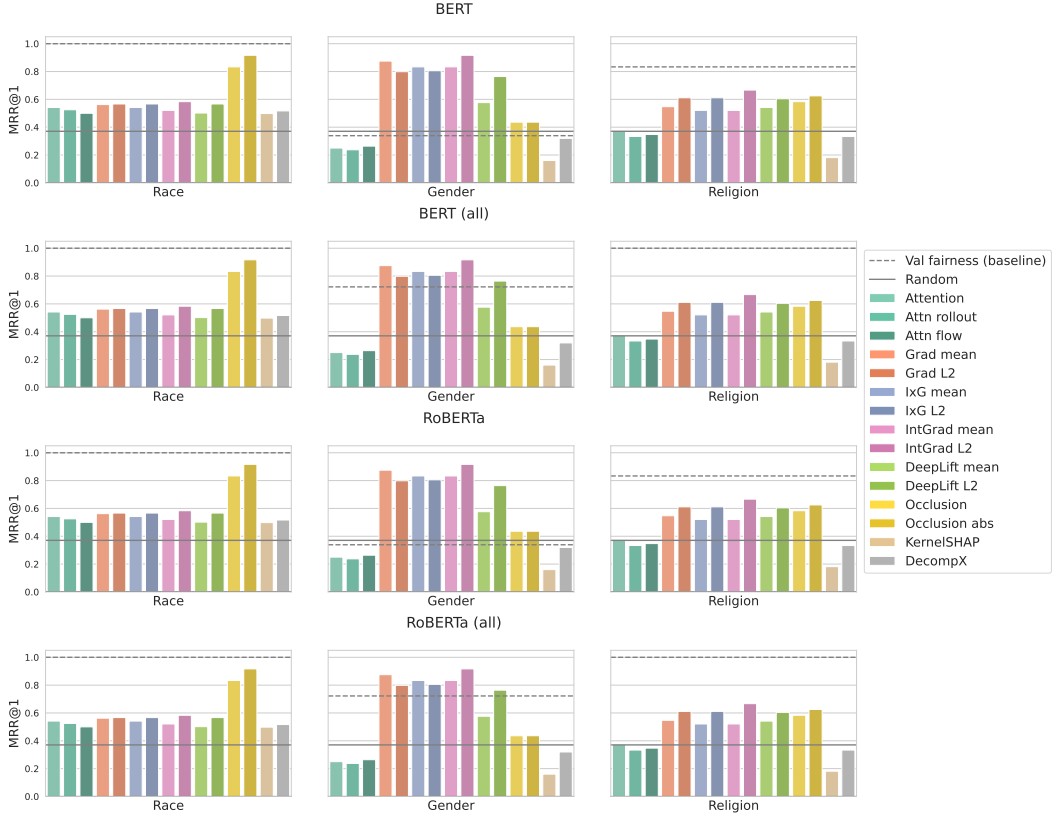

Figure 17: MRR@1 results for encoder-only models on Jigsaw. The validation set size is 200. Higher MRR@1 scores indicate explanations are more effective in selecting the fairest models. *All* indicates the model is trained on all bias types.

Based on that, we believe that a systematic investigation of such hybrid or human-in-the-loop approaches represents an interesting avenue for future work.

Fourth, we do not identify any explanation method that consistently outperforms others across all research questions, which prevents us from offering a single recommendation. We therefore encourage future researchers to choose explanation methods that align with their specific tasks and constraints. Future work could further investigate why certain methods are better suited to particular settings and, ideally, develop practical guidelines for selecting effective methods without requiring extensive empirical comparisons.

Finally, although we consider both group and individual fairness, this work provides a more in-depth analysis of individual fairness (in RQ1 and RQ2), driven by the conceptual alignment between input-based explanations and individual fairness notions. We encourage future work to more thoroughly examine how explanation methods relate to group fairness.

## N    LLM USAGE

Apart from the models evaluated in our experiments and analyses, we used LLMs (ChatGPT) solely to polish the writing in this work.

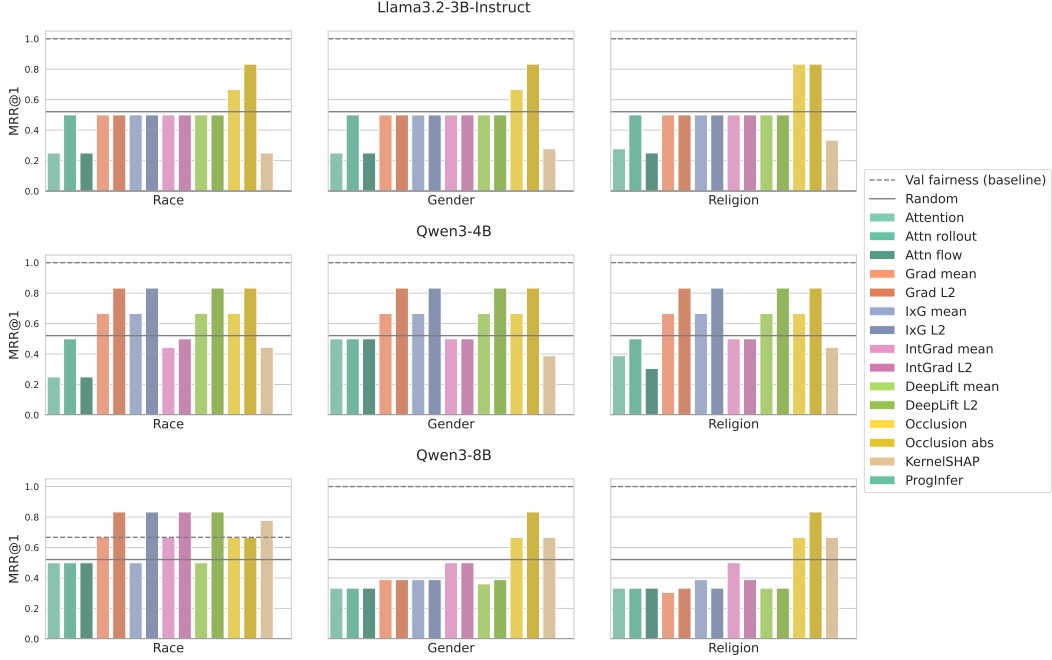

Figure 18: MRR@1 results for decoder-only models on Jigsaw. The validation set size is 200. Higher MRR@1 scores indicate explanations are more effective in selecting the fairest models.

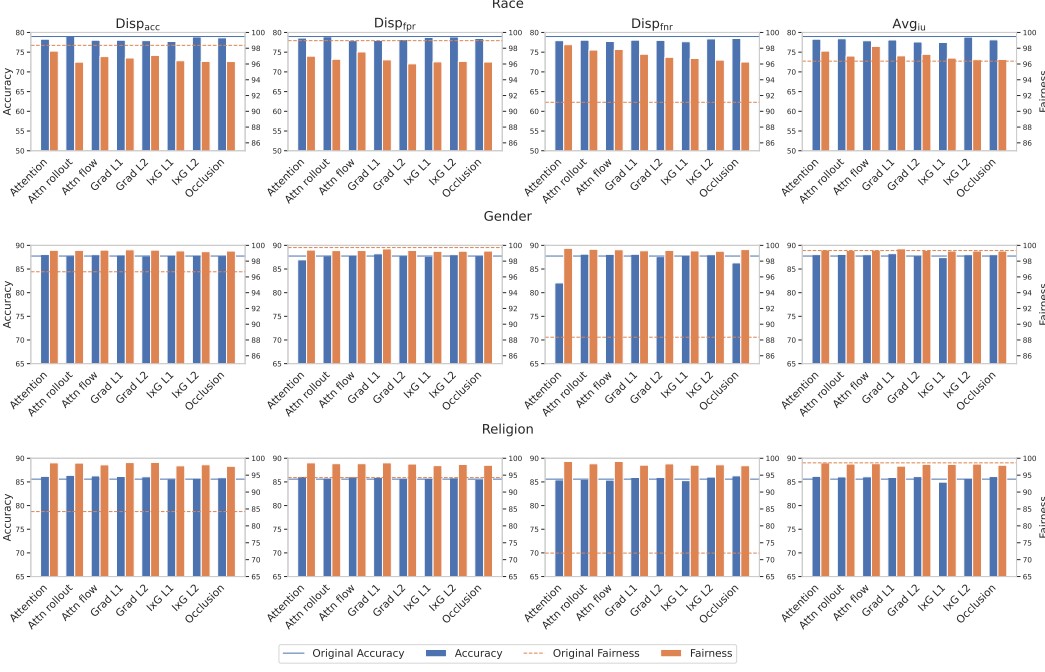

Figure 19: Accuracy and fairness results for bias mitigation in BERT on the Civil Comments dataset, using different explanation methods during training. For consistency with accuracy, fairness results are reported as $100 - \{\text{Disp}_{\text{acc}}, \text{Disp}_{\text{fpr}}, \text{Disp}_{\text{fnr}}, \text{Avg}_{\text{iu}}\}$, so that higher values indicate better debiasing performance. Each column corresponds to models selected by maximizing the fairness-balanced metric with respect to the indicated bias metric.

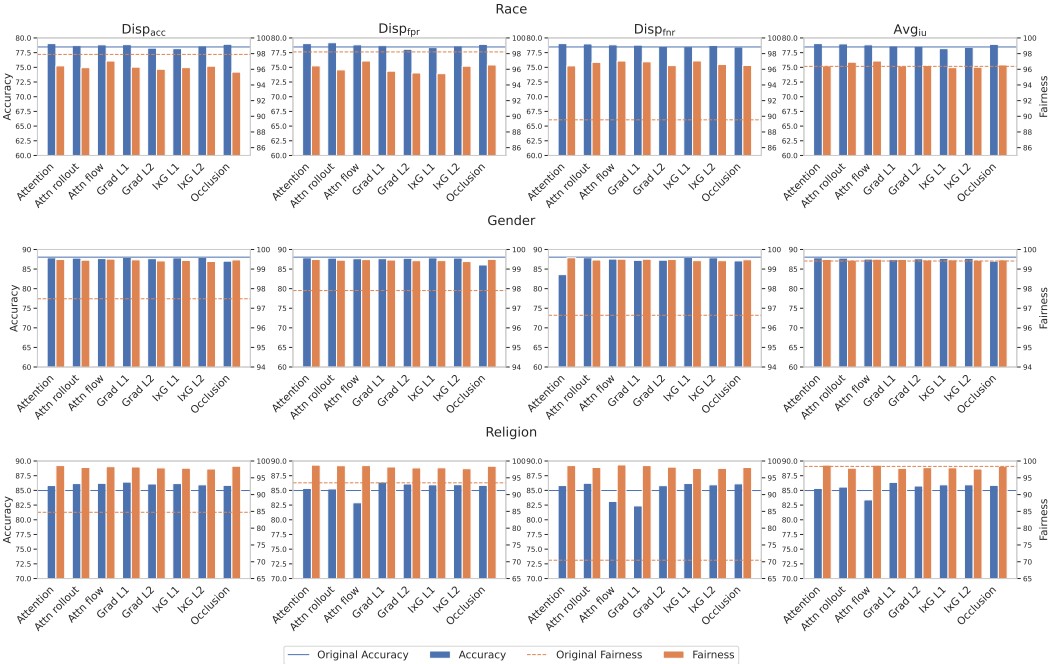

Figure 20: Accuracy and fairness results for bias mitigation in RoBERTa on the Civil Comments dataset, using different explanation methods during training. For consistency with accuracy, fairness results are reported as $100 - \{\text{Disp}_{\text{acc}}, \text{Disp}_{\text{fpr}}, \text{Disp}_{\text{fnr}}, \text{Avg}_{\text{iu}}\}$, so that higher values indicate better debiasing performance. Each column corresponds to models selected by maximizing the fairness-balanced metric with respect to the indicated bias metric.

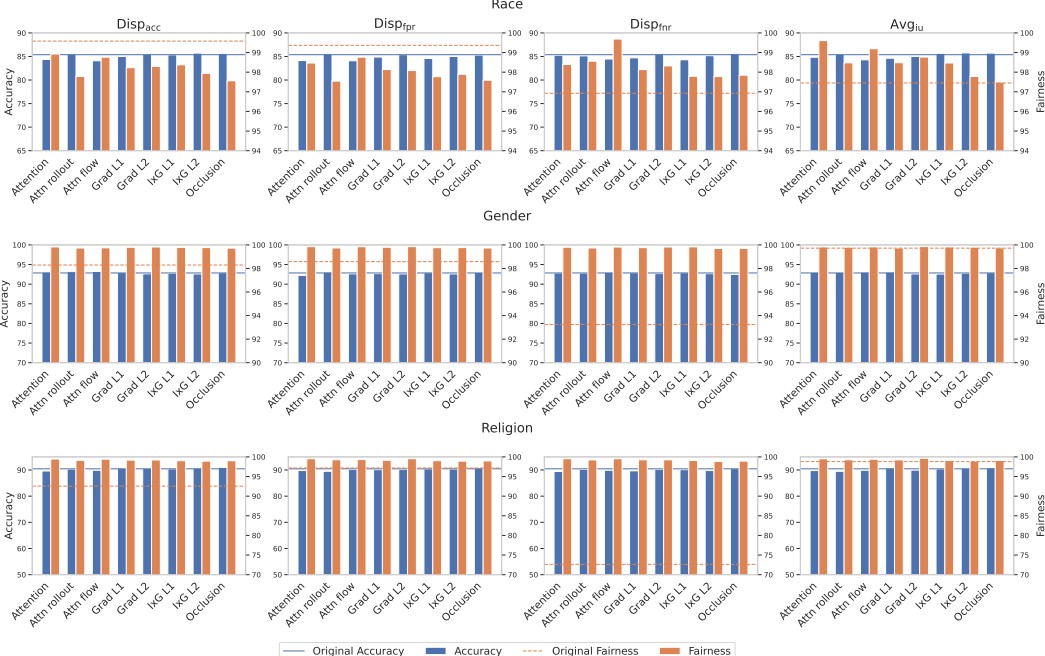

Figure 21: Accuracy and fairness results for bias mitigation in BERT on the Jigsaw, using different explanation methods during training. For consistency with accuracy, fairness results are reported as $100 - \{\text{Disp}_{\text{acc}}, \text{Disp}_{\text{fpr}}, \text{Disp}_{\text{fnr}}, \text{Avg}_{\text{iu}}\}$, so that higher values indicate better debiasing performance. Each column corresponds to models selected by maximizing the fairness-balanced metric with respect to the indicated bias metric.

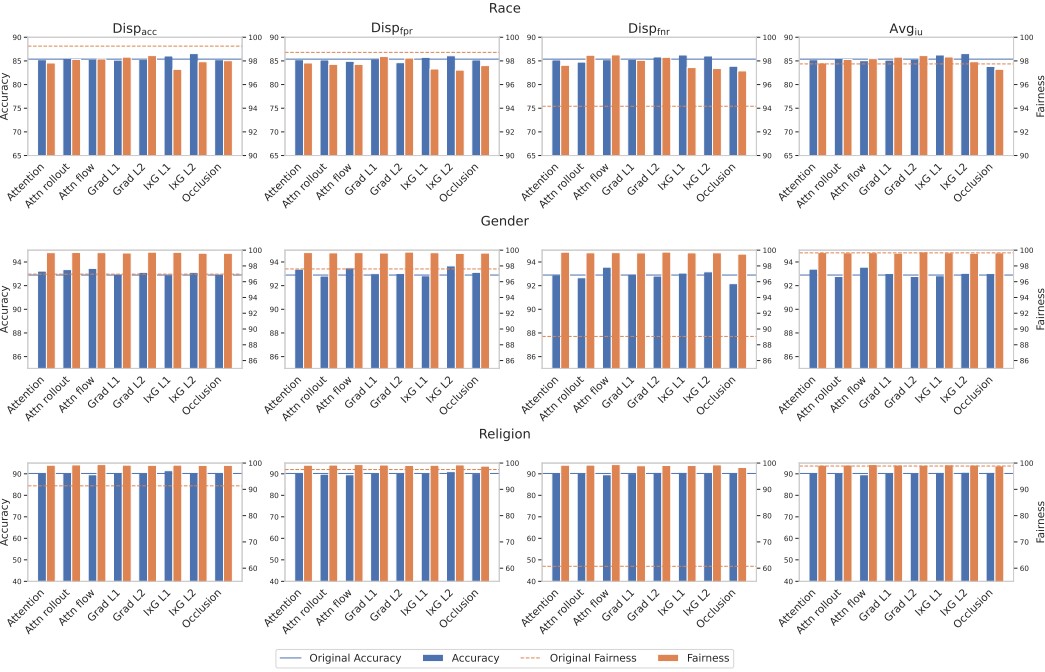

Figure 22: Accuracy and fairness results for bias mitigation in RoBERTa on the Jigsaw dataset, using different explanation methods during training. For consistency with accuracy, fairness results are reported as $100 - \{\text{Disp}_{\text{acc}}, \text{Disp}_{\text{fpr}}, \text{Disp}_{\text{fnr}}, \text{Avg}_{\text{iu}}\}$, so that higher values indicate better debiasing performance. Each column corresponds to models selected by maximizing the fairness-balanced metric with respect to the indicated bias metric.

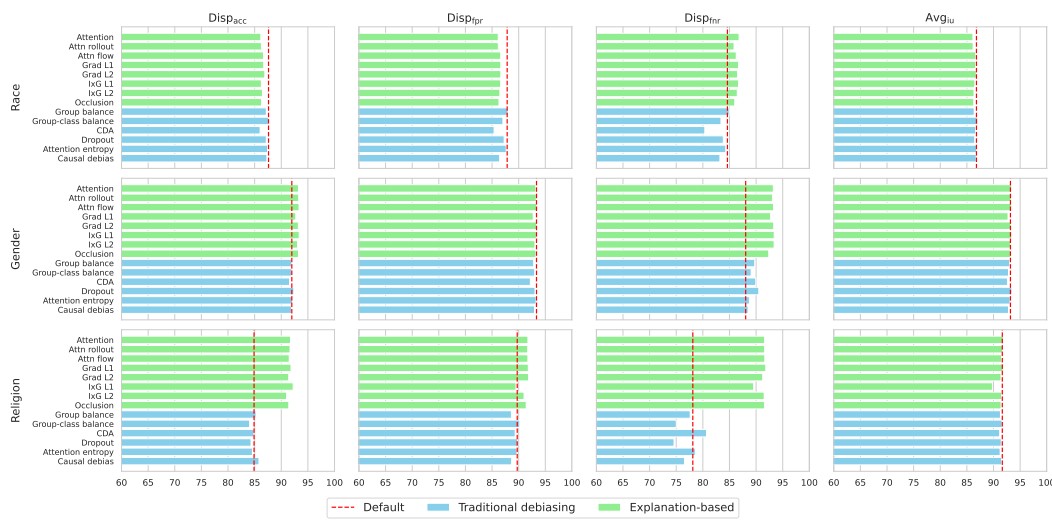

Figure 23: Harmonic mean between accuracy and fairness for established debiasing methods and explanation-based methods for BERT on Civil Comments. A higher score indicates better balance between model performance and fairness.

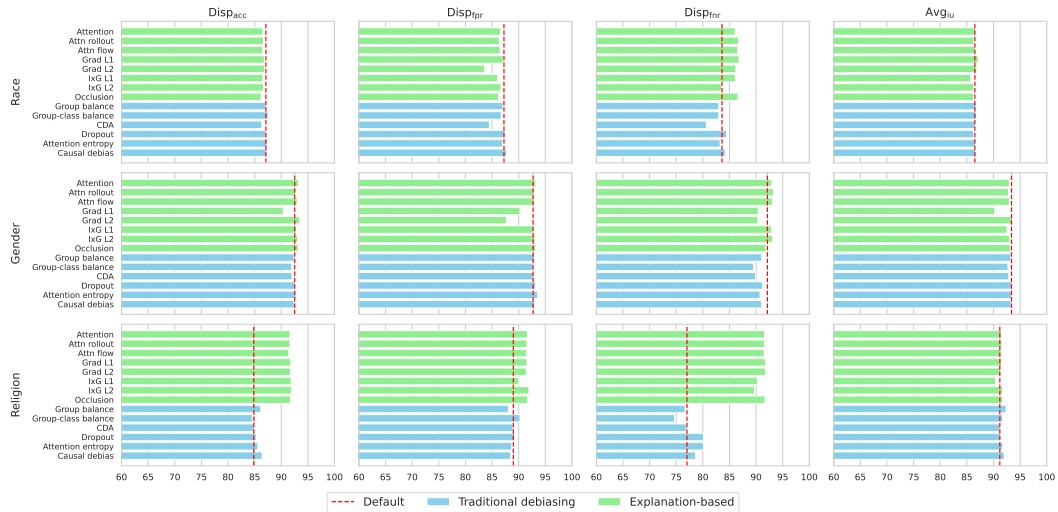

Figure 24: Harmonic mean between accuracy and fairness for established debiasing methods and explanation-based methods for RoBERTa on Civil Comments. A higher score indicates better balance between model performance and fairness.

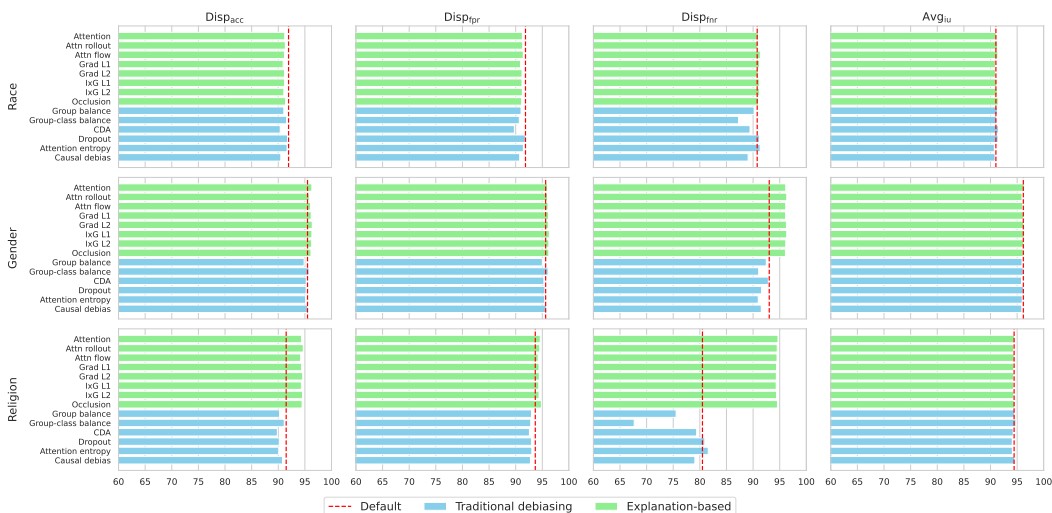

Figure 25: Harmonic mean between accuracy and fairness for established debiasing methods and explanation-based methods for BERT on Jigsaw. A higher score indicates better balance between model performance and fairness.

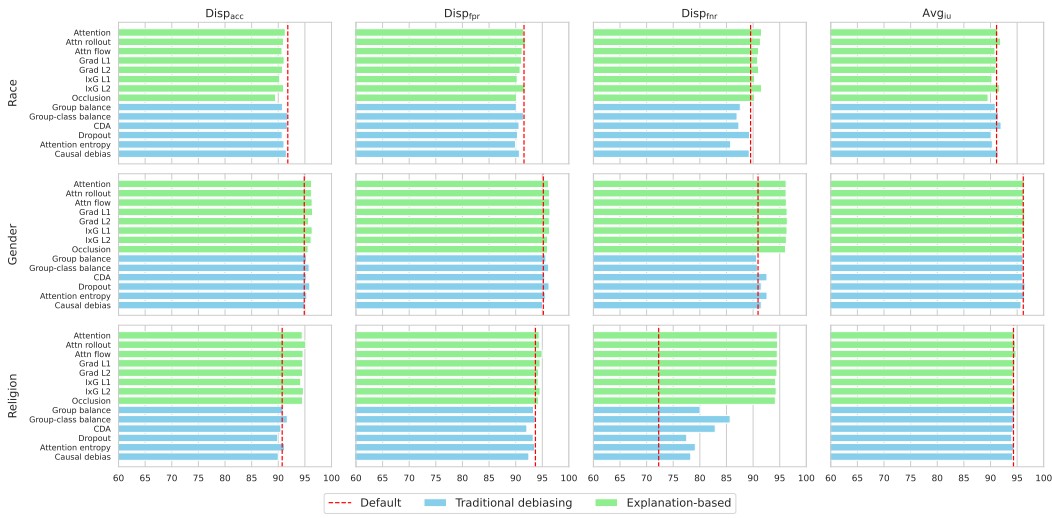

Figure 26: Harmonic mean between accuracy and fairness for established debiasing methods and explanation-based methods for RoBERTa on Jigsaw. A higher score indicates better balance between model performance and fairness.

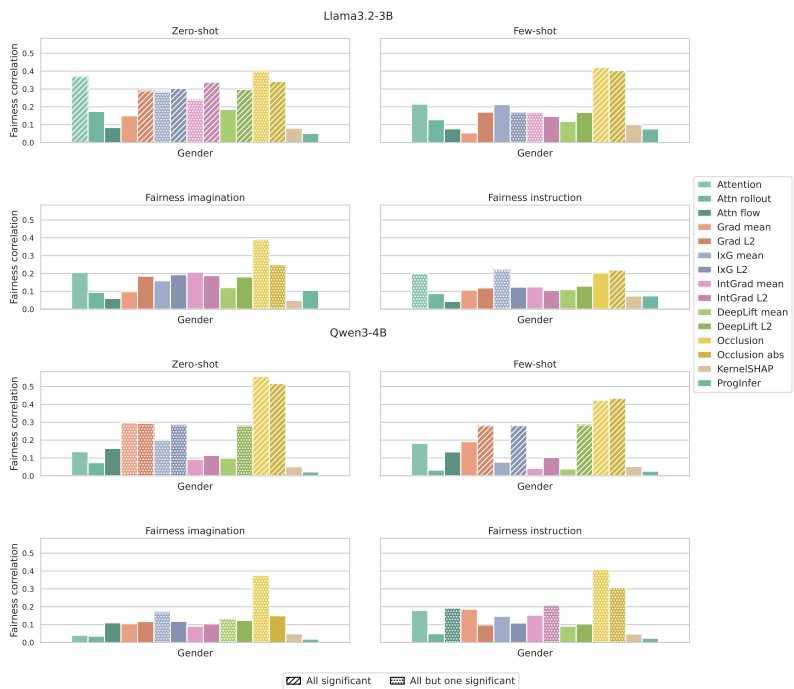

Figure 27: Fairness correlation results for race bias on Twitter Sentiment with Llama3.2-3B-Instruct and Qwen3-4B. Higher values indicate that the method is more effective and reliable in detecting biased predictions at inference time.

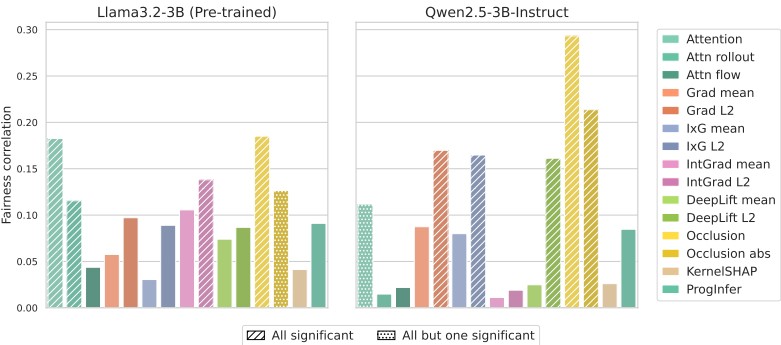

Figure 28: Fairness correlation results for race bias on Civil Comments with Llama3.2-3B and Qwen2.5-3B-Instruct. Both models are differently aligned from models in our main experiments. Higher values indicate that the method is more effective and reliable in detecting biased predictions at inference time.

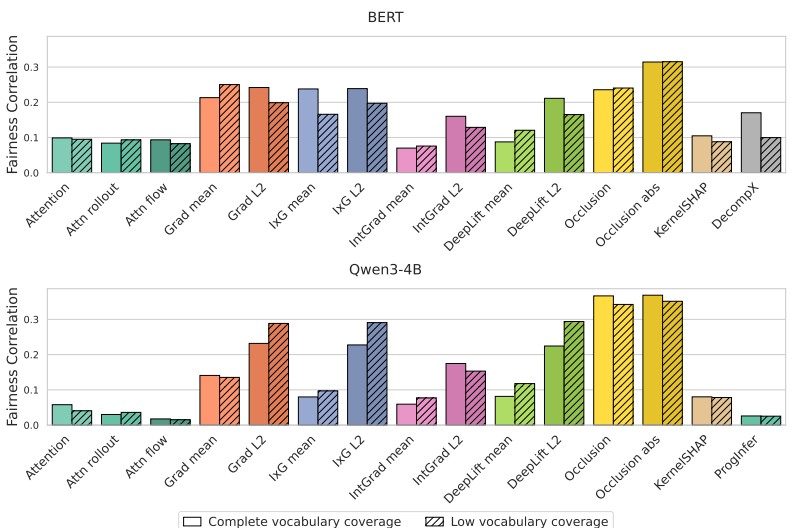

Figure 29: Fairness correlation results when using a reduced sensitive token vocabulary for reliance computation. Results are reported for gender bias on the Civil Comments dataset. Fairness metrics are still computed using the full vocabulary. The reduced vocabulary size has only a marginal effect on fairness correlations.

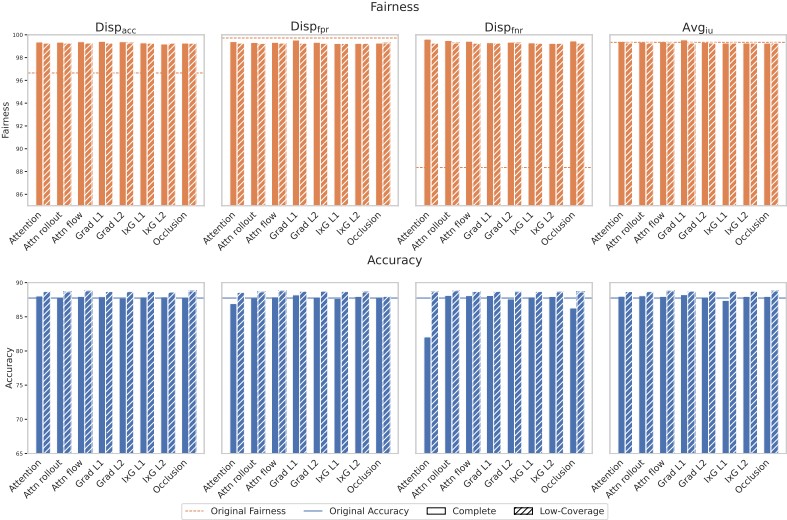

Figure 30: Fairness and accuracy results for gender bias mitigation with a reduced sensitive token vocabulary. Each column corresponds to models selected by maximizing the fairness–balance metric with respect to the indicated bias metric. Using an incomplete vocabulary yields slightly worse debiasing performance than using the complete vocabulary, while preserving task performance more effectively. Overall, the impact of reduced vocabulary coverage is minimal.

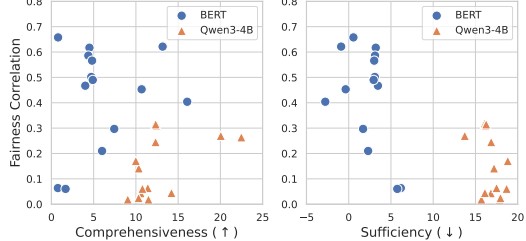

Figure 31: Faithfulness and fairness correlation results of different explanation methods. No clear relationship between explanation faithfulness and their bias detection ability is observed. Each point represents the faithfulness and fairness correlation of one explanation method applied to default/zero-shot models.

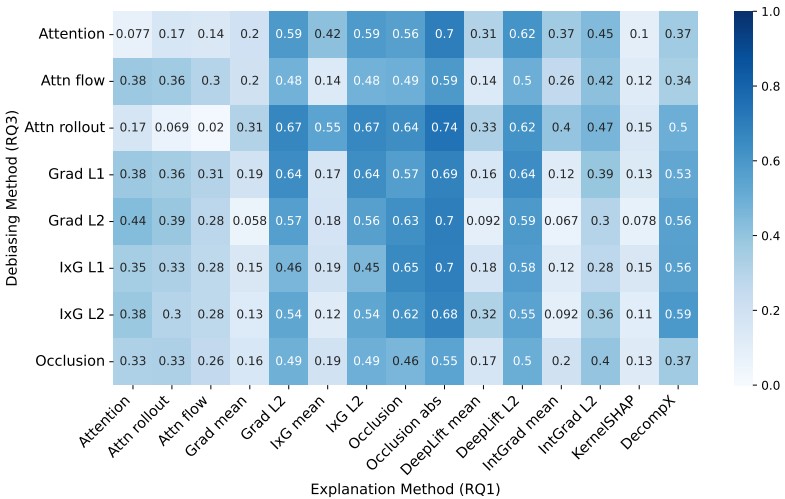

Figure 32: Fairness correlation results on BERT models with race bias mitigated through explanation-based methods on Civil Comments.

