# OpenReview forum: "Bridging Fairness and Explainability: Can Input-Based Explanations Promote Fairness in Hate Speech Detection?"
_ICLR.cc/2026/Conference — ICLR 2026 Poster_

### Official Review · Reviewer_EN8y · 2025-10-29

**Soundness:** 4
**Presentation:** 4
**Contribution:** 3
**Rating:** 8
**Confidence:** 4

**Summary:**

The paper presents a quantitative study of the relationship between input-based explanations and fairness of NLP models in the context of hate speech detection. It focuses on three aspects in particular: (1) identifying biased predictions using input-based explanations, (2) selecting fair models based on input-based explanations, and (3) mitigating bias during model training using input-based explanations. The authors find that input-based explanations are effective at detecting biased predictions and reducing biased predictions of NLP models during training but are not suitable as a selection criterion for fair models.

**Strengths:**

Originality: The paper appears to be the first systematic quantitative study of the relationship between input-based explanations and fairness of NLP models. The paper also investigates a broad set of 14 input-based explanation methods across 4 models (2 encoder-only and 2 decoder-only models) on 2 hate speech detection datasets.

Quality: The metrics used for measurement of bias/fairness are thoughtful and seem to be good approximations of model fairness. While the chosen method is not highly complex, it is well-executed and effective at measuring bias/fairness of models and their predictions and determine suitability of different input-based explanations for detecting model bias.

Clarity: The explanation of the approach and its relevance as well as the presentation of results are very clear. All concepts (e.g., sensitive tokens, attribution scores, sensitive token reliance scores, etc.) are properly introduced with both a mathematical definition and a natural language explanation. The overall argumentation and section structure makes sense and has a logical flow, always leaving the reader aware of what (e.g., which research question) is being discussed in any particular paragraph. The paper also makes use of suitable visual elements such as the takeaway boxes briefly summarizing the answers to the research questions.

Significance: The three research questions raised by the paper are of high significance to the research community and motivate further research into improving model fairness using input-based explanations. The paper also presents a very comprehensive related work section, pointing to other relevant work on related aspects.

**Weaknesses:**

As highlighted in the strengths, the paper is generally very well-written and presents rigorous research, leaving only a few minor weaknesses:

Some figures take a while to fully understand, given the large number of methods, metrics, and settings shown. For instance, in Fig. 2, the caption mentions that no model outperforms the baseline and it is not immediately apparent which bar or line represents the baseline. This becomes clear only after reading the accompanying text mentioning the “individual unfairness on the validation set” as the baseline and then looking back at the figure, where the legend includes one line “Val fairness”. Possibly, the authors could simply adjust the legend entry to “Val fairness (baseline)” to make this clear. Further, the visual style of the “Val fairness” line is different between Figs. 2 and 3 and could be harmonized. Fig. 4 uses the terms “Accuracy” and “Fairness” in both the legend and on the chart axes. One must read the charts very carefully to notice that each subplot has two y-axes, one for accuracy and one for fairness, which follow slightly different scales.

In section 6, the paper presents the important finding that input-based explanations outperform an LLM-as-a-Judge method on bias detection. This would be a highly significant finding given the current prevalence of LLM-as-a-Judge methods. However, the analysis is only based on a Qwen3 LLM judge (presumably the same 4-billion-parameter Qwen3 LLM used for the other experiments), which is a relatively small LLM and cannot be expected to reach the impressive performance levels of state-of-the-art LLMs. While potential budget constraints might make this choice understandable, it limits the generalizability of the finding to LLM-as-a-Judge and LLMs overall.

Across most experiments, the study’s results indicate no consistently best method. While this lies in the nature of empirical research, the practical implications of the findings are not always clear. In the conclusion, the authors claim that they “provide practical recommendations on which explanation methods are best suited” (lines 483f.) but the given recommendations fall somewhat short of being practical and useful. As a practitioner or researcher, it is not always clear what method should be used based on the findings of this study. At the very least, I suggest that the authors add a section discussing the study’s limitations and outlining future work to come closer to such practical recommendations.

**Questions:**

Based on the mentioned weaknesses, two questions can be raised to the authors:

1. How can the results from section 6 (i.e., that input-based explanations outperform LLM-as-a-Judge) be expected to generalize to other, particularly larger and state-of-the-art, LLMs?

2. What are the limitations of the study that should be addressed by future work?

---

> ### Author Response · Authors · 2025-11-23
> **Response to Reviewer EN8y (1/N)**
>
> Thank you for recognizing our contributions and for providing constructive feedback. Below, we provide detailed responses to your questions and comments. If any of our responses do not sufficiently address your concerns, please let us know, and we will follow up promptly. **Please note that we are still integrating new experimental results and discussion into the revised manuscript. The update is ongoing, and we plan to upload the revised version by November 26. We will notify the reviewers once it is available and include a general response summarizing all revisions.**
>
> ## Figure Readability
>
> We appreciate your careful reading and valuable feedback. Following your suggestions, we have updated the "Val fairness" entries in Figures 2 and 3 (RQ2) to "Val fairness (baseline)" and applied consistent colors and styles across both figures to clearly indicate the baseline method. For Figure 4 (RQ3), we added left and right y-axes for each subplot to make it explicit that accuracy and fairness are displayed on different scales. These revisions have also been applied to all corresponding figures in the Appendix and will be included in the revised manuscript.
>
> If you believe further improvements to the revised figures or any additional figures would enhance clarity or readability, we would be glad to update them during the rebuttal period and in the camera-ready version.
>
> ## [W2 & Q1] LLM-as-a-Judge with Larger LLMs
>
> We have incorporated a larger LLM, **GPT-OSS-120B**, as the judge model to assess whether each prediction is biased. To reflect this update, we rename our two LLM-as-a-judge setups (previously self-reflection and self-attribution) to *LLM decision* and *LLM attribution*, which more accurately describe scenarios where an LLM evaluates predictions from another model. The results are presented below.
>
> |LLM|Method|# Biased/Unbiased|$\text{Avg}_\text{iu}$ (Biased/Unbiased)|Fairness Correlation|
> |---|---|----|---|---|
> |**Qwen3-4B**|LLM decision|86/3914|0.065/2.59|-|
> ||LLM attribution (K=5)|2063/1904|3.55/1.49|0.104|
> ||LLM attribution (K=10)|2176/1474|2.93/1.56|0.070|
> |**GPT-OSS-120B**|LLM decision|399/3601|4.42/2.35|0.051|
> ||LLM attribution (K=5)|2153/1843|3.33/1.65|0.092|
> ||LLM attribution (K=10)|2729/1238|2.88/1.74|0.063|
> |—|Grad L2 Binary|2000/2000|5.02/0.09|0.194|
>
> Table: Results of LLM-as-a-judge for bias detection using Qwen3-4B and GPT-OSS-120B. Predictions come from Qwen3-4B on race-related Civil Comments examples. "Biased/Unbiased" denotes whether an example is judged as biased or unbiased by the LLM through LLM decision or LLM attribution. If the judgments are reliable, $\text{Avg}_\text{iu}$ should be higher for biased examples than unbiased ones. Input-based explanations reveal bias more reliably than LLM judgments.
>
> Our updated results show that although GPT-OSS-120B performs substantially better than Qwen3-4B in the LLM decision setup, its performance in the LLM attribution setup does not improve and still markedly underperforms relative to explanation-based bias detection. These results reinforce our conclusion that explanation-based approaches are more reliable than LLM-as-a-judge methods.

---

> > ### Author Response · Authors · 2025-11-23
> > **Response to Reviewer EN8y (2/N)**
> >
> > ## [W3 & Q2] Limitations and Future Work
> >
> > We will add a "Limitations and Future Work" section to discuss the limitations that the reviewers raised and those identified in our experiments.
> > ```
> > Limitations and Future Work
> >
> > Our study has several limitations that we acknowledge and aim to address in future work.
> >
> > First, as the first quantitative investigation of this topic, our study focuses solely on hate speech detection and uses a limited set of experimental setups. Although the results are consistent across these setups and preliminary experiments suggest good generalization to other use cases, broader validation is still needed. Future work could extend our evaluation to additional tasks, datasets, and bias types.
> >
> > Second, several findings are derived under the specific experimental setups used in this work. For instance, in RQ2, we conclude that the proposed attribution-based metrics are not reliable fairness indicators. However, it remains possible that other metrics could be effective. Similarly, our fairness-balanced metric in RQ3 may not be the optimal validation strategy in all settings. As it is infeasible to exhaustively enumerate and evaluate all potential configurations, we believe our conclusions nonetheless offer valuable guidance and highlight important methodological considerations for the community.
> >
> > Third, our work focuses on evaluating standalone explanation-based strategies for improving fairness. Ensembles of multiple explanation methods, or hybrid approaches that combine explanation techniques with established debiasing methods, may yield better outcomes. Additionally, incorporating human oversight may further enhance the effectiveness and robustness of explanation-based fairness auditing. Our preliminary experiments show promising trends in these directions, and a systematic investigation of such hybrid or human-in-the-loop approaches represents an interesting avenue for future work.
> >
> > Fourth, we do not identify any explanation method that consistently outperforms others across all research questions, which prevents us from offering a single recommendation. We therefore encourage future researchers to choose explanation methods that align with their specific tasks and constraints. Future work could further investigate why certain methods are better suited to particular settings and, ideally, develop practical guidelines for selecting effective methods without requiring extensive empirical comparisons.
> >
> > Finally, although we consider both group and individual fairness, this work provides a more in-depth analysis of individual fairness (in RQ1 and RQ2), driven by the conceptual alignment between input-based explanations and individual fairness notions. We encourage future work to more thoroughly examine how explanation methods relate to group fairness.
> > ```
> > We will also include some preliminary results addressing the identified limitations and exploring the future directions in the revised paper.

---

### Official Review · Reviewer_7Yw8 · 2025-10-30

**Soundness:** 2
**Presentation:** 3
**Contribution:** 3
**Rating:** 6
**Confidence:** 5

**Summary:**

The paper does a very nice job scoping the problem and running a broad comparison of 14 explanation methods across two hate-speech datasets and multiple bias types. It also delivers clear, actionable takeaways for practitioners, notably that certain explanation families (occlusion, L2-like gradients) are good local bias sensors and can supervise debiasing, and that explanation signals can be more useful than an LLM-as-a-judge for this task. However, the core claim the paper wants to make, “input-based explanations can support fairness operations in hate speech detection”, rests on a data and measurement setup that is still quite narrow: two toxicity datasets, fixed identity vocabularies, and counterfactuals built via simple token replacement. Furthermore, the paper itself presents a negative result on model selection but does not delve into why explanations fail in this context, nor does it conduct the stronger causal/feature-removal tests that would strengthen the attribution-model-reliance link.

**Strengths:**

S1: This is a relatively large, well-scoped empirical study that ties input-based explanations directly to fairness in hate speech detection, a gap that existing work had mostly treated qualitatively.


S2: Clear question framing that makes the contributions easy to map to real moderation pipelines.

S3: Systematic comparison of 14 explanation methods from different families (attention, gradients, occlusion, SHAP), providing an unusually comprehensive view of which types of attributions are useful for fairness.

S4: Consistent positive result for bias detection: occlusion and L2-style gradient attributions correlate well with individual unfairness across settings.

S5: Demonstration that explanation-based regularization can improve several fairness metrics without collapsing task accuracy, showing a concrete way to operationalize the insight.

S6: Practical comparison showing that explanation-based signals can outperform an LLM-as-a-judge approach for this task, which is directly actionable for current LLM pipelines.

**Weaknesses:**

W1: Although the study is broad within hate speech detection, it is still confined to Civil Comments and Jigsaw, with only three bias types and a fixed identity vocabulary, so it is unclear how far the conclusions extend to other NLP tasks or richer identity expressions.

W2: Sensitive-token detection relies on predefined identity terms, so paraphrased, code-mixed, reclaimed, or multiword identity mentions that arise in practice may be undercounted, potentially underestimating model bias.

W3: The individual-unfairness construction depends on counterfactuals built via simple token replacement and balanced subsampling; while workable, this can introduce measurement artifacts that partially blur how much of the signal comes from explanations vs. from the counterfactual setup.

W4: While the paper does include decoder-side experiments and an LLM-as-a-judge comparison, the decoder evaluation is still relatively thin and focuses on smaller instruction-tuned models, so it is uncertain whether the findings hold for larger or differently aligned LLMs.

W5: Sensitive-token reliance is hard-coded to a particular aggregation rule (max over sensitive tokens); alternative aggregations (sum, top-k, span-aware) are not compared, so part of the conclusion may be an artifact of this choice.

W6: There is no human-in-the-loop or practitioner-style evaluation to show that the surfaced explanations are actually interpretable and actionable for fairness analysts.

**Questions:**

Q1: What's the cost/latency for the 14 explanation methods (time per 1k examples, hardware) to support practicality claims?

Q2: Will the authors be able to add a small human/practitioner evaluation (5–10 cases) on whether the explanations are actionable for fairness auditing?

---

> ### Author Response · Authors · 2025-11-23
> **Response to Reviewer 7Yw8 (1/N)**
>
> We would like to thank the reviewer for recognizing our contributions and for the time and effort dedicated to providing insightful comments and suggestions. We have carefully considered each point raised and provide detailed responses below. We apologize for the length of this response, as we aim to address your questions as clearly and comprehensively as possible. **Please note that we are still integrating new experimental results and discussion into the revised manuscript. The update is ongoing, and we plan to upload the revised version by November 26. We will notify the reviewers once it is available and include a general response summarizing all revisions.**
>
> ## [W1–W5: Generalization of the Current Study]
>
> The reviewer's main concern appears to be whether our findings generalize beyond the specific setups used in this study. We address this concern for each setup as follows. The results will also be included in the revised manuscript.
>
> ### Task (W1)
>
> We evaluated explanation-based bias detection (RQ1) on an additional task, namely **sentiment analysis**, using the Twitter Sentiment dataset. Specifically, we selected 1000 gender-related examples (500 referencing males and 500 referencing females) and ran explanation-based bias detection (RQ1) on them. Below we report the results on zero-shot Llama3.2-3B and Qwen3-4B models, and the complete results will be included in the revised manuscript.
>
> Llama3.2-3B (zero-shot)
>
> ||Attention|Attn rollout|Attn flow|Grad mean|Grad L2|IxG mean|IxG L2|IntGrad mean|IntGrad L2|DeepLift mean|DeepLift L2|Occlusion|Occlusion abs|KernelSHAP|ProgInfer|
> |---|----|----|----|----|----|----|----|----|----|----|----|----|----|----|----|
> |Fairness correlation|0.368 (4)|0.174 (2)|0.084 (0)|0.150 (2)|0.289 (4)|0.283 (3)|0.303 (4)|0.239 (3)|0.337 (4)|0.186 (2)|0.297 (4)|0.401 (3)|0.342 (4)|0.081 (0)|0.051 (1)
>
> Qwen3-4B (zero-shot)
>
> ||Attention|Attn rollout|Attn flow|Grad mean|Grad L2|IxG mean|IxG L2|IntGrad mean|IntGrad L2|DeepLift mean|DeepLift L2|Occlusion|Occlusion abs|KernelSHAP|ProgInfer|
> |---|----|----|----|----|----|----|----|----|----|----|----|----|----|----|----|
> |Fairness correlation|0.135 (2)|0.073 (1)|0.153 (2)|0.299 (3)|0.294 (3)|0.200 (3)|0.286 (3)|0.092 (2)|0.114 (1)|0.099 (1)|0.279 (3)|0.555 (4)|0.517 (4)|0.050 (0)|0.022 (0)|
>
> Tables: Fairness correlation results and numbers of significant group-class category (in parentheses) for each explanation for Llama3.2-3B and Qwen3-4B on the Twitter Sentiment dataset. Higher scores indicate that an explanation method is more effective at detecting biased predictions.
>
> We observe similar patterns in other classification settings, where certain explanation methods (e.g., occlusion-based and L2-based approaches) also achieve high fairness correlation scores. This suggests that our findings likely extend beyond the hate speech detection task.
>
> ### Datasets and Bias Types (W1)
>
> Our study examines two datasets and three bias types, and we observe consistent effectiveness of explanation-based bias detection and mitigation across all of them. These consistent patterns indicate that our findings should generalize beyond the specific datasets and bias categories evaluated in this work.

---

> > ### Author Response · Authors · 2025-11-23
> > **Response to Reviewer 7Yw8 (2/N)**
> >
> > ### Fixed Sensitive Token Vocabulary (W1)
> >
> > We further analyze the gender subset of the Civil Comments dataset to assess the impact of the sensitive token vocabulary. Specifically, we restrict the sensitive token set to only *"he/his/him"* and *"she/her"* for sensitive token reliance computation in bias detection (RQ1) and mitigation (RQ3), while fairness metrics are still computed using the full vocabulary (222 words). The corresponding results are shown below (due to the space limit, we only report the individual fairness metric for RQ3 and the complete results can be found in the revised manuscript).
> >
> > Fairness Correlation (RQ1): BERT
> >
> > ||Attention|Attn rollout|Attn flow|Grad mean|Grad L2|IxG mean|IxG L2|IntGrad mean|IntGrad L2|DeepLift mean|DeepLift L2|Occlusion|Occlusion abs|KernelSHAP|DecompX|
> > |---|---|---|---|---|---|---|---|---|---|---|---|---|---|---|---|
> > |Complete Vocabulary Coverage|0.099|0.084|0.094|0.213|0.242|0.238|0.239|0.070|0.160|0.088|0.211|0.236|0.314|0.105|0.170|
> > |Low Vocabulary Coverage|0.095 (-0.004)|0.093 (+0.009)|0.083 (-0.011)|0.250 (+0.037)|0.199 (-0.043)|0.166 (-0.072)|0.197 (-0.041)|0.076 (+0.006)|0.129 (-0.031)|0.121 (+0.033)|0.165 (-0.046)|0.241 (+0.005)|0.315 (+0.001)|0.088 (-0.017)|0.100 (-0.070)|
> >
> > Fairness Correlation (RQ1): Qwen3-4B
> >
> > ||Attention|Attn rollout|Attn flow|Grad mean|Grad L2|IxG mean|IxG L2|IntGrad mean|IntGrad L2|DeepLift mean|DeepLift L2|Occlusion|Occlusion abs|KernelSHAP|ProgInfer|
> > |---|---|---|---|---|---|---|---|---|---|---|---|---|---|---|---|
> > |Complete Vocabulary Coverage|0.058|0.030|0.018|0.141|0.232|0.080|0.227|0.059|0.175|0.081|0.224|0.367|0.369|0.080|0.026|
> > |Low Vocabulary Coverage|0.041 (-0.017)|0.036 (+0.006)|0.015 (-0.002)|0.135 (-0.006)|0.289 (+0.057)|0.097 (+0.017)|0.291 (+0.063)|0.077 (+0.018)|0.153 (-0.022)|0.118 (+0.036)|0.294 (+0.069)|0.342 (-0.024)|0.351 (-0.017)|0.078 (-0.002)|0.025 (-0.001)|
> >
> > Tables: Fairness correlation results when using an incomplete sensitive token vocabulary, compared with using the full vocabulary for bias detection. Higher scores indicate that an explanation method is more effective at identifying biased predictions.
> >
> > Bias Mitigation on BERT (RQ3): $\text{Avg}_\text{iu}$ (Lower scores indicate less bias.)
> >
> > ||Attention|Attn rollout|Attn flow|Grad L1|Grad L2|IxG L1|IxG L2|Occlusion|
> > |---|---|---|---|---|---|---|---|---|
> > |Complete Vocabulary Coverage|0.651|0.719|0.717|0.586|0.631|0.604|0.441|0.724|
> > |Low Vocabulary Coverage|0.688 (+0.037)|0.737 (+0.019)|0.747 (+0.030)|0.659 (+0.072)|0.696 (+0.065)|0.638 (+0.034)|0.693 (+0.252)|0.693 (-0.031)|
> >
> > Bias Mitigation on BERT (RQ3): Accuracy (Higher scores are better.)
> >
> > ||Attention|Attn rollout|Attn flow|Grad L1|Grad L2|IxG L1|IxG L2|Occlusion|
> > |---|---|---|---|---|---|---|---|---|
> > |Complete Vocabulary Coverage|87.833|87.408|87.992|88.017|88.083|87.992|88.242|87.992|
> > |Low Vocabulary Coverage|87.975 (+0.142)|87.842 (+0.433)|87.858 (-0.133)|87.717 (-0.300)|87.650 (-0.433)|88.125 (+0.133)|87.925 (-0.317)|88.075 (+0.083)|
> >
> > Tables: $\text{Avg}_\text{iu}$ and task accuracy after explanation-based bias mitigation using incomplete vs. complete sensitive token vocabularies.
> >
> > We observe that the reduced vocabulary size has only a marginal impact on the effectiveness of both bias detection and bias mitigation.
> >
> > ### Individual Fairness Measurement (W2 & W3)
> >
> > In our work, we measure fairness using a fixed vocabulary and a simple token replacement procedure to compute individual fairness. We agree that this approach may underestimate the degree of bias in some models. However, this setup is widely adopted in the fairness literature [1-5], making it both standard and comparable across studies.
> >
> > While some prior work employs neural counterfactual generators [6, 7], these methods require training additional models, which introduces substantial complexity and computational overhead. Given that our study is already computationally intensive, incorporating neural counterfactual generation would have made the overall evaluation infeasible. For these reasons, we rely on simple, widely used heuristics for computing individual fairness metrics in this work.

---

> > > ### Author Response · Authors · 2025-11-23
> > > **Response to Reviewer 7Yw8 (3/N)**
> > >
> > > ### Larger and Differently Aligned LLMs (W4)
> > >
> > > We extended our experiments to additional LLMs with different sizes and alignment methods. Specifically, we ran the full set of experiments (RQ1 and RQ2) on a larger model, **Qwen3-8B**, and evaluated explanation-based bias detection (RQ1) on two differently aligned LLMs: **Llama3.2-3B** (pre-trained only, non-instruct, used with few-shot prompting) and **Qwen2.5-3B-Instruct** (instruction-tuned only). Both models are not aligned to human values, which differs from models used in our main study. A subset of their fairness correlation results is shown below.
> > >
> > > Fairness Correlation (RQ1): Llama3.2-3B (pre-trained, few-shot):
> > >
> > > ||Attention|Attn rollout|Attn flow|Grad mean|Grad L2|IxG mean|IxG L2|IntGrad mean|IntGrad L2|DeepLift mean|DeepLift L2|Occlusion|Occlusion abs|KernelSHAP|ProgInfer|
> > > |---|---|---|---|---|---|---|---|---|---|---|---|---|---|---|---|
> > > |Fairness correlation|0.183 (4)|0.115 (4)|0.044 (1)|0.058 (1)|0.097 (2)|0.030 (1)|0.089 (2)|0.106 (2)|0.138 (4)|0.074 (2)|0.087 (2)|0.185 (4)|0.127 (3)|0.041 (0)|0.091 (2)|
> > >
> > > Fairness Correlation (RQ1): Qwen2.5-3B-Instruct (zero-shot):
> > >
> > > ||Attention|Attn rollout|Attn flow|Grad mean|Grad L2|IxG mean|IxG L2|IntGrad mean|IntGrad L2|DeepLift mean|DeepLift L2|Occlusion|Occlusion abs|KernelSHAP|ProgInfer|
> > > |---|---|---|---|---|---|---|---|---|---|---|---|---|---|---|---|
> > > |Fairness correlation|0.112 (3)|0.015 (0)|0.022 (0)|0.087 (1)|0.170 (4)|0.080 (1)|0.165 (4)|0.011 (0)|0.019 (0)|0.025 (0)|0.161 (4)|0.293 (4)|0.214 (4)|0.026 (0)|0.085 (1)|
> > >
> > > Fairness Correlation (RQ1): Qwen3-8B (zero-shot)
> > >
> > > ||Attention|Attn rollout|Attn flow|Grad mean|Grad L2|IxG mean|IxG L2|IntGrad mean|IntGrad L2|DeepLift mean|DeepLift L2|Occlusion|Occlusion abs|KernelSHAP|ProgInfer|
> > > |---|---|---|---|---|---|---|---|---|---|---|---|---|---|---|---|
> > > |Fairness correlation|0.066 (2)|0.035 (0)|0.028 (0)|0.073 (2)|0.105 (2)|0.018 (0)|0.112 (2)|0.045 (1)|0.079 (1)|0.029 (0)|0.108 (2)|0.304 (4)|0.236 (4)|0.093 (2)|0.063 (1)|
> > >
> > > Tables: Fairness correlation results for LLMs of different sizes and alignment settings. Higher scores indicate that an explanation method is more effective at identifying biased predictions.
> > >
> > > Overall, we observe that certain explanation methods, such as Occlusion abs, consistently demonstrate high fairness correlations across LLMs of different sizes and alignment settings. In addition, the full experimental results on Qwen3-8B exhibit patterns that closely align with our main findings in the paper.
> > >
> > > For the LLM-as-a-judge experiment, we have incorporated a larger LLM, **GPT-OSS-120B**, as the judge model to assess whether each prediction is biased. To reflect this update, we rename our two LLM-as-a-judge setups (previously self-reflection and self-attribution) to *LLM decision* and *LLM attribution*, which more accurately describe scenarios where an LLM evaluates predictions from another model. The results are presented below.
> > >
> > > |LLM|Method|# Biased/Unbiased|$\text{Avg}_\text{iu}$ (Biased/Unbiased)|Fairness Correlation|
> > > |---|---|----|---|---|
> > > |**Qwen3-4B**|LLM decision|86/3914|0.065/2.59|-|
> > > ||LLM attribution (K=5)|2063/1904|3.55/1.49|0.104|
> > > ||LLM attribution (K=10)|2176/1474|2.93/1.56|0.070|
> > > |**GPT-OSS-120B**|LLM decision|399/3601|4.42/2.35|0.051|
> > > ||LLM attribution (K=5)|2153/1843|3.33/1.65|0.092|
> > > ||LLM attribution (K=10)|2729/1238|2.88/1.74|0.063|
> > > |—|Grad L2 Binary|2000/2000|5.02/0.09|0.194|
> > >
> > > Table: Results of LLM-as-a-judge for bias detection using Qwen3-4B and GPT-OSS-120B. Predictions come from Qwen3-4B on race-related Civil Comments examples. "Biased/Unbiased" denotes whether an example is judged as biased or unbiased by the LLM through LLM decision or LLM attribution. If the judgments are reliable, $\text{Avg}_\text{iu}$ should be higher for biased examples than unbiased ones. Input-based explanations reveal bias more reliably than LLM judgments.
> > >
> > > Our updated results show that although GPT-OSS-120B performs substantially better than Qwen3-4B in the LLM decision setup, its performance in the LLM attribution setup does not improve and still markedly underperforms relative to explanation-based bias detection. These results reinforce our conclusion that explanation-based approaches are more reliable than LLM-as-a-judge methods.

---

> > > > ### Author Response · Authors · 2025-11-23
> > > > **Response to Reviewer 7Yw8 (4/N)**
> > > >
> > > > ### Sensitive Token Reliance Computation (W5)
> > > >
> > > > To assess the robustness of our reliance computation, we compared two additional aggregation methods, **sum** and **average** (over the absolute reliance scores of all sensitive tokens), in RQ1, alongside their differences to the original max approach. The corresponding results on race bias in Civil Comments are shown below.
> > > >
> > > > Fairness Correlation (RQ1): BERT
> > > >
> > > > |Method|Attention|Attn rollout|Attn flow|Grad mean|Grad L2|IxG mean|IxG L2|IntGrad mean|IntGrad L2|DeepLift mean|DeepLift L2|Occlusion|Occlusion abs|KernelSHAP|DecompX|
> > > > |---|---|---|---|---|---|---|---|---|---|---|---|---|---|---|---|
> > > > |Max aggregation (paper)|0.617|0.586|0.467|0.064|0.496|0.297|0.490|0.453|0.566|0.060|0.502|0.621|0.658|0.210|0.404|
> > > > |Sum aggregation|0.426 (-0.191)|0.396 (-0.190)|0.305 (-0.162)|0.426 (+0.362)|0.597 (+0.101)|0.400 (+0.103)|0.586 (+0.096)|0.346 (-0.108)|0.497 (-0.069)|0.372 (+0.311)|0.608 (+0.106)|0.701 (+0.080)|0.701 (+0.044)|0.241 (+0.031)|0.497 (+0.093)|
> > > > |Avg aggregation|0.337 (-0.280)|0.307 (-0.279)|0.258 (-0.209)|0.371 (+0.307)|0.513 (+0.017)|0.344 (+0.047)|0.508 (+0.018)|0.273 (-0.180)|0.404 (-0.162)|0.302 (+0.241)|0.528 (+0.026)|0.627 (+0.006)|0.627 (-0.031)|0.185 (-0.025)|0.395 (-0.009)|
> > > >
> > > > Fairness Correlation (RQ1): Qwen3-4B
> > > >
> > > > |Method|Attention|Attn rollout|Attn flow|Grad mean|Grad L2|IxG mean|IxG L2|IntGrad mean|IntGrad L2|DeepLift mean|DeepLift L2|Occlusion|Occlusion abs|KernelSHAP|ProgInfer|
> > > > |---|---|---|---|---|---|---|---|---|---|---|---|---|---|---|---|
> > > > |Max aggregation (paper)|0.142|0.019|0.044|0.064|0.318|0.171|0.315|0.045|0.245|0.061|0.311|0.270|0.264|0.019|0.025|
> > > > |Sum aggregation|0.143 (+0.001)|0.021 (+0.003)|0.046 (+0.002)|0.199 (+0.135)|0.330 (+0.012)|0.307 (+0.136)|0.326 (+0.011)|0.142 (+0.097)|0.215 (-0.030)|0.304 (+0.242)|0.322 (+0.011)|0.271 (+0.002)|0.271 (+0.007)|0.053 (+0.035)|0.024 (-0.000)|
> > > > |Avg aggregation|0.127 (-0.014)|0.014 (-0.005)|0.036 (-0.008)|0.169 (+0.105)|0.299 (-0.019)|0.283 (+0.113)|0.296 (-0.019)|0.147 (+0.103)|0.250 (+0.005)|0.265 (+0.204)|0.293 (-0.018)|0.249 (-0.021)|0.249 (-0.015)|0.046 (+0.027)|0.020 (-0.005)|
> > > >
> > > > Tables: Fairness correlation results using different aggregation methods for computing sensitive token reliance. Higher scores indicate that an explanation method is more effective at identifying biased predictions.
> > > >
> > > > Although the results show minor differences compared to the max aggregation approach, our conclusions regarding the effectiveness of explanation-based bias detection and mitigation, and the relative performance of different methods, remain unchanged. Given the infeasibility of exhaustively exploring all possible aggregation strategies, we continue to use max aggregation in our main analysis. Furthermore, as mentioned in Footnote 1, we experimented with several alternative ways of deriving reliance scores from attribution scores (normalization by sequence length, maximum magnitude, or attribution-vector norm), but none provided improvements over using the raw attribution scores.
> > > >
> > > > ### Conclusion for Generalizability of Our Findings
> > > >
> > > > Based on the additional experiments and supporting arguments, we believe our findings generalize to new setups. Nonetheless, we will note this in the newly added **Limitations and Future Work** section and encourage future researchers to verify our conclusions in their own use cases before applying them.

---

> > > > > ### Author Response · Authors · 2025-11-23
> > > > > **Response to Reviewer 7Yw8 (5/N)**
> > > > >
> > > > > ## [W6 & Q2] Explanations for Human Fairness Auditing
> > > > >
> > > > > We appreciate the reviewer's suggestion to conduct a human study on explanations for fairness auditing. Although this lies outside the original scope of our work, we agree that it can meaningfully extend our findings. We are therefore running a human study following the protocol outlined below.
> > > > >
> > > > > We randomly sample 48 correctly predicted examples related to race bias from Civil Comments (4*6=24 from BERT and 24 from Qwen3-4B, balanced across all group–class categories). We evaluate six explanation methods: three directed methods (Occlusion, IxG mean, KernelSHAP) and three undirected methods (Occlusion abs, IxG L2, Attention), chosen to cover diverse explanation families and performance characteristics observed in RQ1.
> > > > >
> > > > > For each example, annotators first read the input text and provide their own toxicity prediction. They are then shown either the three directed explanations or the three undirected explanations for that example. For each explanation, annotators give two ratings on a 1–5 scale: one assessing its interpretability, and another evaluating how much the model's prediction appears to rely on race-related bias or stereotypes, based on the information conveyed by the explanation.
> > > > >
> > > > > After annotation, we collect annotators' perceived bias ratings and measure their correlation with the ground-truth individual unfairness scores. Higher correlation indicates greater effectiveness of an explanation method for human fairness auditing.
> > > > >
> > > > > We will analyze explanation-based fairness auditing performance in relation to interpretability, example category, and whether human predictions align with the model's prediction. Finally, after completing all examples, annotators will be asked which type of explanation, directed or undirected, they find easier for fairness auditing, providing an additional qualitative evaluation dimension.
> > > > >
> > > > > We are currently carrying out the human study, and the full experimental protocol along with the results will be included in the revised version of the paper.
> > > > >
> > > > >
> > > > > ## [Q1] Time and GPU Memory Costs of Explanation Methods
> > > > >
> > > > > We have added the time and GPU memory costs for all 16 explanation methods (the original 14 plus 2 newly added ones).
> > > > >
> > > > > |**BERT**|Time (s/example)|Memory (GB)|**Qwen3-4B**|Time (s/example)|Memory (GB)|
> > > > > |---|---|---|-----|---|---|
> > > > > |Attention|0.027|0.529|Attention|0.112|16.598|
> > > > > |Grad|0.026|0.603|Grad|0.237|19.631|
> > > > > |IxG|0.025|0.603|IxG|0.236|19.631|
> > > > > |IntGrad|0.064|7.010|IntGrad|1.784|101.694|
> > > > > |DeepLift|0.027|0.748|DeepLift|0.323|23.53|
> > > > > |Occlusion|0.330|0.508|Occlusion|4.204|15.639|
> > > > > |KernelSHAP|0.138|0.508|KernelSHAP|1.374|20.013|
> > > > > |DecompX|0.074|7.222|ProgInfer|0.068|15.81|
> > > > >
> > > > > Table: Computational costs per example for generating explanations across 200 instances on BERT and Qwen3-4B. Results are computed on the race subset of the Civil Comments dataset using a batch size of 1 and are averaged over three runs. All methods are run on a single 80-GB H100 GPU, except IntGrad, which uses two H100 GPUs with gradient checkpointing to reduce memory usage. Because explanation methods within the same family incur similar computational costs, we report each family only once.
> > > > >
> > > > > ## References
> > > > >
> > > > > [1] Garg et al., "Counterfactual Fairness in Text Classification through Robustness", AIES 2019
> > > > >
> > > > > [2] Czarnowska et al., "Quantifying Social Biases in NLP: A Generalization and Empirical Comparison of Extrinsic Fairness Metrics", TACL 2021
> > > > >
> > > > > [3] Dorner et al., "Human-Guided Fair Classification for Natural Language Processing", ICLR 2023
> > > > >
> > > > > [4] Chen et al., "Addressing Both Statistical and Causal Gender Fairness in NLP Models", NAACL Findings, 2024
> > > > >
> > > > > [5] Sogancioglu et al., "Fairness in AI-Based Mental Health: Clinician Perspectives and Bias Mitigation", AIES 2024
> > > > >
> > > > > [6] Wu et al., "Polyjuice: Generating Counterfactuals for Explaining, Evaluating, and Improving Models", ACL 2021
> > > > >
> > > > > [7] Qian et al., "Perturbation Augmentation for Fairer NLP", EMNLP 2022

---

> ### Author Response · Authors · 2025-11-27
> **Response to Reviewer 7Yw8 (6/N)**
>
> ## Follow-up: Explanations for Human Fairness Auditing
>
> We present results from our human study that investigates how explanations can support human fairness auditing.
>
> | | Attention (undirected) | IxG L2 (undirected) |  Occlusion abs (undirected) | KernelSHAP (directed) | IxG mean (directed) | Occlusion (directed) |
> |---|---|---|---|---|---|---|
> | Fairness correlation | 0.402 | 0.123 | **0.433** | 0.254 | -0.078 | **0.342** |
> | Interpretability | 2.256 | **3.179** | 2.920 | 2.518 | 2.439 | **2.780** | |
>
> Table: Fairness correlation of explanation methods for human fairness auditing and their interpretability. Best scores in each explanation type are marked in **bold**. Higher fairness correlation scores indicate that explanations can better assist humans to detect bias.
>
> Our result show that certain explanation methods, such as Attention, Occlusion, and Occlusion abs, achieve high fairness correlations, suggesting that they can effectively assist humans in detecting bias.
> Across explanation types, undirected explanations appear more helpful.
> For example, Occlusion and Occlusion abs produce the same attribution patterns that differ only in directional encoding, yet participants were better able to identify bias using the undirected variant (Occlusion abs).
> Furthermore, while some methods support both human and automatic bias detection consistently (e.g., Attention and Occlusion abs), others, such as IxG L2, show substantial gaps in performance. This highlights a potential discrepancy between how humans interpret explanations and how our automatic pipeline evaluates them.
>
> We also observe that high interpretability alone does not guarantee better support for human fairness auditing: methods with strong interpretability scores (e.g., IxG L2) still fail to effectively help humans detect bias.
> Finally, 4 out of 6 annotators reported that undirected explanations helped them detect bias more effectively, noting that they introduce less noise and make annotation easier.
>
> | | Attention (undirected) | IxG L2 (undirected) |  Occlusion abs (undirected) | KernelSHAP (directed) | IxG mean (directed) | Occlusion (directed) |
> |---|---|---|---|---|---|---|
> | **All examples** | 0.402 | 0.123 | 0.433 | 0.254 | -0.078 | 0.342 |
> | **Correct predictions only** | *0.572* | *0.202* | *0.602* | 0.217 | *0.029* | *0.451* |
> | **Toxic examples only** | *0.637* | 0.118 | 0.374 | 0.231 | *0.134* | 0.315 |
> | **High interpretability (≥ 3)**| 0.138 | *0.227* | 0.288 | 0.154 | *-0.043* | *0.404* |
>
> Table: Fairness correlation of explanation methods for human fairness auditing under different conditions.
> Higher fairness correlation scores indicate that explanations can better assist humans to detect bias.
> Results in *italics* indicate that the fairness correlation scores are higher than the baseline (all examples).
>
> The table above further analyzes explanation-assisted human fairness auditing under different conditions.
> For correctly predicted examples, explanations generally provide stronger support for bias detection.
> However, the effects of label type and explanation interpretability appear more nuanced and vary across methods.
> Overall, these results suggest that explanation-assisted human fairness auditing is a promising and interesting direction for future work and warrants further investigation.

---

> > ### Comment · Reviewer_7Yw8 · 2025-11-27
> >
> > I would like to thank the author for their massive effort in addressing my comments. Honestly, I didn't reasonably expect my comments would be addressed within such a short time frame. I greatly appreciate the considerable effort! The responses and experiments have adequately addressed my concerns, and integrating them makes the paper much stronger for acceptance. In view of this, I will bump up my scores for this paper.

---

> > > ### Author Response · Authors · 2025-11-27
> > >
> > > Thank you for the prompt feedback. We truly believe your suggestions helped us deepen and broaden the work, and we’re grateful for the positive follow-up.

---

### Official Review · Reviewer_MpuR · 2025-10-31

**Soundness:** 4
**Presentation:** 3
**Contribution:** 3
**Rating:** 6
**Confidence:** 4

**Summary:**

This paper presents a systematic study with a focus on the intersection between fairness and explainability, both central topics in AI ethics. Oriented by three clearly defined research questions, the authors explore the utilities of SOTA feature attribution methods, a particular form of explanations, in machine learning fairness for NLP. Based on the empirical observations, in-depth discussions are carried out, which bring together the standard fairness metrics and the new explanation-oriented perspective. The discussions deliver new insights that provide potential guidelines for future practice in debiasing language models.

**Strengths:**

- This paper presents decent work that is clearly motivated and smoothly unfolds the storytelling, following three research questions that are inherently connected.
- The experiments are well structured, with reasonable choices for the fairness metrics and a collection of feature attribution methods that represent SOTA.
- The studies that compare the (traditional) statistical perspective with the explanation-driven perspective are insightful; a conclusion is reached from each research question, presented in the form of takeaways.

**Weaknesses:**

- The experiments mainly rely on existing methodologies. Although the authors bring standard and explanation-driven debiasing techniques to the same table, no further exploration is conducted for possible combinations across categories.
- While the deliveries are generally insightful, the observations for RQ2 are not sufficiently discussed, rendering a rather hasty conclusion (see the first two points in questions for more details).

**Questions:**

I do not have particular questions regarding the content, as the presentation of the work is clean and clear. Instead, I would like to establish a discussion with the authors, so the questions below serve more for the purpose of brainstorming.

- What do the authors think about the observations for RQ2 beyond the takeaway? Indeed, there are mismatches between the statistical and attribution perspectives; however, directly determining what metrics are responsible is non-trivial. Particularly, there could be “implicit” biased attributions that are exempted from the standard fairness metrics. By “implicit”, I refer to the predictions that are correct but contain falsely attributed features. The current fairness metrics do not consider such implicit biases as long as the prediction results are balanced among demographic groups. These implicit biases can be a problem when attributions to other features are not strong enough to cover them.
- Could a cross-validation be helpful in further revealing the mismatches? For example, by quantifying implicit biases in terms of attributions for correctly predicted samples.
- Could the authors share some thoughts about combining traditional debiasing approaches (the data augmentation perspective) and explanation-driven approaches?
- Since this paper considers different bias categories, have the authors observed any internal connections among different bias categories? A previous work reported indirect impacts during debiasing [1], and I’m wondering whether that is observed in the broader settings considered in this work. Particularly, does debiasing on one category also suppress other forms of bias? Or, does it transfer the biases to other categories while explicitly regulating for one specific type?

The presented work appears complete as it is; yet, deepening the investigations and discussions will certainly enhance its impact.

[1] Cai, Yi, et al. "Power of explanations: Towards automatic debiasing in hate speech detection." *2022 IEEE 9th International Conference on Data Science and Advanced Analytics (DSAA)*. IEEE, 2022.

---

> ### Author Response · Authors · 2025-11-23
> **Response to Reviewer MpuR (1/N)**
>
> We are glad to see that the reviewer recognizes our contributions, and we greatly appreciate your valuable feedback. We have carefully considered all the questions raised and provide our responses and discussions below. **Please note that we are still integrating new experimental results and discussion into the revised manuscript. The update is ongoing, and we plan to upload the revised version by November 26. We will notify the reviewers once it is available and include a general response summarizing all revisions.**
>
> ## [W1 & Q3] Exploring New Bias Mitigation Methods
>
> The primary contribution of our work is to examine the feasibility of using input-based explanations to improve fairness. Accordingly, we focus on basic explanation-based bias detection and mitigation methods, and the development of new mitigation techniques is beyond the scope of this study. Nevertheless, we appreciate the reviewer's excellent suggestion to combine explanation-based approaches with existing debiasing methods. Motivated by this, we conducted preliminary experiments that integrate several pre-processing techniques (group balance, group–class balance, and CDA) with an effective explanation-based debiasing method (InputXGradient L1/L2). The corresponding individual fairness results after combined debiasing and comparison to traditional and explanation-based methods are presented below (lower scores are better).
>
> Comparison to traditional pre-processing/explanation-based debiasing methods (race):
>
> ||Group balance|Group-class balance|CDA|
> |---|---|---|---|
> |IxG L1|0.012 (-4.492/-1.461)|0.000 (-3.048/-1.473)|0.001 (-0.547/-1.473)|
> |IxG L2|2.162 (-2.342/-0.598)|2.110 (-0.938/-0.650)|0.210 (-0.338/-2.550)|
>
> Comparison to traditional pre-processing/explanation-based debiasing methods (gender):
> ||Group balance|Group-class balance|CDA|
> |---|---|---|---|
> |IxG L1|0.005 (-0.594/-0.548)|0.001 (-0.836/-0.552)|0.002 (-0.488/-0.551)|
> |IxG L2|0.308 (-0.291/-0.331)|0.546 (-0.292/-0.093)|0.368 (-0.122/-0.271)|
>
> Tables: Each entry represents the individual fairness score after debiasing with the combination of corresponding pre-processing and explanation methods. Lower scores indicate less bias. Scores in parentheses indicate difference from using only the traditional/explanation-based debiasing methods and negative scores indicate improved debiasing performance.
>
> We observe that the hybrid method achieves better bias mitigation effects than using each debiasing method alone. This improvement is observed in both race and gender bias. We will report these combined-debiasing results in the revised paper and incorporate this direction into the newly added **Limitations and Future Work** section, encouraging more systematic exploration of hybrid approaches in future research.
>
> ## [W2 & Q1] Discussion on RQ2: Fair Model Selection
>
> We acknowledge that drawing a negative conclusion for RQ2 is non-trivial, as it is infeasible to exhaustively enumerate all possible attribution-based metrics and rigorously evaluate their effectiveness for fair model selection. However, our goal in this work is to provide the first empirical study that systematically links fairness and explainability. For this reason, we focus on basic attribution-based metrics rather than exploring more specialized metrics. To avoid misleading readers, we will make more explicit references to the set of attribution-based metrics we evaluate and clarify this limitation in the Limitations and Future Work section.

---

> > ### Author Response · Authors · 2025-11-23
> > **Response to Reviewer MpuR (2/N)**
> >
> > ## [Q1 & Q2] Capturing Implicit Bias
> >
> > Regarding Q1, we agree that "implicit bias" can indeed exist: a biased model may rely on sensitive features or flawed reasoning patterns while still producing the correct prediction. In such cases, the prediction is right for the wrong reasons, and the bias is not reflected in output-level group statistics. However, we argue that *individual fairness* metrics, which measure the change in prediction confidence under identity term perturbations, provide a more appropriate correspondence to input-based attributions. If a model truly uses sensitive features for its prediction, perturbing those features should affect its confidence, even if the predicted label does not flip. Moreover, if an explanation method can faithfully capture such faulty reasoning, it should be effective at identifying biased predictions. This inherently deeper link between input-based explanations and individual fairness helps to explain why, in RQ2, explanation-based metrics correlate better with individual fairness than with group fairness (even though they still do not match the reliability of validation set fairness).
> >
> > As a side note, the discussion above reflects only a simplified view of the relationship between fairness and explanations. For example, if a model does not rely on sensitive features in the original input but does so in the counterfactual variant, the prediction can still demonstrate individual unfairness, though not captured by explanations. Similarly, if the model exhibits the same level of bias in both original and counterfactual inputs, individual fairness metrics will report no bias, while explanations may still indicate biased reasoning. This relationship becomes even more complex in the model selection setup (RQ2), where attribution-based metrics can be affected by the distributional properties of attributions across models (e.g., some models produce spikier or flatter distributions than others). This complexity further motivates our empirical evaluation, rather than relying solely on intuitive assumptions as many earlier studies have done.
> >
> > Regarding Q2, if we understand correctly, the reviewer is asking whether biased attributions can be measured in correctly predicted examples in order to capture implicit bias. Our reliance statistics do reveal such patterns. For illustration, we consider Occlusion explanations on the BERT model with race bias on Civil Comments. We find that although correctly predicted examples have lower reliance on sensitive tokens than incorrectly predicted ones (0.050 vs. 0.086), their reliance on sensitive tokens is still substantially higher than the average reliance across all input tokens (0.050 vs. 0.017). This high reliance on sensitive tokens in correctly predicted examples suggests the presence of implicit bias.
> >
> > We also observe more complex relationships when considering other factors. For example, the reliance gap between sensitive tokens and other tokens is larger in Black examples than White examples, and the reliance difference between correct and incorrect predictions is also larger for Black examples. These patterns indicate a higher degree of both implicit and explicit bias in examples referencing black identities. Similar trends appear when looking at model predictions (e.g., greater bias in toxic predictions), and the findings are consistent across explanation methods and models.
> >
> > While we do observe evidence of implicit bias, quantifying it in a principled way remains challenging, as there is no established reference for how much attribution a sensitive token should receive. We welcome further discussion from the reviewer on how to formalize this measurement and how it might be incorporated into cross-validation.

---

> > > ### Author Response · Authors · 2025-11-23
> > > **Response to Reviewer MpuR (3/N)**
> > >
> > > ## [Q4] Generalization of Debiasing Effects across Bias Types
> > >
> > > In summary, we observe complex generalization behavior in our study. We examined two transfer directions: race → gender and race → religion. Specifically, we evaluated the fairness of BERT models debiased on the *source* bias category (using both traditional and explanation-based methods) on examples from the *target* category, and compared their performance to the default model. All experiments were conducted on Civil Comments, and for simplicity we report individual fairness results only.
> > >
> > > Race to gender:
> > >
> > > ||Default|Group balance|Group-class balance|CDA|Dropout|Attention entropy|Causal debias|Attention|Attn rollout|Attn flow|Grad L1|Grad L2|IxG L1|IxG L2|IntGrad L1|IntGrad L2|Occlusion|
> > > |---|---|---|---|---|---|---|---|---|---|---|---|---|---|---|---|---|---|
> > > |$\text{Avg}_\text{iu}$|1.03|1.32 (+0.29)|1.23 (+0.20)|1.25 (+0.22)|1.07 (+0.04)|1.10 (+0.07)|0.79 (-0.24)|2.26 (+1.23)|1.75 (+0.71)|1.46 (+0.42)|1.66 (+0.63)|1.46 (+0.43)|1.35 (+0.32)|1.39 (+0.36)|1.35 (+0.32)|1.47 (+0.43)|1.44 (+0.41)|
> > >
> > > Race to religion:
> > >
> > > ||Default|Group balance|Group-class balance|CDA|Dropout|Attention entropy|Causal debias|Attention|Attn rollout|Attn flow|Grad L1|Grad L2|IxG L1|IxG L2|IntGrad L1|IntGrad L2|Occlusion|
> > > |---|---|---|---|---|---|---|---|---|---|---|---|---|---|---|---|---|---|
> > > |$\text{Avg}_\text{iu}$|1.94|2.08 (+0.14)|2.04 (+0.10)|1.19 (-0.75)|1.85 (-0.09)|1.72 (-0.22)|2.21 (+0.27)|1.18 (-0.76)|1.27 (-0.67)|2.42 (+0.48)|1.70 (-0.24)|1.83 (-0.11)|1.30 (-0.65)|1.81 (-0.13)|1.22 (-0.72)|1.52 (-0.42)|2.46 (+0.52)|
> > >
> > > Tables: Generalization of bias mitigation effects across bias types. Lower $\text{Avg}_\text{iu}$ scores indicate less bias in the model.
> > >
> > > We find that in the race → gender direction, most debiasing methods actually amplify bias, whereas in the race → religion direction, debiasing effects generalize for many methods. Moreover, even within the race → religion direction, the fairness outcomes on the target category vary substantially across methods. Taken together, these findings indicate that the generalization of bias mitigation is a complex phenomenon influenced by multiple factors, including the specific source and target categories and the debiasing method applied. Our observations are consistent with prior work showing that the transferability of debiasing effects is inherently multi-factorial and unpredictable [1, 2].
> > >
> > > # References
> > >
> > > [1] Zhao et al., "Men Also Do Laundry: Multi-Attribute Bias Amplification", PMLR 2023
> > >
> > > [2] Elsafoury and Katsigiannis, "On Bias and Fairness in NLP: Investigating the Impact of Bias and Debiasing in Language Models on the Fairness of Toxicity Detection", Arxiv 2024

---

### Official Review · Reviewer_K4TD · 2025-11-03

**Soundness:** 3
**Presentation:** 2
**Contribution:** 1
**Rating:** 4
**Confidence:** 4

**Summary:**

The paper examines the relationship between input-based explanations and fairness in the context of hate speech detection. The authors show that the tested XAI techniques can effectively identify biased predictions in both encoder-only and decoder-only models, and can also assist in guiding bias mitigation.

**Strengths:**

The paper is well-written and easy to read, providing all the details necessary for reproducibility in the appendices. (On this point, I suggest double-checking that all appendices are properly referenced in the main body of the paper.)
The experimental setup is comprehensive, as the authors evaluate multiple models, datasets, XAI methods, and fairness techniques.
The research questions addressed in the paper are compelling, and the contribution is well-positioned at the intersection of XAI and fairness, a space where existing literature remains limited.

**Weaknesses:**

Consider expanding the motivation for incorporating both fairness and explainability in hate speech detection, particularly in the Introduction and Related Work sections, to better contextualize the importance of these aspects.
It would be helpful to further elaborate on the rationale behind focusing specifically on input-based explanations.
Please clarify the distinction between hate speech detection and abusive language detection. While the study focuses on hate speech, the labels in Notations (toxic vs. non-toxic) appear to align more closely with the abusive language perspective. Again in Notations it would be useful to explicitly define what is meant by social bias. Additionally, the sensitive token reliance scores would benefit from clearer motivation and stronger connections to related literature. For instance, is this a new measure introduced by the authors, or is it adapted from prior work? Furthermore, when the hateful nature of a sentence stems directly from a sensitive token, how should reliance on that token be interpreted? Clarifying these aspects would strengthen the reader’s understanding of what the metric captures.
The contributions of the paper could be stated more explicitly. If the main contribution is an evaluation framework, it might be helpful to revise Section 3 to better highlight its novelty and unique aspects. Moreover, including a visual overview or workflow diagram of the experimental setup could greatly improve clarity and accessibility.
Consider adding a short Future Work section to outline possible extensions. Additionally, frequent use of red and green in tables can be visually distracting: using bold text to indicate the best results might improve readability. Some figures could also be refined to enhance their legibility.

**Questions:**

It would be valuable to discuss the generalizability of the proposed approach beyond hate speech detection. To what extent can the methods and findings be applied to other NLP tasks or domains? Clarifying this would help readers better understand the broader applicability and impact of the work.

---

> ### Author Response · Authors · 2025-11-23
> **Response to Reviewer K4TD (1/N)**
>
> Thank you for your time and effort in reviewing our work and for providing valuable feedback. We address your concerns point by point below:
> 1. Generalizability beyond Hate Speech Detection
> 2. Sharpening Our Contributions
> 3. Defining Hate Speech and Social Bias
> 4. Motivating Fairness and Input-Based Explanations in Hate Speech Detection
> 5. Clarifying Metrics and Experimental Setups
> 6. Workflow Diagram and Result Presentation
> 7. Limitations and Future Work
>
> Please find our detailed responses below. We apologize for the length of this response, as we aim to address your questions as clearly and comprehensively as possible. **Please note that we are still integrating new experimental results and discussion into the revised manuscript. The update is ongoing, and we plan to upload the revised version by November 26. We will notify the reviewers once it is available and include a general response summarizing all revisions.**
> ## 1. Generalizability beyond Hate Speech Detection
> Reviewer: *It would be valuable to discuss the generalizability of the proposed approach beyond hate speech detection. To what extent can the methods and findings be applied to other NLP tasks or domains? Clarifying this would help readers better understand the broader applicability and impact of the work.*
>
> ---
> To verify generalizability of our findings, we have evaluated explanation-based bias detection (RQ1) on an additional task, namely **sentiment analysis**, using the Twitter Sentiment dataset. Specifically, we selected 1000 gender-related examples (500 referencing males and 500 referencing females) and run explanation-based bias detection (RQ1) on them. Below we report the fairness correlation scores and numbers of significant group-class category (in parentheses) for each explanation method on zero-shot Llama3.2-3B and Qwen3-4B models, and the complete results will be included in the revised manuscript.
>
> Llama3.2-3B (zero-shot)
> ||Attention|Attn rollout|Attn flow|Grad mean|Grad L2|IxG mean|IxG L2|IntGrad mean|IntGrad L2|DeepLift mean|DeepLift L2|Occlusion|Occlusion abs|KernelSHAP|ProgInfer|
> |---|---|---|---|---|---|---|---|---|---|---|---|---|---|---|---|
> |Fairness correlation|0.368 (4)|0.174 (2)|0.084 (0)|0.150 (2)|0.289 (4)|0.283 (3)|0.303 (4)|0.239 (3)|0.337 (4)|0.186 (2)|0.297 (4)|0.401 (3)|0.342 (4)|0.081 (0)|0.051 (1)|
>
> Qwen3-4B (zero-shot)
> ||Attention|Attn rollout|Attn flow|Grad mean|Grad L2|IxG mean|IxG L2|IntGrad mean|IntGrad L2|DeepLift mean|DeepLift L2|Occlusion|Occlusion abs|KernelSHAP|ProgInfer|
> |---|---|---|---|---|---|---|---|---|---|---|---|---|---|---|---|
> |Fairness correlation|0.135 (2)|0.073 (1)|0.153 (2)|0.299 (3)|0.294 (3)|0.200 (3)|0.286 (3)|0.092 (2)|0.114 (1)|0.099 (1)|0.279 (3)|0.555 (4)|0.517 (4)|0.050 (0)|0.022 (0)|
>
> Tables: Fairness correlation results for Llama3.2-3B and Qwen3-4B on Twitter Sentiment. Higher scores indicate that an explanation method is more effective at detecting biased predictions.
>
> We observe similar patterns in other classification settings, where certain explanation methods (e.g., occlusion-based and L2-based approaches) also achieve high fairness correlation scores. This suggests that our findings likely extend beyond the hate speech detection task.
> ## 2. Sharpening our Contributions
> Reviewer: *The contributions of the paper could be stated more explicitly. If the main contribution is an evaluation framework, it might be helpful to revise Section 3 to better highlight its novelty and unique aspects.*
>
> ---
> As stated in the abstract and introduction, our work presents the first empirical study that quantitatively and comprehensively examines whether and how input-based explanations can be used to promote fairness in NLP systems. We also offer practical guidance on which explanation methods align best with different fairness goals. We will make these contributions more explicit in the revised manuscript.
>
> We would like to clarify that we do not position our work as a full evaluation framework. As the first large-scale empirical investigation in this area, some of our metrics and experimental setups, such as the attribution-based metrics in RQ2 and the fairness-balanced metric in RQ3, are intentionally basic and simple. This is because there are no established standards yet, and our primary objective is to explore the potential of explanation-based approaches for bias detection and mitigation, rather than to identify the strongest possible methods.

---

> > ### Author Response · Authors · 2025-11-23
> > **Response to Reviewer K4TD (2/N)**
> >
> > ## 3. Defining Hate Speech and Social Bias
> > We agree that clarifying key concepts would make our contributions more accessible. In the revised manuscript, we will add a section in the Appendix defining hate speech and social bias.
> >
> > Reviewer: *Please clarify the distinction between hate speech detection and abusive language detection. While the study focuses on hate speech, the labels in Notations (toxic vs. non-toxic) appear to align more closely with the abusive language perspective.*
> >
> > ---
> > ```
> > We follow Fortuna and Nunes [1] in defining hate speech as a specific form of abusive or toxic language that targets and attacks protected or identifiable social groups. Under this view, hate speech is a subset of abusive language. This definition is consistent with widely adopted formulations in prior work on hate speech detection (e.g., [2, 3]). Because our study focuses on fairness and, in particular, analyzes model behavior on examples involving specific social groups (race, gender, religion), this standard definition of hate speech aligns well with the scope and goals of our work.
> > ```
> > We still use the toxic vs. non-toxic labels following the terminology used in the Civil Comments [4] and Jigsaw [5] datasets. Although these datasets include multiple subtypes of abusive content, they group them under the broader notion of toxicity.
> >
> > Reviewer: *Again in Notations it would be useful to explicitly define what is meant by social bias.*
> >
> > ---
> > ```
> > Following the conceptualization of Blodgett et al. [6], we define social bias as the presence of stereotypical associations between social groups and certain attributes, as well as disparities in how these groups are treated as a result. Such biases can lead to both representational harms (e.g., demeaning or misrepresenting targeted groups) and allocational harms (e.g., unfair distribution of opportunities or resources). Given the potential for NLP systems to reproduce or amplify these harms, and their growing influence in everyday life, it is essential to detect and mitigate social bias in these models.
> > ```
> > ## 4. Motivating Fairness and Input-Based Explanations in Hate Speech Detection
> > Reviewer: *Consider expanding the motivation for incorporating both fairness and explainability in hate speech detection, particularly in the Introduction and Related Work sections, to better contextualize the importance of these aspects.*
> >
> > ---
> > In the revised paper, we will include an extended discussion of fairness and explainability in hate speech detection. We hope this will better convey the importance of both fairness and explainability in the task and clarify how the extensive prior work on bias and interpretability in this area motivates our choice to center our study on hate speech detection. If you wish, we can also present these sections directly in our response.
> >
> > Reviewer: *It would be helpful to further elaborate on the rationale behind focusing specifically on input-based explanations.*
> >
> > ---
> > We will add the following paragraph to clarify why we focus on input-based explanations in this work.
> > ```
> > We focus on input-based explanations because they offer the most direct view into which parts of the input influence a model's prediction [7], and they have long been regarded as central tools for fairness auditing in ML [8, 9]. Their methodological diversity also makes them an ideal testbed for our study, enabling a comprehensive examination of whether and how explanations can improve fairness [10]. In addition, both automated and human-centric metrics for evaluating explanation properties (e.g., faithfulness, interpretability) are well established [11, 12, 13]. This allows us to analyze how these properties relate to an explanation method's (in)effectiveness in fairness-related tasks. Finally, input-based explanations are often mandated by laws, such as the EU Artificial Intelligence Act, making it practically important to understand how their use interacts with fairness considerations.
> > ```

---

> > > ### Author Response · Authors · 2025-11-23
> > > **Response to Reviewer K4TD (3/N)**
> > >
> > > ## 4. Clarifying Metrics and Experimental Setups
> > > Reviewer: *Additionally, the sensitive token reliance scores would benefit from clearer motivation and stronger connections to related literature. For instance, is this a new measure introduced by the authors, or is it adapted from prior work? Clarifying these aspects would strengthen the reader's understanding of what the metric captures.*
> > >
> > > ---
> > > Our sensitive token reliance measure is inspired by Dimanov et al. [14], who attribute model predictions to a single sensitive feature. We extend their formulation to handle multiple sensitive features by computing attribution scores for all sensitive tokens and taking the maximum absolute value. This captures the strongest reliance signal in more realistic multi-token scenarios. We will include the reference and discuss our difference from existing work in the revised paper.
> > >
> > > Reviewer: *Furthermore, when the hateful nature of a sentence stems directly from a sensitive token, how should reliance on that token be interpreted?*
> > >
> > > ---
> > > Our sensitive token vocabulary and data sampling process are specifically designed to avoid this issue. In particular, the sensitive token vocabulary excludes all offensive or derogatory terms, ensuring that the presence of a sensitive token does not itself make a sentence hateful. Moreover, as noted in the **Details on experimental setups** section in the Appendix, we remove any examples containing insulting references to social groups. The complete sensitive token and forbidden token lists for each group are provided under utils/vocabulary in our supplementary materials.
> > > ## 6. Workflow Diagram and Result Presentation
> > > Reviewer: *Moreover, including a visual overview or workflow diagram of the experimental setup could greatly improve clarity and accessibility.*
> > >
> > > ---
> > > We appreciate the reviewer's helpful suggestion. In the revised manuscript, we will add a pipeline overview figure that illustrates the workflow for each research question. We hope this addition will help readers more easily grasp the structure of our study and its key contributions.
> > >
> > > Reviewer: *Additionally, frequent use of red and green in tables can be visually distracting: using bold text to indicate the best results might improve readability. Some figures could also be refined to enhance their legibility.*
> > >
> > > ---
> > > Thank you for the insightful feedback. Since our goal is to examine whether existing methods are effective for bias mitigation, instead of identifying the best performing method, it is crucial for us to highlight whether each method performs better or worse than the baseline. For this reason, we will retain color coding in the tables and figures. If the reviewer has alternative color pairs than green/red that may improve readability, we would be happy to adopt them.
> > >
> > > We have changed some figures to enhance their clarity and readability. For example, we have updated the "Val fairness" entries in Figures 2 and 3 (RQ2) to "Val fairness (baseline)" and applied consistent colors and styles across both figures to clearly indicate the baseline method. For Figure 4 (RQ3), we added left and right y-axes for each subplot to make it explicit that accuracy and fairness are displayed on different scales. These revisions have also been applied to all corresponding figures in the Appendix and will be included in the revised manuscript. We hope these changes enhance readability and help readers more easily interpret our results.
> > > ## 7. Limitations and Future Work
> > > Reviewer: *Consider adding a short Future Work section to outline possible extensions.*
> > >
> > > ---
> > > We will include a **Limitations and Future Work** section to discuss the limitations that the reviewers raised and those identified in our experiments. These limitations primarily concern the relatively narrow experimental setups in the current work, the limited exploration of connections to group fairness metrics, and the absence of a universal recommendation for selecting explanation methods. We also encourage future work to extend the study to more diverse use cases and to explore hybrid approaches that combine explanation-based bias detection and mitigation with other explanation techniques, established debiasing methods, and human oversight. We will also include preliminary results to address the identified limitations and to motivate the future work directions discussed in the paper.

---

> > > > ### Author Response · Authors · 2025-11-23
> > > > **Response to Reviewer K4TD (4/N)**
> > > >
> > > > # References
> > > > [1] Fortuna and Nunes, "A Survey on Automatic Detection of Hate Speech in Text", ACM Computing Surveys 2018
> > > >
> > > > [2] Davidson et al., "Automated Hate Speech Detection and the Problem of Offensive Language", AAAI 2017
> > > >
> > > > [3] Nobata et al., "Abusive Language Detection in Online User Content", WWW 2016
> > > >
> > > > [4] Borkan et al., "Nuanced Metrics for Measuring Unintended Bias with Real Data for Text Classification", WWW 2019
> > > >
> > > > [5] https://www.kaggle.com/competitions/jigsaw-unintended-bias-in-toxicity-classification
> > > >
> > > > [6] Blodgett et al., "Language (Technology) is Power: A Critical Survey of "Bias" in NLP", ACL 2020
> > > >
> > > > [7] Wang and Yin, "Are Explanations Helpful? A Comparative Study of the Effects of Explanations in AI-Assisted Decision-Making", IUI 21
> > > >
> > > > [8] Balkir et al., "Challenges in Applying Explainability Methods to Improve the Fairness of NLP Models", TrustNLP 2022
> > > >
> > > > [9] Deck et al., "A Critical Survey on Fairness Benefits of Explainable AI", FAccT 2024
> > > >
> > > > [10] Lyu et al., "Towards Faithful Model Explanation in NLP: A Survey", CL 2024
> > > >
> > > > [11] DeYoung et al., "ERASER: A Benchmark to Evaluate Rationalized NLP Models", ACL 2020
> > > >
> > > > [12] Jacovi and Goldberg, "Towards Faithfully Interpretable NLP Systems: How Should We Define and Evaluate Faithfulness?", ACL 2020
> > > >
> > > > [13] Lage et al., "An Evaluation of the Human-Interpretability of Explanation", Arxiv 2019
> > > >
> > > > [14] Dimanov et al., "You Shouldn't Trust Me: Learning Models Which Conceal Unfairness From Multiple Explanation Methods", ECAI 2020

---

### Author Response · Authors · 2025-11-27
**Response to All Reviewers**

We would like to thank all reviewers for their very constructive feedback. We are encouraged by the positive recognition of our contributions, especially regarding the comprehensiveness of our experimental setups and the clarify of writing. This also aligns with the intended contribution of our paper, namely presenting the first quantitative and comprehensive study examining whether and how input-based explanations can help promote fairness in NLP systems.

To address the reviewers' concerns and strengthen the work, we added new experiments, elaborated on key discussions, and revised several sections. For ease of review, all new experimental results are placed at the beginning of the Appendix. In the camera-ready version, we will reorder them to follow their sequence of appearance in the main text. Below we summarize our revisions:

## **Additional Experimental Results Suggested by Reviewers**

1. **Added GPT-OSS-120B for LLM-as-a-judge**: New results (Section 6) further confirm explanation-based bias detection remains more reliable than LLM-as-judge (*Reviewers K4TD/EN8y*).

2. **Extended to Qwen3-8B (larger LLM)**: Added fairness evaluation + RQ1 + RQ2 results (Appendix H/I/K). Findings align with observations in the main text, further supporting our conclusions (*Reviewer 7Yw8*).

3. **Validated generalization to new task, alignment types, and vocabulary settings**: Added experiments on a sentiment analysis task, two differently aligned models, and low-coverage sensitive token vocabulary (Appendix C). All results show consistent effectiveness of explanation-based bias detection/mitigation, demonstrating generalizability of findings (*Reviewers K4TD/7Yw8*).

4. **Human study for explanation-based fairness auditing**: Added a human study (Appendix D) that shows explanations can assist human bias detection (*Reviewer 7Yw8*).

5. **Hybrid debiasing methods**: Combined pre-processing methods and explanation-based debiasing methods (Appendix E) and found that hybrid methods outperform either strategy alone (*Reviewer MpuR*).

6. **Time and GPU memory costs for each explanation method**: Appendix F (*Reviewer 7Yw8*).

## **Experiments for Comprehensiveness**

7. **Updated Occlusion + Integrated Gradients results in RQ3**: Selected a more appropriate hyperparameter range for Occlusion (Footnote 4, Figure 5, Appendix L) and reported results on reducing race bias using Integrated Gradients (Appendix L). New results are consistent with the original conclusions.

8. **Added two recent explanation methods**: DecompX [1] (encoder-only) and Progressive Inference (ProgInfer) [2] (decoder-only), integrated into all relevant figures and tables.

## **Expanded Discussion & Improved Presentation**

9. **Added Limitations and future work**: Discussed limitations and potential future directions raised by reviewers and identified through our study (Appendix B). Preliminary results addressing these limitations and illustrating potential future directions are presented, as summarized above (*Reviewers K4TD/EN8y*).

10. **Clarified Key Concepts**: Added clear definitions of hate speech and social bias in Appendix A (*Reviewer K4TD*).

11. **Expanded discussion of related work**: Added an overview of fairness + explainability research in hate speech detection (Appendix A) to clarify our task choice and motivation (*Reviewer K4TD*).

12. **Clarified motivation for input-based explanations**: Appendix A (*Reviewer K4TD*).

13. **Added workflow diagram**: Figure 1 (*Reviewer K4TD*).

14. **Improved Figure Readability**: Figure 3/4/5 and relevant figures in Appendix K/L  (*Reviewers K4TD/EN8y*).

## **Fixed Errors**

15. **Fixed error in MRR@1 computation**: Figure 4 and Appendix K. The updated results still support our conclusion that explanations are not reliable for fair model selection.

## **Minor Revisions**

16. **Sharpened contribution**: Explicitly clarified research questions, key findings, and contributions in abstract, introduction, related work and conclusion (*Reviewer K4TD*).

17. **Clarified sensitive token reliance**: Added reference to clarify the metric and its difference from previous work (*Reviewer K4TD*). Compared multiple aggregation methods and confirmed similar findings, as noted in Footnote 1 (*Reviewer 7Yw8*).

18. **Added attribution-metric reference**: Footnote 3 now clarifies which attribution metrics were examined in RQ2 (*Reviewer MpuR*).

**We are glad to see that Reviewer 7Yw8 raised their score from 6 to 8, and we hope that our revisions also address the concerns of the other reviewers, even though they were unable to respond due to the unexpected situation. We sincerely thank all reviewers again for their invaluable feedback.**

[1] Modarressi et al., "DecompX: Explaining Transformers Decisions by Propagating Token Decomposition", ACL 2023

[2] Kariyappa et al., "Progressive Inference: Explaining Decoder-Only Sequence Classification Models Using Intermediate Predictions", ICML 2024

---

### Meta-Review · Area_Chair_kK4X · 2025-12-22

**Summary:**

The paper studies the relationship between input-based explanations and fairness of NLP models in the context of hate speech detection. Overall, reviewers were favorable, with reviewers praising the writing and exhaustiveness of the analysis.

Some concerns the lead to a rating of 4 were related to motivation and positioning (Reviewer K4TD). Other minor concerns included clarify of some parts of the experiments (e.g. due to dense exposition).

**Reviewer Concerns:**

The authors provided an exhaustive rebuttal. Reviewer 7Yw8 expressed satisfaction with the response and willingness to raise their score.

**Reviewer Scores:**

Reviewer 7Yw8 explicitly stated they will raise. Reviewer K4TD did not respond before the discussions were closed. However, their concerns were mainly related to positioning, motivation and exposition and the authors prepared an extensive response addressing these. Therefore, it is likely that Reviewer K4TD could have raised their score.

---

### Decision · Program_Chairs · 2026-01-26

Accept (Poster)